# SSVPO: Effective Step-Level Credit Assignment for RL Training of Language Models

**Yugu Li & Zehong Cao** [*]
School of CSIT, Adelaide University
Adelaide, SA 5000, Australia
`{yugu.li,jimmy.cao}@adelaide.edu.au`

**Jianglin Qiao**
ACFR, The University of Sydney
Camperdown, NSW 2050, Australia
School of CSIT, Adelaide University
Adelaide, SA 5000, Australia
`jianglin.qiao@sydney.edu.au`

**Siyi Hu**
School of EECMS, Curtin University
Bentley, WA 6102, Australia
School of CSIT, Adelaide University
Adelaide, SA 5000, Australia
`siyi.hu@curtin.edu.au`

## Abstract

Language models have shown strong performance on mathematical reasoning tasks. Post-training with outcome-based reinforcement learning (RL) can further enhance reasoning but is inefficient because it relies solely on final rewards. Recent credit assignment–based RL methods provide intermediate feedback, yet they often struggle to fairly evaluate each step's importance, especially in partially correct reasoning chains. We propose Sequential Shapley Value Policy Optimization (SSVPO), a step-level credit assignment framework inspired by multi-agent RL. SSVPO introduces an insertion MDP and Sequential Shapley Values (SSV), which measure each step's marginal contribution by reordering reasoning steps into alternative chains, ensuring fair credit assignment to all possible steps. By identifying steps with zero credit, SSVPO can shorten reasoning chains to improve training efficiency. We further provide a theoretical proof that SSV fairness to allocate credits and demonstrate that SSV as the new advantage baseline is consistent with Proximal Policy Optimization (PPO). Across 7 benchmarks, SSVPO outperforms state-of-the-art RL methods, both outcome-based (RLOO, GRPO, DAPO) and credit assignment–based (VinePPO, SPO), achieving up to an 11.6% gain in accuracy, an 18.1% reduction in token usage, and a 1.6× improvement in reasoning efficiency over vanilla methods. Our findings highlight that SSVPO provides effective step-level credit assignment, advancing post-training LLM reasoning performance while reducing token budgets.

## 1 Introduction

Lightweight large language models (LLMs), such as Qwen3 (Yang et al., 2025), DeepSeek-R1 (Guo et al., 2025a), and Gemini-2.5 (Comanici et al., 2025) with fewer than 7B parameters, have demonstrated comparable reasoning performance to much larger models. These compact models can be widely applied to many reasoning tasks, with mathematical reasoning serving as a critical benchmark (Hendrycks et al., 2021; Cobbe et al., 2021; Mathematical Association of America, 2023; 2024; 2025). Mathematics requires multi-step logical reasoning chains, handling symbolic manipulation, and solving structured problems, making it a rigorous testbed for evaluating a language model's reasoning capability (Wang & Lu, 2023; Forootani, 2025; Patil & Jadon, 2025).

For post-training LLMs (e.g., fine-tuning) approaches, particularly reinforcement learning (RL) with outcome-based methods (e.g., reinforcement fine-tuning), have shown promise in enhancing mathematical reasoning performance from 2024 (Gao et al., 2024; Lyu et al., 2025; Han & Zhang, 2025),

---

[*]Corresponding Author.

such as Reward Leave-One-Out (RLOO) (Ahmadian et al., 2024), Group Relative Policy Optimization (GRPO) (Shao et al., 2024), and Dynamic Sampling Policy Optimization (DAPO) (Yu et al., 2025). Unlike earlier work on supervised fine-tuning (Zelikman et al., 2022; Yu et al., 2023; Yue et al., 2023), outcome-based RL relies purely on a final correctness reward signal, avoiding biases introduced by human preference feedback. This enables LLMs to tackle challenging math problems more effectively, such as algebraic manipulation, equation solving, and those tasks involving geometry, number theory, and combinatorics (Yang et al., 2024; Zou et al., 2024). However, outcome-based RL has significant limitations in training efficiency: rewards are only granted at the end of the reasoning chain when the final answer is correct (Lightman et al., 2023; Uesato et al., 2022; Gao et al., 2024). There is no explicit signal provided to intermediate steps that help measure their contribution to the full reasoning process.

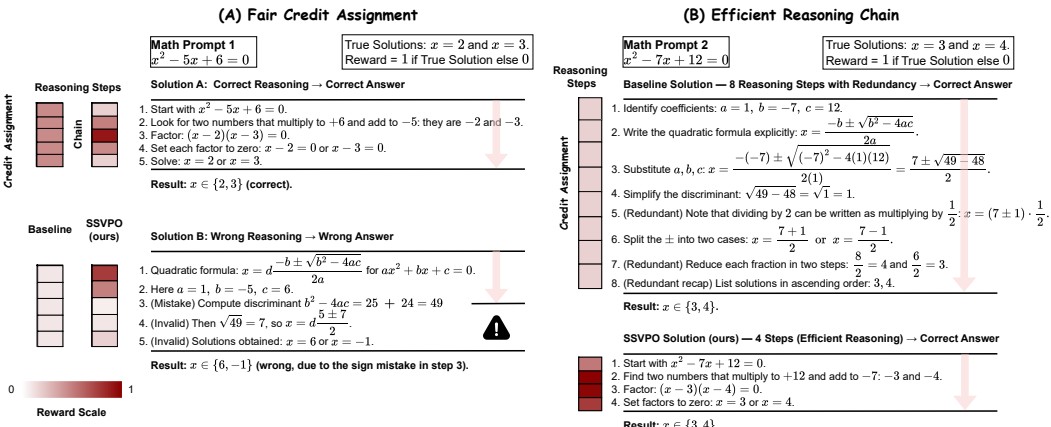

Figure 1: Overview of the differences between existing RL methods and our proposed SSVPO in mathematical reasoning tasks. (A) Baseline methods evenly distribute credit across all steps in a reasoning chain, equivalent to outcome-based final rewards. In contrast, SSVPO assigns credit according to each step's importance, ensuring fair evaluation and allowing correct early or intermediate steps to receive positive credit even when the final answer is wrong. (B) Baseline methods explore all possible reasoning chains without emphasizing key steps, whereas SSVPO focuses training on important steps identified through fair credit assignment, reducing chain length (e.g., from 8 to 4 steps) while still producing the correct answer.

As a result, post-training LLMs tend to treat all reasoning steps as equally important for obtaining the final answer, crediting the same reward to each step. This often leads them to elongate reasoning chains to maximize the reward accumulation, thereby generating many redundant steps. A further consequence of these elongated chains is the risk that, as the number of reasoning steps grows, LLMs may fail to model coherent reasoning paths and instead directly memorize final answers (Wu et al., 2025; Sui et al., 2025; Wang et al., 2025; Li et al., 2025a). This becomes a significant problem when applying the model to unseen mathematical problems, where generalization ability is a must.

To mitigate this issue and enhance LLMs' reasoning efficiency, very recent studies such as VinePPO (Kazemnejad et al., 2025), Segment Policy Optimization (SPO) (Guo et al., 2025b), and Group Token Policy Optimization (GTPO) (Tan & Pan, 2025), have introduced credit assignment for post-training. These credit assignment-based RL methods aim to decompose the outcome reward and allocate intermediate rewards to reasoning steps, using techniques such as Monte Carlo returns, value differences between adjacent steps, and dynamic entropy weighting. While they show empirical gains in reasoning performance, they lack a fair estimation of step-level rewards derived from the final reward.

> **Key Challenge:** *Design a credit assignment method with theoretical guarantees that faithfully capture each reasoning step's marginal contribution, such that important steps can be leveraged to improve training efficiency in subsequent RL training of LLMs.*

Inspired by prior work on fair credit assignment using Shapley values (Li et al., 2021; Wang et al., 2022a; 2020; Han et al., 2022) in multi-agent reinforcement learning (MARL) (Foerster et al., 2018;

Sunehag et al., 2018; Rashid et al., 2020; Wei et al., 2022; Li et al., 2025b), we address this gap by modeling the reasoning process as a sequence of agents, where each step in a Chain of Thought (CoT) (Yao et al., 2023; Stern et al., 2019; Lightman et al., 2023) is treated as an individual agent. To achieve this, we design a new form of Shapley value that, unlike the classical formulation which assumes independent and exchangeable participants, extends from the spatial domain in MARL to the temporal domain of reasoning, enabling exploration of all possible combinations of reasoning steps to be new chains for computing their marginal contributions while explicitly capturing sequential dependencies and order sensitivity. This provides a theoretically fair credit allocation for each step when reordered into new chains. We hypothesize that such fair step-level credit assignment can improve reasoning efficiency over baseline methods and mitigate overthinking issues (e.g., redundant tokens) commonly observed in outcome-based RL. A detailed review of outcome-based and credit assignment RL training for LLMs, along with the adaptation of original credit assignment methods from MARL to LLMs, is provided in **Appendix A**.

Therefore, in this paper, we develop **Sequential Shapley value Policy Optimization (SSVPO)** to achieve two goals: (1) Fair step-level credit assignment via the proposed Sequential Shapley value (SSV), which provides theoretical guarantees by allocating credit to each reasoning step based on its marginal contribution, and is applicable even to partially correct reasoning chains; and (2) Efficient reasoning through policy optimization, where SSV acts as a new value baseline for the advantage estimator. This ensures that steps with high marginal contributions are prioritized during training, leading to improved sample efficiency and shorter reasoning chains, as illustrated in Figure 1.

Our contributions are summarized as follows:

- **Fair credit assignment for each reasoning step.** We offer an effective step-level credit assignment method (SSV) for outcome-based RL training of LLMs, which fairly evaluates each step's marginal contribution by constructing new chains from the insertion MDP.
- **Effective advantage estimator for RL training.** We propose a new advantage estimator (SSVPO) that leverages the SSV advantage function to guide policy optimization, providing an unbiased, minimal-variance value baseline for stable convergence and enabling shorter, more effective reasoning chains.
- **Superior benchmark performance.** On 7 mathematical reasoning benchmarks, our proposed method achieves superior performance and efficiency compared to existing outcome-based and credit assignment RL methods, which improves accuracy by 11.6%, reduces token usage by 18.1%, and achieves a 1.6-fold gain in reasoning efficiency over vanilla methods.

## 2 METHOD

Our proposed SSVPO method introduces a new RL post-training paradigm that ensures the credit decomposed from the final reward is fairly allocated to each reasoning step under outcome-based RL settings. This fairness enables the formulation of a new advantage function for fine-tuning (e.g., policy optimization), thereby improving both reasoning performance and efficiency.

### 2.1 SSVPO WORKFLOW

Given a math prompt, we first use the vanilla model to generate rollouts, then apply an extraction-and-aggregation procedure that compresses long reasoning chains (Appendix G.2) by identifying and merging intermediate reasoning steps. This reduces the number of steps involved in reordering, lowers computational overhead, and removes the final answer sentence from each reordered chain to prevent direct access to the ground-truth answer. This forms the SSVPO pre-stage workflow, followed by the three formal stages (Stages 1–3) of reasoning and training illustrated in Figure 2.

In Stage 1, by using the extracted steps, we construct reordered chains based on the Insertion MDP (Sec. 2.2). In SSVPO, step reordering is not used to generate new trajectories. Instead, permutations are used solely to construct alternative prompts. This process does not rely on the language model's autoregressive next-token generation; rather, it enables modelling the marginal contribution of each step under different chain orders. In Stage 2, we feed these reordered chains to the LLM as prompts, require it to answer directly from the given chain, and evaluate correctness to obtain rewards. These rewards are then used to compute the Sequential Marginal Contribution (SMC) (Sec.2.2) and subsequently the Sequential Shapley Value (SSV) (Sec. 2.3) for each reasoning step. In Stage 3, the

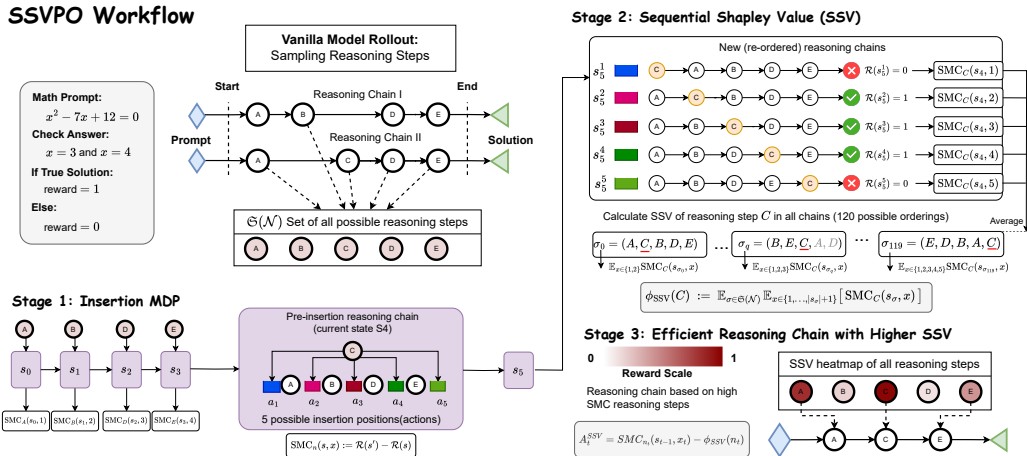

Figure 2: Overview of the proposed SSVPO workflow. Given a math prompt, the LLM vanilla model first generates reasoning steps through rollout. The SSVPO workflow then proceeds in three stages: (1) represent the reasoning process as an insertion-based MDP, enabling the creation of new (re-ordered) chains by re-positioning the reasoning steps; (2) compute the Sequential Shapley value (SSV) for each step across multiple new chains, assigning fair credit by averaging its Sequential Marginal Contributions (SMC); and (3) fine-tune LLMs vanilla model via policy optimization, where the SSV advantage serves as a new value baseline to retain steps with higher contributions, ultimately yielding shorter and more effective reasoning chains that improve answer correctness.

SSV is used as a step-level baseline, combined with its corresponding SMC to produce a fair advantage estimate (Sec. 2.4). This advantage is then used in PPO-style policy updates, providing more granular step-level credit assignment within the standard Reinforcement Learning from Verifiable Rewards (RLVR) framework.

## 2.2 Modeling LLMs Reasoning Process as an Insertion MDP

The reasoning process in language generation models is often formulated as a Markov Decision Process (MDP) (Kazemnejad et al., 2025; Guo et al., 2025b), where each token (or reasoning step) is appended sequentially to the end of the current sequence (chain), and the state transition is defined by deterministic concatenation. However, under the standard next-token prediction perspective, such an MDP only cares about what token (or step) is generated at the end of the sequence, and the final reward is assigned only once after the entire reasoning chain is completed, making it difficult to distinguish the specific contribution of each intermediate step across different chains. To address this limitation, we propose an "Insertion MDP" that operates only on the already generated candidate steps: in this new MDP, a state is a current reasoning chain, an action is to insert a candidate step at an arbitrary position in this chain, and by comparing the rewards obtained before and after inserting this step, we compute the marginal contribution of this step under the given context and insertion position. Note this new Insertion MDP itself does not involve the language model's generation process; instead, it is a credit-assignment model that is specifically designed to finely model and estimate marginal contributions through reordering steps. The insertion process continues until all steps have been considered, thereby generating all possible re-ordered new chains. This formulation enables the estimation of each step's marginal contribution by evaluating its role across chains with different insertion orders. Importantly, these insertions are independent of semantics: we do not require the reordered chains to maintain a coherent or human-readable reasoning structure. Instead, whenever the model is still able to produce the correct answer from such a reordered chain, we interpret this as evidence that the permutation reveals an additional, model-specific reasoning path.

Specifically, we formalize the *Insertion MDP* as a tuple $(\mathcal{Q}, \mathcal{N}, \mathcal{S}, \mathcal{A}, \mathcal{P}, \mathcal{R})$. $\mathcal{Q}$ denotes the math prompt, which serves as the input to the vanilla LLM. $\mathcal{N} = \{n_1, \ldots, n_{|\mathcal{N}|}\}$ is a finite set of reasoning steps generated from the vanilla rollout, with $|\mathcal{N}| < \infty$. $\mathcal{S} = \{(\mathcal{Q}, n_{1:t}) \mid t \in \mathbb{Z}_{\geq 0}, n_k \in \mathcal{N}, \forall k \in 1, \ldots, t\}$ defines the state space, i.e., a set of reasoning chains. Each state $s = (\mathcal{Q}, n_{1:t})$ contains the prompt and the first $t$ reasoning steps. $\mathcal{A}$ denotes the insertion action. For a given state $s$, $\mathcal{A}(s) = \{(n, x) \mid n \in \mathcal{N}, x \in 1, \ldots, |s| + 1\}$, where each action $a = (n, x)$ corresponds to

inserting a step $n$ at position $x$ in $s$, yielding a new chain $s' = \text{Ins}(s, n, x) := (\mathcal{Q}, n_{1:x-1}, n, n_{x:t})$. $\mathcal{P} : \mathcal{S} \times \mathcal{A}(s) \to \Delta(\mathcal{S})$ is the transition function of *Insertion MDP*, represent as: $\mathcal{P}(s' \mid s, (n, x)) = \mathbf{1}\{s' = \text{Ins}(s, n, x)\}$. $\mathcal{R} : \mathcal{S} \to \{0, 1\}$ is the reward function, where $\mathcal{R}(s) = 1$ if the reasoning chain $s$ produces a correct solution to $\mathcal{Q}$, and $\mathcal{R}(s) = 0$ otherwise. To quantify the marginal contribution of a reasoning step $n$ under *Insertion MDP*, we compare the reward of the chain before insertion ($\mathcal{R}(s)$) and after insertion ($\mathcal{R}(s')$). This leads to the notion of the *Sequential Marginal Contribution (SMC)*, defined below.

**Definition 1** (Sequential Marginal Contribution). Given an *Insertion MDP* $(\mathcal{Q}, \mathcal{N}, \mathcal{S}, \mathcal{A}, \mathcal{P}, \mathcal{R})$, for any current reasoning chain $s = (\mathcal{Q}, n_{1:t}) \in \mathcal{S}$ and an insertion action $a = (n, x)$ with $x \in \{1, \ldots, t+1\}$, let $s' = \text{Ins}(s, n, x)$ denote the new (re-ordered) reasoning chain after insertion. The *Sequential Marginal Contribution (SMC)* of inserting a reasoning step $n$ at position $x$ of the new chain $s'$ is defined as

$$\text{SMC}_n(s, x) = \mathcal{R}(s') - \mathcal{R}(s). \tag{1}$$

## 2.3 Fair Credit Assignment for Each (Inserted) Reasoning Step

In the Insertion MDP, a reasoning step's marginal contribution must be evaluated fairly, as it depends on both the ordering of its predecessors and its specific insertion position. To ensure fairness, we introduce the *Sequential Shapley value (SSV)*, which attributes credit to each step within a reasoning chain by accounting for order- and position-dependent effects. For each step, SSV is computed by averaging the SMC values across all possible re-ordered new chains under alternative insertion actions. The formal definition of SSV is provided below.

**Definition 2** (Sequential Shapley Value). Consider an *Insertion MDP* $(\mathcal{Q}, \mathcal{N}, \mathcal{S}, \mathcal{A}, \mathcal{P}, \mathcal{R})$, an *ordering* of $\mathcal{N}$ refers to a permutation that arranges all reasoning steps into a chain, with each step included exactly once. Let $\mathfrak{S}(\mathcal{N})$ be the set of all possible orderings. For $n \in \mathcal{N}$ and $\sigma \in \mathfrak{S}(\mathcal{N})$, let $\text{pred}_\sigma(n)$ denote the predecessor of reasoning step $n$ in ordering $\sigma$, which indicates the ordered reasoning steps that appear before step $n$ in $\sigma$ (e.g., if $\mathcal{N} = \{n_1, n_2, n_3\}$ and $\sigma = (n_2, n_1, n_3)$, then $\text{pred}_\sigma(n_1) = (n_2)$ and $\text{pred}_\sigma(n_3) = (n_2, n_1)$). Set $s_\sigma := (\mathcal{Q}, \text{pred}_\sigma(n))$. The *Sequential Shapley value (SSV)* of reasoning step $n$ is then

$$\phi_{\text{SSV}}(n) = \mathbb{E}_{\sigma \in \mathfrak{S}(\mathcal{N})} \, \mathbb{E}_{x \in \{1, \ldots, |s_\sigma|+1\}} \big[ \text{SMC}_n(s_\sigma, x) \big]. \tag{2}$$

To guarantee that SSV achieves *fair credit assignment* in LLMs reasoning, we impose the Shapley fairness properties (Wang et al., 2020) (see **Appendix B.1**) on reasoning chains, with four criteria: (i) **Sequential Efficiency**: the final reward must be fully distributed across all reasoning steps, without under- or over-estimation; (ii) **Sequential Symmetry**: if two reasoning steps make equal contributions across all chains, then those two steps must receive equal credit. (iii) **Sequential Additivity**: when two reasoning chains are merged into a new chain, the credit of the merged chain must equal the sum of the credits of the originals; and (iv) **Sequential Null Step**: any step that does not affect the final reward must receive zero credit.

**Theorem 1** (Fairness of SSV Credit Assignment). Given an *Insertion MDP* $(\mathcal{Q}, \mathcal{N}, \mathcal{S}, \mathcal{A}, \mathcal{P}, \mathcal{R})$, let $\phi_{\text{SSV}} : \mathcal{N} \to \mathbb{R}$ denote the credit assignment over all reasoning steps $n \in \mathcal{N}$, where each $\phi_{\text{SSV}}(n)$ is assigned credit to reasoning step $n$ (computed according to Eq. 2). If $\phi_{\text{SSV}}$ satisfies the following four criteria, then $\phi_{\text{SSV}}$ constitutes a *fair credit assignment*:

1.1 *Complete Credit Across Reasoning Chains (Sequential Efficiency).* The final reward is fully allocated across reasoning steps:

$$\sum_{n \in \mathcal{N}} \phi_{\text{SSV}}(n) = \mathbb{E}_{\sigma, x} \big[ \mathcal{R}(s_{\sigma, x}^{\text{full}}) - \mathcal{R}(s^\emptyset) \big].$$

1.2 *Equitable Credit for Equivalent Steps in All Chains (Sequential Symmetry).* If two reasoning steps make equal contributions across all possible chains, they must be assigned same credit:

$$\phi_{\text{SSV}}(i) = \phi_{\text{SSV}}(j), \quad \forall i, j \in \mathcal{N}.$$

1.3 *Loss-Free Credit Preservation in Chain Combination (Sequential Additivity).* Credit is preserved under the chain combination. For independent chains with disjoint step sets $\mathcal{N}_1, \mathcal{N}_2$ and rewards $\mathcal{R}_1, \mathcal{R}_2$, let $\mathcal{N} = \mathcal{N}_1 \cup \mathcal{N}_2$ and $\mathcal{R} = \mathcal{R}_1 + \mathcal{R}_2$. Then

$$\sum_{n \in \mathcal{N}} \phi_{\text{SSV}}^{\mathcal{R}}(n) = \sum_{n \in \mathcal{N}_1} \phi_{\text{SSV}}^{\mathcal{R}_1}(n) + \sum_{n \in \mathcal{N}_2} \phi_{\text{SSV}}^{\mathcal{R}_2}(n).$$

1.4 *Zero Credit to Null or Redundant Steps (Sequential Null Step).* Any step that never changes the final reward receives zero credit:

$$\phi_{\text{SSV}}(n) = 0, \quad \text{if } \text{SMC}_n(s, x) = 0 \ \ \forall(s, x).$$

The complete proofs of Theorems 1.1–1.4 appear in **Appendix D.1**.

## 2.4   FAIR ADVANTAGE ESTIMATION FOR EFFICIENT POLICY OPTIMIZATION

To leverage SSV fairness as an *Advantage* for RL training of LLMs, SSV must serve as a value baseline that reduces variance without introducing bias, thereby facilitating convergence in policy optimization, such as Proximal policy optimization (PPO)  (Schulman et al., 2017; Ouyang et al., 2022) (refer to preliminary in **Appendix B.2**). Since SSV can be a step-level fair baseline, which is invariant to the current reasoning chain and insertion position, we define the step-level advantage function as

$$A_t^{\text{SSV}} = \text{SMC}_{n_t}(s_t, x_t) - \phi_{\text{SSV}}(n_t). \tag{3}$$

where $\text{SMC}_{n_t}(s_t, x_t)$ is defined in Eq. 1 and $\phi_{\text{SSV}}(n_t)$ is defined in Eq. 2. Note that SSV, as a new value baseline for advantage estimation, is independent of the insertion position and preserves the total credit across steps. This baseline guarantees that the direction of strategy optimization does not deviate, while reducing variance and guiding policy updates in accordance with the fairness principle (Theorem 1). Therefore, the RL training objective of SSV policy optimization (SSVPO) follows SSV advantage defined in Eq. 3 as the new value baseline while preserving PPO's clipping-based stability.

$$\mathcal{J}_{\text{SSVPO}}(\theta) = \mathbb{E}\Big[ \min\Big\{ r_t(\theta) \, A_t^{\text{SSV}}, \ \text{clip}\big(r_t(\theta), 1 - \epsilon, 1 + \epsilon\big) \, A_t^{\text{SSV}} \Big\} - \beta \, \text{D}_{\text{KL}}\big(\pi_\theta(\cdot \mid s_t, \mathcal{Q}) \, \| \, \pi_{\text{ref}}(\cdot \mid s_t, \mathcal{Q})\big) \Big]. \tag{4}$$

where $a_t = (n_t, x_t)$ is the insertion action, $r_t(\theta) = \dfrac{\pi_\theta(a_t \mid s_t, \mathcal{Q})}{\pi_{\theta_{\text{old}}}(a_t \mid s_t, \mathcal{Q})}$ is importance ratio that reweights samples gathered under the behavior policy $\pi_{\theta_{\text{old}}}$ to estimate the objective under the updated policy $\pi_\theta$, $\mathcal{Q}$ denotes the prompt, $\pi_{\text{ref}}$ is a fixed reference policy, $\epsilon > 0$ is the clipping parameter, and $\beta \geq 0$ controls the KL strength.

To justify SSV as a new value baseline capable of inducing effective advantages, we provide two intuitive examples in **Appendix C**, illustrating how SSVPO (i) handles useless steps and (ii) assigns credit to trajectories whose final answers are incorrect but contain partially correct reasoning steps. And we show SSVPO is both *compatible* with insertion-action optimization and *variance-efficient*. First, the SSV baseline is independent of the sampled insertion action, so replacing an insertion-action advantage with $A_t^{\text{SSV}}$ leaves the expected policy gradient unchanged. Second, among all baselines that keep the policy gradient unbiased and that depend only on the reasoning step $n$ in a given math prompt $\mathcal{Q}$ and not on the insertion action, the SSV baseline uniquely minimizes the variance of the update. In practice, this reduces the frequency of PPO clipping and the resulting clipping bias, thereby improving policy optimization stability and sample efficiency. The following theorem formalizes that our advantage construction is unbiased and variance-minimal for policy optimization.

**Theorem 2** (Unbiased and Variance-Minimal Advantage with SSV)**.** Given an *Insertion MDP* $(\mathcal{Q}, \mathcal{N}, \mathcal{S}, \mathcal{A}, \mathcal{P}, \mathcal{R})$, let $A_t^{\text{SSV}}$ (Eq. 3) be an advantage function. Then, it is satisfied with:

2.1 *Unbiasedness (Action-Independence).* For any policy $\pi_\theta$, replacing advantage by $A_t^{\text{SSV}}$ leaves the expected policy gradient unchanged

$$\mathbb{E}\big[\nabla_\theta \log \pi_\theta(a_t \mid s_t) \, A_t^{\text{SSV}}\big] = \mathbb{E}\big[\nabla_\theta \log \pi_\theta(a_t \mid s_t) \, \text{SMC}_t\big].$$

2.2 *Variance Minimality (Optimal Baseline).* Among all value baselines $b(n)$ that do not depend on the insertion position,

$$\phi_{\text{SSV}}(n) = \arg\min_b \, \mathbb{V}\big[\nabla_\theta \log \pi_\theta(a_t \mid s_t) \, \big(\text{SMC}_t - b(n_t)\big)\big].$$

2.3 *Convergence for Policy Optimization.* Under standard policy-gradient conditions (bounded rewards (Sutton et al., 1999), square-integrable scores (Borkar & Borkar, 2008), Robbins–Monro stepsizes (Robbins & Monro, 1951)), using $A_t^{\text{SSV}}$ gives an unbiased, variance-minimal gradient (Greensmith et al., 2004; Thomas & Brunskill, 2017) estimator baselines, so stochastic gradient ascent converges to a stationary point of $J_{SSVPO}(\theta)$ in Eq. 4.

The complete proofs of Theorems 2.1–2.3 appear in **Appendix D.2**.

## 3 EXPERIMENTAL SETUP

**Base Models.** We selected two widely adopted open-source pre-trained language models as a vanilla method, which are commonly fine-tuned for mathematical reasoning tasks: *Qwen3-4B* (Yang et al., 2025) and *DeepSeek-R1-Distill-Qwen-1.5B* (Guo et al., 2025a). Qwen3-4B (with thinking mode) is the latest generation of pretrained language models under 7B parameters for best reasoning performance according to the technical report of Qwen3 (Yang et al., 2025), while DeepSeek-R1-Distill-Qwen-1.5B is even more lightweight parameters, yet still achieves competitive performance through knowledge distillation from much larger pretrained models (Guo et al., 2025a).

**Baselines.** We compare our method against state-of-the-art outcome-based RL approaches and their credit-assignment variants. Among the outcome-based RL approaches, we include RLOO (Ahmadian et al., 2024), GRPO (Shao et al., 2024), and DAPO (Yu et al., 2025). For credit assignment–based RL approaches, we include VinePPO (Kazemnejad et al., 2025) and SPO (Guo et al., 2025b), both of which were released very recently. All baselines use on-policy RL, and we further discuss how SSVPO can extend to off-policy settings in **Appendix H.1**.

**Datasets and Benchmarks.** We train and evaluate SSVPO and all baselines on seven mathematical reasoning datasets. For training, we use: (1) *GSM8K* (Cobbe et al., 2021), which contains 7,500 grade-school math problems requiring multi-step CoT reasoning and is widely used in RL for LLMs; (2) *MATH* (Hendrycks et al., 2021) contains 7,500 challenging problems spanning algebra, geometry, number theory, and other advanced topics, many of which require 10–20 reasoning steps, which often evaluate on a representative subset, *MATH-500* (Lightman et al., 2023); (3) *MultiArith* (Rae et al., 2021), with 420 arithmetic word problems; and (4) *SVAMP* (Patel et al., 2021), with 700 problems testing shallow reasoning. To assess generalization, we also evaluate on three competition-style benchmarks: *AMC'23* (Mathematical Association of America, 2023), *AIME'24* (Mathematical Association of America, 2024), and *AIME'25* (Mathematical Association of America, 2025), which feature more challenging integer-answer problems requiring long CoT and advanced reasoning. Since no training data are available for these competitions, they are used only for testing in a zero-shot setting. We also discuss extending SSVPO to zero-RL scenarios in **Appendix H.2**.

**Training Details and Source Code.** Across all experiments, we set the training batch size to match the generation batch. The learning rate is fixed at $5 \times 10^{-7}$, and we use ADAM (Kingma, 2014) as the optimizer. Full training configurations are provided in **Appendix G**. We also release open-source code with instructions and scripts in an Anonymous GitHub repository, detailed in **Appendix F**.

**Evaluation Metrics.** We evaluate SSVPO and all baselines along two dimensions: *reasoning accuracy* (performance of the finetuned base model) and *reasoning efficiency* (tokens required to reach a correct answer). We further assess performance under constrained token budgets to analyze the accuracy and efficiency trade-off. In addition, we evaluate the quality of fair credit assignment through case studies. We evaluate each experiment three times and report the mean performance. To ensure a fair and consistent comparison with baseline methods, our experiments fix the total computation budget for each reasoning benchmark by setting a maximum number of samples for all algorithms throughout the entire training process, detailed in **Appendix G.1**

## 4 RESULTS

### 4.1 PERFORMANCE IN OUTCOME-BASED RL TRAINING

In this set of experiments, we evaluate SSVPO by comparing its reasoning performance with other outcome-based RL approaches, including RLOO, GRPO, and DAPO. We focus on reasoning accuracy, which is defined as generating the correct final answer to a given math prompt, while also

considering the token usage required to produce the solution (answer). All methods are based on the same base model, *Qwen3-4B*, with the vanilla method also included as a base model. Results are summarized in Table 1.

**Accuracy.** We first examine the overall performance of SSVPO across seven mathematical reasoning benchmarks. Although each benchmark differs in question difficulty, average reasoning length, and topic coverage, SSVPO achieves state-of-the-art performance on six of them, ranking second only to RLOO on SVAMP. On challenging benchmarks, such as MATH-500, AIME 2024, and AIME 2025, SSVPO outperforms the previous SOTA algorithm by a large margin (3%–5% absolute accuracy). Averaged across all seven benchmarks, SSVPO clearly leads, improving over the vanilla method by 4.6% and over the previous SOTA RLOO by 1.8%. Further accuracy analysis of the case study results is provided in **Appendix I.1**.

**Efficiency.** We next evaluate reasoning efficiency, measured by the number of tokens required to generate a correct answer. As shown in Table 1, each benchmark reports the average token usage per question. Among the seven benchmarks, SSVPO achieves the best efficiency on five, with only two (MultiArith and SVAMP) slightly behind RLOO. The most significant gain is observed on GSM8K, where SSVPO reduces token usage to 1,320 per reasoning chain, compared to 2,215 for the base model and 1,665 for RLOO. On average, SSVPO delivers the best efficiency, reducing token usage by 18.1% relative to the baseline and outperforming all existing methods aimed at improving the reasoning potential of pretrained LLMs. Further efficiency analysis of the case study results is provided in **Appendix I.2**.

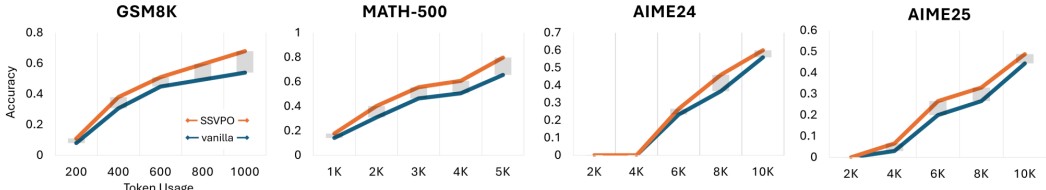

Figure 3: Reasoning accuracy comparison between SSVPO and Qwen3-4B (vanilla) under varying reasoning length constraints (measured by token usage) on four benchmarks: GSM8K, MATH-500, AIME 2024, and AIME 2025. SSVPO consistently achieves higher accuracy across all length constraints. Higher curves indicate better performance at a given token budget.

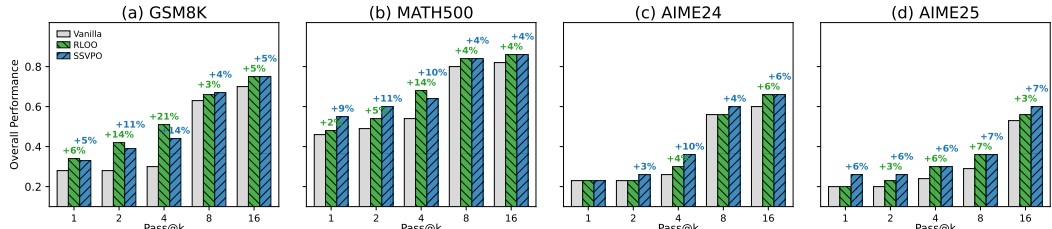

Figure 4: Reasoning accuracy with $K$ queries (success if at least one answer is correct), reported as Pass@$K$. Bars show Pass@1 through Pass@16 under the same token cap. Higher bars indicate the performance gains achieved by SSVPO.

**Reasoning with constrained token budget.** To further examine reasoning performance under limited token resources, we constrain the token budget: the fine-tuning methods must produce a correct

Table 1: Performance comparison between SVPPO and outcome-based RL approaches for mathematical reasoning across seven benchmarks. "Acc" denotes accuracy, and "Tokens" denotes token count (token budget fixed at 16144 tokens per sample). The best performances are shown in **bold**, while suboptimal results are marked with underlines. In the last column, red indicates the average improvement over the base model (vanilla) across all benchmarks, whereas blue indicates a decrease.

| Method | MultiArith | | SVAMP | | GSM8K | | MATH-500 | | AMC 2023 | | AIME 2024 | | AIME 2025 | | Overall | |
|---|---|---|---|---|---|---|---|---|---|---|---|---|---|---|---|---|
| | Acc | Tokens | Acc | Tokens | Acc | Tokens | Acc | Tokens | Acc | Tokens | Acc | Tokens | Acc | Tokens | Acc | Tokens |
| Vanilla | 98.8 | 1032 | 93.6 | 1632 | 93.6 | 2215 | 91.2 | 5167 | 87.5 | 7687 | 60.0 | 12205 | 46.6 | 15385 | 82.1$_{+0.0}$ | 6327$_{-0\%}$ |
| RLOO | 100.0 | **984** | **94.8** | **1529** | 94.6 | 1665 | 92.2 | 4966 | 94.1 | 6875 | 63.3 | 11373 | 55.5 | 12159 | 84.9$_{+2.8}$ | 5650$_{-10.7\%}$ |
| GRPO | 99.8 | 1012 | 94.2 | 1613 | 94.6 | 2117 | 92.4 | 5088 | 91.6 | 6634 | 61.1 | 12125 | 50.0 | 12668 | 83.4$_{+1.3}$ | 5894$_{-6.84\%}$ |
| DAPO | 99.6 | 1032 | 93.8 | 1642 | 94.8 | 2312 | 91.8 | 6742 | 91.6 | 9404 | 62.2 | 14445 | 47.7 | 12920 | 83.0$_{+0.9}$ | 6828$_{+7.91\%}$ |
| SSVPO | 100.0 | 996 | 94.6 | 1585 | **95.2** | **1320** | **95.0** | 4185 | **95.0** | 6436 | **66.6** | 10168 | 61.1 | 11559 | **86.7$_{+4.6}$** | **5178$_{-18.1\%}$** |

Table 2: Performance comparison between SVPPO and other credit assignment algorithms on the DeepSeek backbone for mathematical reasoning.

| Method | MultiArith | | SVAMP | | GSM8K | | MATH-500 | | AMC 2023 | | AIME 2024 | | AIME 2025 | | Overall | |
|---|---|---|---|---|---|---|---|---|---|---|---|---|---|---|---|---|
| | Acc | Tokens | Acc | Tokens | Acc | Tokens | Acc | Tokens | Acc | Tokens | Acc | Tokens | Acc | Tokens | Acc | Tokens |
| Vanilla | 92.1 | 324 | 74.0 | 439 | 76.1 | 1617 | 69.0 | 6018 | 52.2 | 8819 | 23.3 | 13702 | 13.3 | 14450 | 57.1$_{+0.0}$ | 6481$_{-0\%}$ |
| VinePPO | 93.8 | 307 | 74.6 | 439 | 85.4 | 1462 | 81.8 | 4640 | 67.5 | 8413 | 23.3 | **12518** | 20.0 | 12477 | 63.7$_{+6.6}$ | 5750$_{-11.2\%}$ |
| SPO | 94.3 | 311 | 75.0 | **399** | 88.9 | 1337 | 82.4 | 4530 | 77.5 | **7137** | 26.6 | 12679 | 20.0 | **11420** | 66.3$_{+9.2}$ | **5402**$_{-16.6\%}$ |
| SSVPO | **95.5** | **298** | **76.6** | 419 | **90.2** | **1331** | **86.4** | 4343 | **79.6** | 8059 | **28.8** | 13377 | **24.4** | 13035 | **68.7**$_{+11.6}$ | 5838$_{-9.9\%}$ |

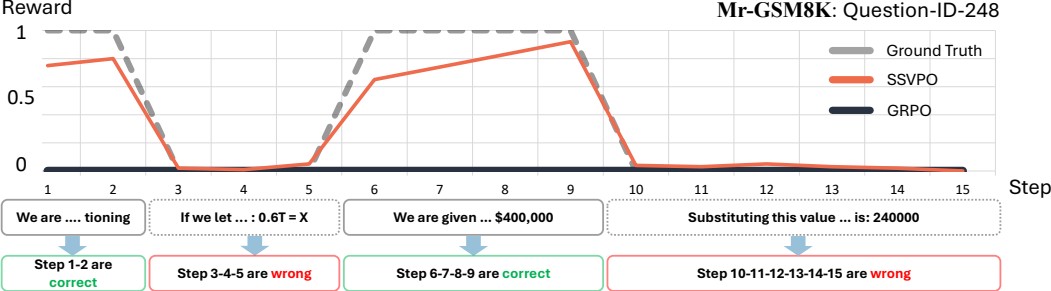

Figure 5: Credit assignment results on a reasoning sequence that fails to solve the math problem. Among 15 reasoning steps, steps 1,2,6-9 are correct, while steps 3-5,10-15 are wrong. Compared to GRPO, SSVPO provides estimates that are closer to the ground truth on all reasoning steps.

answer using fewer tokens than a specified cap. The results are shown in Figure 3. We observe that SSVPO achieves steady accuracy gains across all token budgets in all tested benchmarks including GSM8K, MATH-500, AIME24, and AIME25. Even under severe constraints (40%–60% of the tokens typically required), SSVPO still outperforms its base model (vanilla) by a significant margin (10%-30%). This result is consistent with SSVPO's reasoning path, which emphasizes key reasoning steps to reach the answer efficiently when tokens are limited.

**Reasoning with multiple trials.** Another statistical analysis we conduct to more comprehensively evaluate reasoning capability is Pass@K, which measures the probability that a model generates at least one correct answer over multiple trials. We evaluate this metric on GSM8K, MATH-500, AIME 2024, and AIME 2025, with results shown as histograms in Figure 4. As $K$ increases from 1 to 16, reasoning accuracy improves for both the base model and SSVPO. Importantly, SSVPO shows consistent gains over the base model (vanilla), with particularly strong improvements on GSM8K and MATH-500, where it achieves over 10% average accuracy increase compared to Qwen3-4B. This experiment demonstrates that while SSVPO emphasizes key reasoning steps for accuracy and efficiency, it also retains the capacity to organize more diverse reasoning chains, suggesting substantial room for further improvement when given additional time and resources.

## 4.2 EFFECTIVENESS IN CREDIT ASSIGNMENT

In the second part of the experiment, we replace the base model *Qwen3-4B* with *DeepSeek-R1-Distill-Qwen-1.5B*, a lighter-weight backbone specialized for reasoning tasks with a stronger focus on credit assignment, aligning with existing work (Dang & Ngo, 2025; Chen et al., 2025; Liu et al., 2025b; Ren et al., 2025). Based on the new backbone model, we aim to (1) discover the accuracy–efficiency trade-off in the reasoning process by comparing SSVPO with other credit-assignment variants, including VinePPO and SPO, and (2) visualize how credit assignment operates the fairness during reasoning and how SSVPO discovers better strategies compared to the baseline.

**Accuracy and efficiency trade-off.** Results in Table 2 show that existing credit assignment methods achieve lower token usage than SSVPO (e.g., SPO on AMC 2023 and AIME 2025, VinePPO on AIME 2024). However, these efficiency gains often come at the cost of reduced accuracy performance, as they skip some reasoning steps that are crucial for reaching the correct answer. For instance, on AMC 2023, SPO uses fewer tokens than SSVPO, but accuracy drops from 79.6 to 77.5. A similar pattern was observed on AIME 2025, where SPO achieves 20.0 accuracy with 12,420 tokens, while SSVPO achieves 24.4 with only slightly more tokens (13,035). Another observation is on AIME 2024, SPO reaches 19.3 with 12,679 tokens, whereas SSVPO achieves 28.8 with 13,377

tokens, showing that a small increase in tokens yields a substantial performance gain. Overall, these results suggest that efficiency gains often come at the expense of accuracy, particularly on complex reasoning tasks. SSVPO, however, balances this trade-off more adaptively: on easier tasks, it achieves both higher accuracy and lower token usage, while on more challenging tasks, it maintains accuracy with only minimal additional token usage. Further accuracy and efficiency trade-off analysis of the case study results is provided in **Appendix I.3**.

**Fairness through step-wise alignment to ground-truth reward.** SSVPO incentivizes fair credit assignment in a long reasoning chain. Here, we present a case study to illustrate the empirical fairness property, complementing our theoretical analysis (Sec2.3). As shown in Figure 5, we sample question ID 248 from Mr-GSM8K (Zeng et al., 2023). We let SSVPO and GRPO generate reasoning chains and estimate the contribution of each reasoning step. Unluckily, both SSVPO and GRPO arrive at incorrect final answers. However, their treatment of this failure differs. GRPO treats all steps in the reasoning chain as equally useless, assigning each a reward of $0$. By contrast, SSVPO distinguishes among steps: even though they do not ultimately lead to the correct answer, some steps are credited for making meaningful progress in the reasoning chain. Such steps may not solve this particular problem, but could be crucial in solving others.

To validate this, we overlay a human-labeled ground-truth reward curve (e.g., process reward models) on the reasoning chain produced by SSVPO. Unsurprisingly, SSVPO's credit assignment closely aligns with this ground truth: steps that are logically correct still receive positive credit, regardless of the final outcome, especially in cases where the final answer is wrong. A similar effect occurs when the final answer is correct but some intermediate steps are flawed: SSVPO identifies these and assigns them lower credit. Thus, SSVPO enhances LLMs reasoning by ensuring that each step is rewarded fairly by assigning more credit to important steps and penalizing misleading ones. This strategy ultimately leads to higher accuracy and greater efficiency in mathematical reasoning and beyond.

### 4.3 ABLATION STUDY

To evaluate SSVPO's generalization across model scales and task difficulties, we also train Qwen3-1.7B (Yang et al., 2025), with full results in **Appendix E**. We further study the effect of the SSVPO-specific hyperparameter `reorder_num` using 2-, 3-, and 4-step reordering configurations, reporting accuracy (Acc) and response length (Len) on GSM8K, MATH-500, AMC23, AIME24, and AIME25. As shown in Table 3, increasing the number of reordered steps consistently improves accuracy—particularly on harder benchmarks—but also increases generation cost. Gains range from 1.3 points on GSM8K to 5.0 on AMC23 and 3.3 on AIME24/25 (7–8% relative). Thus, deeper reordering yields modest benefits on everyday reasoning but substantial improvements on competition-level tasks. Additional training dynamics appear in Table 5.

Table 3: Performance comparison of different update steps on mathematical reasoning benchmarks.

| Method | GSM8K | | MATH-500 | | AMC 2023 | | AIME 2024 | | AIME 2025 | |
|--------|-------|-------|----------|------|----------|------|-----------|-------|-----------|-------|
| | Acc | Len | Acc | Len | Acc | Lens | Acc | Len | Acc | Len |
| 2 Steps | 90.5 | 1492 | 79.4 | 3909 | 77.5 | 7084 | 43.3 | 11088 | 40.0 | 12986 |
| 3 Steps | 91.3 | 1617 | 80.0 | 3996 | 80.0 | 8109 | 43.3 | 13057 | 43.3 | 13424 |
| 4 Steps | 91.8 | 1692 | 82.0 | 4166 | 82.5 | 8766 | 46.6 | 13603 | 43.3 | 13742 |

## 5 CONCLUDING REMARKS

This work introduced SSVPO, a sequential Shapley value–based credit assignment framework for enhancing post-training LLMs reasoning. We provided a theoretical proof showing that SSVPO ensures fair step-level credit allocation within reasoning chains and establishes a variance-efficient value baseline for policy optimization, enabling shorter and more effective reasoning processes. Empirical evaluations across diverse math reasoning benchmarks demonstrate that SSVPO consistently outperforms existing RL training methods in terms of accuracy, token efficiency, and reasoning quality. Our findings highlight the importance of fair credit assignment as a guiding principle for future RL approaches in LLMs reasoning.

## REPRODUCIBILITY STATEMENT

We have taken extensive measures to ensure reproducibility. The detailed related work on RL for LLMs post-training and MARL is summarized in **Appendix A**, and the necessary preliminaries for SSVPO (including Shapley value axioms and advantage functions) are introduced in **Appendix B**. Intuitive examples illustrating SSVPO credit assignment are provided in **Appendix C**, while theoretical foundations, including formal claims and complete proofs of Theorems 1 and 2, are presented in **Appendix D**. Additional experimental results on Qwen3-1.7B and training dynamics analysis are reported in **Appendix E**. Implementation details such as the full algorithm, pseudocode, and anonymous source code are provided in **Appendix F**, with the repository including all scripts for training and evaluation to be made publicly available upon publication. **Appendix G** reports full experimental settings, including random seeds, hyperparameters, computational infrastructure (hardware, operating system, and software libraries), and the partition strategy for reasoning chains. Discussions on SSVPO applicability to zero-RL and off-policy settings are provided in **Appendix H**. Finally, we empirically validate all theoretical claims by reporting reasoning accuracy and efficiency, with extended result analysis and additional case studies given in **Appendix I**.

## ETHICS STATEMENT

We propose SSVPO, a method that introduces a fair credit assignment mechanism to enhance the efficiency of LLMs reasoning. All experiments are conducted on open-source LLMs and publicly available mathematical reasoning datasets, which contain no personally identifiable or sensitive information. No new data involving human subjects were collected, and thus, no privacy concerns arise. We therefore do not anticipate any negative ethical impacts from this work.

## ACKNOWLEDGEMENT

This study is supported by Australian Research Council (ARC) Projects DE220100265, DP250103612, and LP240200636.

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

## DECLARATION OF LLMS USAGE

The development of SSVPO did not rely on LLMs as essential, original, or non-standard components of the research. GPT-5 was used solely to assist with proofreading tasks, including polishing text, correcting grammar and spelling, and ensuring consistency in American English usage.

## A  RELATED WORKS

**Outcome-Based RL Training of LLMs**   Post-training for LLMs is shifting from supervised fine-tuning toward RL with verifiable rewards. Among these approaches, outcome-based RL has become central to enhancing LLMs reasoning capabilities. By rewarding only the final outcome (e.g., the correct answer to a math prompt), it encourages models to implicitly connect intermediate reasoning steps, thereby mimicking the human CoT process. Representative methods include Reward Leave-One-Out (RLOO) (Ahmadian et al., 2024), which introduces leave-one-out baselines for improved trajectory-level variance reduction, and Group Relative Policy Optimization (GRPO) (Shao et al., 2024), which reduces variance and instability by sampling multiple trajectories relative to token-level PPO baselines. More recently, Decoupled Clip and Dynamic Sampling Policy Optimization (DAPO) (Yu et al., 2025) extends GRPO with overlong reward shaping, improving stability without relying on KL regularization. While these outcome-based RL methods improve reasoning by leveraging final rewards, they still face optimization challenges. Most notably, the difficulty of determining each step's true contribution within a reasoning chain. Accurate estimation of step-level contributions is crucial for maintaining efficiency, as it allows redundant or even detrimental steps to be pruned, while preserving those that meaningfully contribute to the final answer.

**Credit Assignment in RL Training of LLMs**   To improve LLMs' reasoning efficiency, recent works have introduced credit assignment strategies into outcome-based RL (Kazemnejad et al., 2025; Guo et al., 2025b; Tan & Pan, 2025). These methods explicitly model the contribution of each reasoning step by decomposing the final reward signal of the entire CoT into step-level credits. Specifically, VinePPO (Kazemnejad et al., 2025) defines intermediate steps and generates multiple continuations (vine rollouts), using Monte Carlo returns to estimate per-step advantages. Segment Policy Optimization (SPO) (Guo et al., 2025b) introduces a probabilistic masking framework to derive advantage estimators based on value differences between adjacent reasoning steps. More recently, Group Token Policy Optimization (GTPO) (Tan & Pan, 2025) applies dynamic entropy weighting, reweighting returns within correct trajectories to reduce variance and stabilize training.

Although these credit assignment methods outperform purely outcome-based RL approaches, they still fall short of fairly and accurately estimating each step's true contribution. For example, VinePPO aggregates Monte Carlo returns, which tends to overweight early steps in a reasoning chain while under-crediting steps closer to the terminal state, thereby misrepresenting their marginal value. SPO provides more reliable credit when reasoning chains are fully correct, but fails in partially correct cases, when an early step is logically valid but later steps lead to an incorrect final answer, its contribution is ignored. GTPO, by contrast, uses entropy as a heuristic, which can over-credit lexically diverse but unimportant steps while under-crediting crucial low-entropy ones. Such misalignments underscore the need for a step-level credit assignment method with theoretical guarantees of fairness and provable stable variance in policy optimization.

We also note that recent studies, such as Serial-GRPO (Dai et al., 2025) and Unary Feedback as Observation (UFO) (Liu et al., 2025a), extend the single-turn paradigm to a multi-turn setting in order to capture intermediate reasoning steps. However, they still rely on additional policies or feedback to improve performance, indicating they cannot be regarded as pure RL methods with credit assignment driven solely by outcome-based signals. More recently, LLM agent work has explored step-level, fine-grained credit assignment, including Agentic Reinforced Policy Optimization (ARPO) (Dong et al., 2025) and Group-in-Group Policy Optimization (GiGPO) (Feng et al., 2025). These methods refine trajectory-level returns into step-level credit in agent-based environments, thereby improving the scheduling of multi-turn LLM tool use. Specifically, they leverage explicit environment states and interaction signals (e.g., retrieval success, subgoal completion) and derive step-level credit through anchored states, within-group comparison, or group-in-group aggregation. This enables higher success rates and better sample efficiency in retrieval and web-operation tasks. However, such approaches assume that each step has an observable external success criterion

(e.g., whether a tool invocation succeeds), which fundamentally differs from mathematical reasoning tasks. In reasoning, intermediate steps rarely provide verifiable immediate feedback, making it difficult to deliver consistent and accurate supervision at the step level. In contrast, our focus is on a single-problem, single-turn setting, where we derive step-level credit assignment purely from outcome-based signals.

**Original Credit Assignment from MARL to LLMs** The concept of credit assignment originates from multi-agent reinforcement learning (MARL) (Foerster et al., 2018; Rashid et al., 2020; Li et al., 2025b), where the goal is to evaluate each agent's actual contribution to the team's performance. For example, the difference reward method (Castellini et al., 2025) defines an agent-specific utility function by contrasting the team return with and without that agent, directly measuring its incremental contribution. The counterfactual advantage method (Foerster et al., 2018) employs a centralized critic that holds other agents fixed and marginalizes over the focal agent's action to construct a counterfactual baseline, attributing only the advantage beyond this baseline to that agent. More recently, the Shapley-Q approach (Wang et al., 2022a) distributes the team reward according to each agent's expected marginal contribution across coalitions and orderings, aligning well with fairness principles central to our work.

However, directly adapting this notion from MARL to LLMs reasoning is non-trivial. Original Shapley values do not capture the inherently temporal nature of reasoning processes such as CoT, where steps unfold sequentially and exhibit strong positional dependencies. Inspired by insertion-based text generation Stern et al. (2019) and reasoning path search methods (Wang et al., 2022b; Yao et al., 2023), we assume each reasoning step can be an "agent" and insert those steps into the position to form a reasoning chain, which can be an "action." This formulation allows the design of a new Shapley value tailored for LLMs training, while preserving the axiomatic fairness properties of the original Shapley value in MARL, when computing the expected marginal contribution of each reasoning step in CoTs.

## B PRELIMINARY

### B.1 SHAPLEY VALUE AND ITS AXIOMS

We recall the *Shapley value* from cooperative game theory (Shapley et al., 1953), widely used as a fairness reference for credit assignment in MARL (Wang et al., 2022a). At a high level, the Shapley value measures how much *extra value* each participant contributes on average across all cooperative contexts. Imagine forming a coalition by letting players join one at a time in a uniformly random order; whenever a player joins, we record the *incremental payoff* (marginal contribution) created by their participation. A player's Shapley value is the average of this marginal contribution over *all* possible arrival orders. Its appeal as a fairness rule stems from the fact that it uniquely satisfies four natural axioms: efficiency, symmetry, additivity, and the dummy (null player) property. Formally, let $\mathcal{N}$ be a finite player set and let $v : 2^{\mathcal{N}} \to \mathbb{R}$ be a (coalitional) set function that assigns a value to every coalition $S \subseteq \mathcal{N}$. For $i \in \mathcal{N}$, the Shapley value is defined by the original subset–sum formula

$$\phi_i(v) = \sum_{S \subseteq \mathcal{N} \setminus \{i\}} \frac{|S|! \left(|\mathcal{N}| - |S| - 1\right)!}{|\mathcal{N}|!} \Big( v(S \cup \{i\}) - v(S) \Big), \tag{5}$$

where the factorial weight depends only on $|S|$ and $|\mathcal{N}|$.

**Shapley Value Axioms.** A value assignment $\Phi(v) = (\phi_i(v))_{i \in \mathcal{N}}$ is called *Shapley* if it satisfies the following axioms:

**Axiom 1** (Efficiency). The split neither creates nor destroys value: the total credit equals the coalition's total value (relative to the empty set).

$$\sum_{i \in \mathcal{N}} \phi_i(v) = v(\mathcal{N}) - v(\varnothing).$$

In particular, if $v(\varnothing) = 0$, then $\sum_{i \in \mathcal{N}} \phi_i(v) = v(\mathcal{N})$.

**Axiom 2** (Symmetry). Players who are indistinguishable in their impact on every coalition receive the same credit. If $i, j \in \mathcal{N}$ satisfy $v(S \cup \{i\}) = v(S \cup \{j\})$ for all $S \subseteq \mathcal{N} \setminus \{i, j\}$, then $\phi_i(v) = \phi_j(v)$.

**Axiom 3** (Additivity). Credits add across games: for any $v, w : 2^{\mathcal{N}} \to \mathbb{R}$ and every $i \in \mathcal{N}$, one has $\phi_i(v + w) = \phi_i(v) + \phi_i(w)$.

**Axiom 4** (Dummy (Null Player)). A player who never changes any coalition's value receives zero credit. If $v(S \cup \{i\}) = v(S)$ for all $S \subseteq \mathcal{N} \setminus \{i\}$, then $\phi_i(v) = 0$.

**Fairness of the Shapley value.** By the classical characterization of Shapley et al. (1953), the Shapley value in Eq. 5 is the *unique* credit assignment that satisfies the four axioms stated above: Efficiency (Axiom 1), Symmetry (Axiom 2), Additivity (Axiom 3), and Dummy (Null Player) (Axiom 4). Consequently, the Shapley value constitutes a principled and *fair* rule for credit allocation, and we adopt it as our fairness reference throughout.

### B.2 Advantage Function in RL Training of LLMs

We place the *advantage function* at the center of outcome-based RL fine-tuning for LLMs. Beyond serving as the signal that tilts probability mass toward better actions, a well-chosen baseline yields lower-variance, more stable updates. We first discover review policy gradients and the invariance to action-independent baselines, then introduce Monte Carlo (MC) and generalized advantage estimation (GAE). We finally review PPO's clipped surrogate (with optional KL regularization) and assume SSV enters as a *value baseline*. Throughout, notation matches the *Insertion MDP*: $s_t$ denotes the before insertion chain and $a_t$ the insertion action (e.g., $a_t = (n_t, x_t)$).

**Why advantage matters.** Intuitively, the advantage function $A_t$ measures how much better (or worse) the sampled action is than the state's average. In long-horizon reasoning with final-reward LLMs settings, deriving the advantage value from a strong baseline can drastically reduce gradient variance, thereby stabilizing training and improving sample efficiency.

**Policy gradient, advantage, and baseline invariance.** For a stochastic policy $\pi_\theta$, the policy-gradient theorem gives

$$\nabla_\theta J(\theta) \;=\; \mathbb{E}[\nabla_\theta \log \pi_\theta(a_t \,|\, s_t) \, A_t^\pi], \tag{6}$$

where the step-level advantage compares action- and state-values,

$$A_t^\pi \;=\; Q^\pi(s_t, a_t) \;-\; V^\pi(s_t). \tag{7}$$

Subtracting *any* baseline $b(s_t)$ that *does not depend* on the sampled action $a_t$ preserves the expected policy gradient and typically lowers variance:

$$\mathbb{E}\big[\nabla_\theta \log \pi_\theta(a_t \,|\, s_t) \, \big(A_t^\pi - b(s_t)\big)\big] = \mathbb{E}[\nabla_\theta \log \pi_\theta(a_t \,|\, s_t) \, A_t^\pi]. \tag{8}$$

Practically, $b(\cdot)$ is learned as a value function $V_\psi(\cdot)$ or replaced by a task-specific baseline (e.g., SSV in our method). The identity in Eq. 8 follows from $\mathbb{E}_{a \sim \pi(\cdot|s)}[\nabla_\theta \log \pi_\theta(a \mid s)] = 0$.

**Monte Carlo and GAE estimators.** Let $R_t$ be the scalar reward at time $t$, $\gamma \in (0, 1]$ a discount, and define the (bootstrapped) return

$$G_t \;=\; \sum_{k=t}^{T-1} \gamma^{k-t} \, R_k \;+\; \gamma^{T-t} \, V_\psi(s_T) \;\; \text{(if not terminal at } T\text{)}. \tag{9}$$

A basic estimator is $\widehat{A}_t = G_t - V_\psi(s_t)$, which is low bias, high variance for long-horizon reasoning. The generalized advantage estimator (GAE) with $\lambda \in [0, 1]$ trades bias and variance via temporal-difference (TD) residuals

$$\delta_t \;=\; R_t + \gamma \, V_\psi(s_{t+1}) - V_\psi(s_t), \tag{10}$$

$$\widehat{A}_t^{\mathrm{GAE}(\gamma, \lambda)} \;=\; \sum_{\ell=t}^{T-1} (\gamma \lambda)^{\ell-t} \delta_\ell \;=\; \delta_t + \gamma \lambda \, \widehat{A}_{t+1}^{\mathrm{GAE}}. \tag{11}$$

In the policy loss, $\widehat{A}_t$ (or $\widehat{A}_t^{\mathrm{GAE}}$) is used with `stop-gradient`. For numerical stability, it is standard to normalize advantages per batch to zero mean and unit variance.

**Practical bias–variance guidance.** Monte Carlo (MC) is unbiased but may have very high variance on long sequences; GAE with $\gamma \approx 1$ and $\lambda \in [0.9, 0.97]$ is a robust default for LLMs training. Ensure reward and value scales match (value-loss weighting or reward scaling can help), and consider grouping normalization by prompt to avoid cross-task scale interference.

**PPO clipped surrogate and clipping bias.** Let the importance ratio be

$$r_t(\theta) \;=\; \frac{\pi_\theta(a_t \mid s_t)}{\pi_{\theta_{\text{old}}}(a_t \mid s_t)}, \tag{12}$$

and let $\epsilon > 0$ be the clipping parameter. The PPO surrogate maximizes

$$J_{\text{PPO}}(\theta) \;=\; \mathbb{E}\Big[\min\Big\{ r_t(\theta)\,\widehat{A}_t, \; \text{clip}\big(r_t(\theta),\, 1-\epsilon,\, 1+\epsilon\big)\,\widehat{A}_t\Big\}\Big]. \tag{13}$$

Clipping prevents excessively large updates and implicitly enforces a trust region, but introduces *clipping bias*: positive advantages are truncated more often than negative ones, nudging updates toward conservatism. Reducing the variance of $\widehat{A}_t$ (e.g., through a stronger baseline such as SSV) lowers clipping frequency and the induced bias, improving the sample efficiency.

**KL-regularized variant and early stopping.** A KL-penalized objective augments Eq. 13 with

$$- \beta\,\mathrm{D}_{\text{KL}}\big(\pi_\theta(\cdot \mid s_t) \,\big\|\, \pi_{\theta_{\text{old}}}(\cdot \mid s_t)\big), \tag{14}$$

or monitors $\mathrm{D}_{\text{KL}}$ for early stopping when it exceeds a threshold. An adaptive $\beta$ that targets a desired per-update KL (e.g., per-token) is often effective, especially for text policies where distributional drift must be tightly controlled.

**Plugging in SSV as a value baseline.** In Insertion MDP, a reasoning step's contribution depends on order and position. The SSV baseline (defined earlier) removes position-dependent noise while remaining action-independent for position-only updates. Concretely, we substitute the baseline in the advantage with

$$A_t^{\text{SSV}} \;=\; \text{SMC}_{n_t}(s_{t-1}, x_t) \;-\; \phi_{\text{SSV}}(n_t),$$

which preserves unbiased policy gradients and uniquely minimizes update variance among baselines that depend only on $(\mathcal{Q}, n)$, under standard conditions discussed earlier. In practice, $A_t^{\text{SSV}}$ reduces clipping frequency and clipping bias in Eq. 13, improving optimization stability and sample efficiency.

**Implementation notes for Insertion MDP.** If the policy factorizes as $\pi_\theta(a \mid s) = \pi_\theta(n \mid s)\,\pi_\theta(x \mid n, s)$ and only the position component is updated while $\pi_\theta(n \mid s)$ is held fixed, then $\phi_{\text{SSV}}(n)$ is action-independent and safe to use as a baseline. If both components are updated jointly, either (i) use separate baselines that are independent of each component's action, or (ii) stage the updates so that the action-independence condition holds for the component being updated.

## C INTUITIVE EXAMPLES OF SSVPO CREDIT ASSIGNMENT

### C.1 EXAMPLE 1: DISTINGUISHING USELESS STEPS

Consider four reasoning chains: $A \to$ correct, $B \to$ correct, $A \to B \to$ correct, $B \to A \to$ correct.

Let the reward be 1 for correct answers. When optimising the trajectory $A \to B \to$ correct, VinePPO obtains

$$\hat{V}^{\text{MC}}(A) = 1, \qquad\qquad \hat{V}^{\text{MC}}(B) = 1.$$

The advantages for $A$ and $B$ are:

$$A^{\text{MC}}(A, B) = r_A + \gamma \hat{V}^{\text{MC}}(B) - \hat{V}^{\text{MC}}(A) \leq 0,$$

$$\hat{A}^{\text{MC}}(B, \text{terminal state}) = r_{\text{final}} + \gamma \hat{V}^{\text{MC}}(\text{terminal state}) - \hat{V}^{\text{MC}}(B) = 0.$$

This means VinePPO *cannot accurately detect useless steps*: on the trajectory $A \to B \to$ correct, both the essential step $A$ (which already makes the answer correct) and the redundant step $B$ (which

does not change the outcome) receive the same zero advantage. The VinePPO estimator advantage does not distinguish between redundant steps and necessary steps, so useless step will have zero-advantage does not imply that VinePPO can reliably identify useless steps in practice.

In contrast, SSVPO yields:

$$\text{SMC}_A(\varnothing, 0) = 1, \qquad\qquad \text{SMC}_B(A, 1) = 0,$$

and

$$\text{SSV}(A) = \tfrac{1}{3}\big(\text{SMC}_A(\varnothing, 0) + \text{SMC}_A(B, 1) + \text{SMC}_A(B, 0)\big) = \tfrac{2}{3},$$
$$\text{SSV}(B) = \tfrac{1}{3}\big(\text{SMC}_B(\varnothing, 0) + \text{SMC}_B(A, 1) + \text{SMC}_B(A, 0)\big) = \tfrac{2}{3}.$$

The advantages become:

$$A_A^{\text{SSV}} = \text{SMC}_A(\varnothing, 0) - \phi_{\text{SSV}}(A) = 1 - \tfrac{2}{3} = \tfrac{1}{3},$$
$$A_B^{\text{SSV}} = \text{SMC}_B(A, 1) - \phi_{\text{SSV}}(B) = 0 - \tfrac{2}{3} = -\tfrac{2}{3}.$$

Thus, SSVPO assigns a negative advantage to the redundant step $B$ in the trajectory that needs optimisation, while giving a positive advantage to the useful step $A$. This illustrates why SSVPO provides more accurate step-level credit than VinePPO.

### C.2 EXAMPLE 2: POSITIVE ASSIGNMENT FOR USEFUL STEP IN AN INCORRECT TRAJECTORY

We give an intuitive example showing why SSVPO (which uses SMC) offers more accurate step-level credit assignment than VinePPO. Assume four possible reasoning paths for the same math problem: $A \to$ correct, $B \to$ correct, $A \to B \to$ correct, $B \to A \to$ incorrect.

Let the reward be 1 for correct answers. Consider optimizing the trajectory $B \to A \to$ incorrect. VinePPO value estimates:

$$\hat{V}^{\text{MC}}(A) = \tfrac{2}{3}, \qquad\qquad \hat{V}^{\text{MC}}(B) = \tfrac{2}{3}.$$

VinePPO advantages:

$$A^{\text{MC}}(B, A) = \gamma \hat{V}^{\text{MC}}(A) - \hat{V}^{\text{MC}}(B) \leq 0,$$
$$\hat{A}^{\text{MC}}(A, \text{terminal state}) = \gamma \hat{V}^{\text{MC}}(\text{terminal state}) - \hat{V}^{\text{MC}}(A) = -\tfrac{2}{3}.$$

Thus, VinePPO fails to assign *positive* credit to step $B$, even though $B$ contributes to correct reasoning in other trajectories.

SSVPO (SMC-based) value estimates:

$$\text{SMC}_A(B, 1) = -1, \qquad\qquad \text{SMC}_B(\varnothing, 0) = 1,$$

and the corresponding Sequential Shapley values are

$$\text{SSV}(A) = \tfrac{1}{3}\big(\text{SMC}_A(\varnothing, 0) + \text{SMC}_A(B, 1) + \text{SMC}_A(B, 0)\big) = \tfrac{1}{3},$$
$$\text{SSV}(B) = \tfrac{1}{3}\big(\text{SMC}_B(\varnothing, 0) + \text{SMC}_B(A, 1) + \text{SMC}_B(A, 0)\big) = \tfrac{1}{3}.$$

SSVPO (SMC-based) advantages become

$$A_B^{\text{SSV}} = \text{SMC}_B(\varnothing, 0) - \phi_{\text{SSV}}(B) = 1 - \tfrac{1}{3} = \tfrac{2}{3},$$
$$A_A^{\text{SSV}} = \text{SMC}_A(B, 1) - \phi_{\text{SSV}}(A) = -1 - \tfrac{1}{3} = -\tfrac{4}{3}.$$

This shows that even if the final answer is wrong, SSVPO correctly identifies $B$ as a useful step and assigns a strong penalty to the erroneous step $A$.

## D THEORETICAL ANALYSIS

### D.1 PROOF OF THEOREM 1

For an insertion MDP $(\mathcal{Q}, \mathcal{N}, \mathcal{S}, \mathcal{A}, \mathcal{P}, \mathcal{R})$, a chain $s = (\mathcal{Q}, n_{1:t})$, an action $a = (n, x)$ with $x \in \{1, \ldots, t+1\}$, and $s' = \text{Ins}(s, n, x)$, the sequential marginal contribution (SMC) is

$$\text{SMC}_n(s, x) := \mathcal{R}\big(\text{Ins}(s, n, x)\big) - \mathcal{R}(s).$$

Let $\mathfrak{S}(\mathcal{N})$ be all orderings of $\mathcal{N}$ and, for $\sigma \in \mathfrak{S}(\mathcal{N})$, let $\mathrm{pred}_\sigma(n)$ be the ordered list of steps before $n$ under $\sigma$; set $s_\sigma := (\mathcal{Q}, \mathrm{pred}_\sigma(n))$. The SSV of step $n$ is

$$\phi_{\mathrm{SSV}}(n) \ = \ \mathbb{E}_{\sigma \in \mathfrak{S}(\mathcal{N})} \, \mathbb{E}_{x \in \{1, \ldots, |s_\sigma|+1\}} \big[ \mathrm{SMC}_n(s_\sigma, x) \big]. \tag{15}$$

Eq. 15 can be written as the following explicit finite sums.

$$\phi_{\mathrm{SSV}}(n) \ = \ \frac{1}{|\mathfrak{S}(\mathcal{N})|} \sum_{\sigma \in \mathfrak{S}(\mathcal{N})} \frac{1}{|s_\sigma|+1} \sum_{x=1}^{|s_\sigma|+1} \Big[ \mathcal{R}\big( \mathrm{Ins}(s_\sigma, n, x) \big) - \mathcal{R}(s_\sigma) \Big]. \tag{16}$$

Equivalently, let $\mathrm{pos}_\sigma(n) := |s_\sigma| + 1$ be the position of $n$ in $\sigma$ (one plus its predecessor length); then

$$\phi_{\mathrm{SSV}}(n) \ = \ \frac{1}{|\mathfrak{S}(\mathcal{N})|} \sum_{\sigma \in \mathfrak{S}(\mathcal{N})} \frac{1}{\mathrm{pos}_\sigma(n)} \sum_{x=1}^{\mathrm{pos}_\sigma(n)} \Big[ \mathcal{R}\big( \mathrm{Ins}(s_\sigma, n, x) \big) - \mathcal{R}(s_\sigma) \Big]. \tag{17}$$

(2) Sum grouped by predecessor length. For $k \in \{0, \ldots, |\mathcal{N}| - 1\}$, let $\mathcal{O}_k(\mathcal{N} \setminus \{n\})$ be the set of all ordered $k$-tuples (orderings) drawn from $\mathcal{N} \setminus \{n\}$, and for $o = (o_1, \ldots, o_k) \in \mathcal{O}_k$ write $s_o := (\mathcal{Q}, o_1, \ldots, o_k)$. The probability that a uniformly random permutation has $o$ as the exact ordered predecessor of $n$ is $(|\mathcal{N}| - k - 1)!/|\mathcal{N}|!$.

Hence, the full calculation of SSV can be rewritten as follows:

$$\phi_{\mathrm{SSV}}(n) \ = \ \sum_{k=0}^{|\mathcal{N}|-1} \frac{(|\mathcal{N}| - k - 1)!}{|\mathcal{N}|!} \sum_{o \in \mathcal{O}_k(\mathcal{N} \setminus \{n\})} \frac{1}{k+1} \sum_{x=1}^{k+1} \Big[ \mathcal{R}\big( \mathrm{Ins}(s_o, n, x) \big) - \mathcal{R}(s_o) \Big]. \tag{18}$$

### D.1.1 PROOF OF THEOREM 1.1: COMPLETE CREDIT ACROSS REASONING CHAINS (SEQUENTIAL EFFICIENCY)

**Lemma 1** (Complete Credit Across Reasoning Chains). Let $n := |\mathcal{N}|$ and $\Pi = \mathfrak{S}(\mathcal{N})$ be the set of all orderings of $\mathcal{N}$. Sample $\sigma = (\sigma_1, \ldots, \sigma_n) \sim \mathrm{Unif}(\Pi)$ and, conditional on $\sigma$, sample $X_t \sim \mathrm{Unif}\{1, \ldots, t\}$ independently for $t = 1, \ldots, n$. Define the insertion process

$$s_0^\sigma := (\mathcal{Q}, \emptyset), \qquad s_t^\sigma := \mathrm{Ins}\big( s_{t-1}^\sigma, \sigma_t, X_t \big) \quad (t = 1, \ldots, n).$$

Then

$$\sum_{u \in \mathcal{N}} \phi_{\mathrm{SSV}}(u) \ = \ \mathbb{E}_{\sigma, X_{1:n}} \big[ \mathcal{R}(s_n^\sigma) - \mathcal{R}(s_0^\sigma) \big].$$

In particular, if $\mathcal{R}(s_0^\sigma) = \mathcal{R}(\mathcal{Q}, \emptyset) = 0$, then $\sum_{u \in \mathcal{N}} \phi_{\mathrm{SSV}}(u) = \mathbb{E}_{\sigma, X_{1:n}} \big[ \mathcal{R}(s_n^\sigma) \big]$.

*Proof.* We proceed in four steps. Throughout, we assume $\mathcal{R}$ is integrable so that interchanging sums and expectations is justified.

Step 1: Unfold the SSV definition. By Eq. 15, for each $u \in \mathcal{N}$ and ordering $\sigma$ let $T(u, \sigma) \in \{1, \ldots, n\}$ be the unique index such that $\sigma_{T(u,\sigma)} = u$, and let

$$s_\sigma(u) \ := \ (\mathcal{Q}, \mathrm{pred}_\sigma(u)) \ = \ s_{T(u,\sigma)-1}^\sigma.$$

Then

$$\phi_{\mathrm{SSV}}(u) = \mathbb{E}_\sigma \, \mathbb{E}_{x \in \{1, \ldots, |s_\sigma(u)|+1\}} \Big[ \mathrm{SMC}_u \big( s_\sigma(u), x \big) \Big]. \tag{19}$$

Step 2: One-step increment identity. Fix $\sigma$ and $t := T(u, \sigma)$. Since $|s_\sigma(u)| = t - 1$ and $X_t \sim \mathrm{Unif}\{1, \ldots, t\}$,

$$\mathbb{E}_x \Big[ \mathrm{SMC}_u \big( s_\sigma(u), x \big) \Big] = \mathbb{E}_{X_t} \Big[ \mathcal{R}\big( \mathrm{Ins}(s_{t-1}^\sigma, u, X_t) \big) - \mathcal{R}(s_{t-1}^\sigma) \Big]$$

$$= \mathbb{E}_{X_t} \Big[ \mathcal{R}(s_t^\sigma) - \mathcal{R}(s_{t-1}^\sigma) \Big], \tag{20}$$

where the last equality uses the construction $s_t^\sigma := \mathrm{Ins}(s_{t-1}^\sigma, \sigma_t, X_t)$ and the fact that $u = \sigma_t$.

Step 3: Reindex by insertion time and telescope. Summing Eq. 19 over $u \in \mathcal{N}$ and applying Eq. 20 yields

$$
\begin{aligned}
\sum_{u \in \mathcal{N}} \phi_{\text{SSV}}(u) &= \mathbb{E}_{\sigma} \sum_{u \in \mathcal{N}} \mathbb{E}_{X_{T(u,\sigma)}} \Big[ \mathcal{R}(s^{\sigma}_{T(u,\sigma)}) - \mathcal{R}(s^{\sigma}_{T(u,\sigma)-1}) \Big] \\
&= \mathbb{E}_{\sigma} \sum_{t=1}^{n} \mathbb{E}_{X_t} \Big[ \mathcal{R}(s^{\sigma}_t) - \mathcal{R}(s^{\sigma}_{t-1}) \Big] \\
&= \mathbb{E}_{\sigma} \, \mathbb{E}_{X_{1:n}} \Big[ \sum_{t=1}^{n} \big( \mathcal{R}(s^{\sigma}_t) - \mathcal{R}(s^{\sigma}_{t-1}) \big) \Big] \\
&= \mathbb{E}_{\sigma, \, X_{1:n}} \Big[ \mathcal{R}(s^{\sigma}_n) - \mathcal{R}(s^{\sigma}_0) \Big],
\end{aligned}
$$

where we used that $\{T(u, \sigma) : u \in \mathcal{N}\} = \{1, \ldots, n\}$ and the inner sum telescopes.

Step 4: Conclude the stated identities. This proves the main identity. If $\mathcal{R}(s^{\sigma}_0) = 0$, the special case follows immediately. $\qquad\square$

### D.1.2 Proof of Theorem 1.2: Loss-Free Credit Preservation in Chain Combination (Sequential Additivity)

**Lemma 2** (Loss-Free Credit Preservation in Chain Combination). Let $\mathcal{R}_1, \mathcal{R}_2 : \mathcal{S} \to \mathbb{R}$ be reward functions and let $\alpha, \beta \in \mathbb{R}$. Define the combined reward $\mathcal{R}_{\oplus} := \alpha \mathcal{R}_1 + \beta \mathcal{R}_2$. Then for every $n \in \mathcal{N}$,
$$
\phi^{\mathcal{R}_{\oplus}}_{\text{SSV}}(n) = \alpha \, \phi^{\mathcal{R}_1}_{\text{SSV}}(n) + \beta \, \phi^{\mathcal{R}_2}_{\text{SSV}}(n).
$$
In particular, for $\alpha = \beta = 1$ we have $\phi^{\mathcal{R}_1 + \mathcal{R}_2}_{\text{SSV}}(n) = \phi^{\mathcal{R}_1}_{\text{SSV}}(n) + \phi^{\mathcal{R}_2}_{\text{SSV}}(n)$.

*Proof.* We establish that the SSV operator is linear in the reward.

Step 1: Linearity of SMC in the reward. For any chain $s \in \mathcal{S}$, admissible position $x \in \{1, \ldots, |s| + 1\}$, and $n \in \mathcal{N}$,

$$
\begin{aligned}
\text{SMC}^{\mathcal{R}_{\oplus}}_n(s, x) &:= \mathcal{R}_{\oplus}\big( \text{Ins}(s, n, x) \big) - \mathcal{R}_{\oplus}(s) \\
&= \alpha \Big[ \mathcal{R}_1\big( \text{Ins}(s, n, x) \big) - \mathcal{R}_1(s) \Big] + \beta \Big[ \mathcal{R}_2\big( \text{Ins}(s, n, x) \big) - \mathcal{R}_2(s) \Big] \\
&= \alpha \, \text{SMC}^{\mathcal{R}_1}_n(s, x) + \beta \, \text{SMC}^{\mathcal{R}_2}_n(s, x).
\end{aligned} \tag{21}
$$

This uses only the algebra of differences and does not depend on the dynamics.

Step 2: Take expectations in the SSV definition. Fix $n \in \mathcal{N}$ and write $s_{\sigma} = (\mathcal{Q}, \text{pred}_{\sigma}(n))$. By Eq. 15 and Eq. 21,

$$
\begin{aligned}
\phi^{\mathcal{R}_{\oplus}}_{\text{SSV}}(n) &= \mathbb{E}_{\sigma \in \mathfrak{S}(\mathcal{N})} \, \mathbb{E}_{x \in \{1, \ldots, |s_{\sigma}|+1\}} \Big[ \text{SMC}^{\mathcal{R}_{\oplus}}_n(s_{\sigma}, x) \Big] \\
&= \mathbb{E}_{\sigma} \, \mathbb{E}_{x} \Big[ \alpha \, \text{SMC}^{\mathcal{R}_1}_n(s_{\sigma}, x) + \beta \, \text{SMC}^{\mathcal{R}_2}_n(s_{\sigma}, x) \Big] \\
&= \alpha \, \mathbb{E}_{\sigma} \, \mathbb{E}_{x} \Big[ \text{SMC}^{\mathcal{R}_1}_n(s_{\sigma}, x) \Big] + \beta \, \mathbb{E}_{\sigma} \, \mathbb{E}_{x} \Big[ \text{SMC}^{\mathcal{R}_2}_n(s_{\sigma}, x) \Big] \\
&= \alpha \, \phi^{\mathcal{R}_1}_{\text{SSV}}(n) + \beta \, \phi^{\mathcal{R}_2}_{\text{SSV}}(n),
\end{aligned}
$$

where linearity of expectation justifies moving $\alpha, \beta$ outside the integrals.

Step 3: Conclude. The desired identities follow immediately, including the additivity case $\alpha = \beta = 1$. $\qquad\square$

### D.1.3 Proof of Theorem 1.3: Equitable Credit Across All Possible Chains (Sequential Symmetry)

**Lemma 3** (Equitable Credit Across All Possible Chains). Suppose $i, j \in \mathcal{N}$ are *symmetric* in the following sense: for every chain $s \in \mathcal{S}$ and every admissible position $x$, if $s^{\text{swap}}$ is obtained from $s$ by swapping the labels $i \leftrightarrow j$, then
$$
\mathcal{R}(s) = \mathcal{R}(s^{\text{swap}}) \quad \text{and} \quad \mathcal{R}\big( \text{Ins}(s, i, x) \big) = \mathcal{R}\big( \text{Ins}(s^{\text{swap}}, j, x) \big).
$$

Then $\phi_{\text{SSV}}(i) = \phi_{\text{SSV}}(j)$.

*Proof.* Let $i, j \in \mathcal{N}$ satisfy the symmetry condition of Lemma 3. We prove $\phi_{\text{SSV}}(i) = \phi_{\text{SSV}}(j)$ by constructing a weight-preserving bijection between $i$-contexts and $j$-contexts that preserves marginal contributions.

Step 1: Triple spaces and SSV regrouping. Define

$$\mathcal{T}_i = \{(S, \tau, k) : S \subseteq \mathcal{N} \setminus \{i\}, \ \tau \in \mathfrak{S}(S), \ 0 \leq k \leq |S|\},$$

$$\mathcal{T}_j = \{(T, \sigma, m) : T \subseteq \mathcal{N} \setminus \{j\}, \ \sigma \in \mathfrak{S}(T), \ 0 \leq m \leq |T|\}.$$

For $(S, \tau)$, let $s(S, \tau)$ be the predecessor chain formed by ordering $S$ according to $\tau$ (suppressing $\mathcal{Q}$ in notation). For $u \in \{i, j\}$ define

$$\text{SMC}_u(S, \tau, k) := \mathcal{R}\big(\text{Ins}(s(S, \tau), u, k{+}1)\big) - \mathcal{R}\big(s(S, \tau)\big),$$

where gaps are indexed by $k \in \{0, \ldots, |S|\}$ and Ins uses 1-based positions. Regrouping the SSV expectation over orderings and positions yields

$$\phi_{\text{SSV}}(u) = \sum_{(S, \tau, k) \in \mathcal{T}_u} w(S, k) \, \text{SMC}_u(S, \tau, k), \qquad w(S, k) = \frac{|S|! \, (|\mathcal{N}| - |S| - 1)!}{|\mathcal{N}|!} \cdot \frac{1}{|S| + 1},$$

so $w(S, k)$ depends only on $|S|$ and the gap index $k$.

Step 2: Construct a bijection $\Phi : \mathcal{T}_i \to \mathcal{T}_j$. Given $(S, \tau, k) \in \mathcal{T}_i$, define $\Phi(S, \tau, k) = (T, \sigma, m)$ as follows.

*(a) Set mapping.*

$$T = \begin{cases} S, & j \notin S, \\ (S \setminus \{j\}) \cup \{i\}, & j \in S. \end{cases}$$

*(b) Sequence and gap mapping.* Let $\text{repl}(\tau, j \to i)$ be the sequence obtained by replacing $j$ with $i$ while preserving the relative order of all other elements.

$$\sigma = \begin{cases} \tau, & j \notin S, \\ \text{repl}(\tau, j \to i), & j \in S, \end{cases} \qquad \boxed{m = k}.$$

Since swapping labels does not change the chain length nor the set of gaps, the gap index is preserved.

*Example.* If $\tau = (a, j, b, c)$, then $\sigma = (a, i, b, c)$ and the gap mapping is $m = k$ for all $k \in \{0, \ldots, 4\}$.

Step 3: Equal contributions under symmetry. By construction, $s(T, \sigma)$ is obtained from $s(S, \tau)$ by swapping labels $i \leftrightarrow j$ while fixing all other labels and their order. By the lemma's symmetry condition on $\mathcal{R}$ (for the predecessor chain and for single-step insertions at the *same* gap),

$$\mathcal{R}\big(\text{Ins}(s(S, \tau), i, k{+}1)\big) - \mathcal{R}\big(s(S, \tau)\big) = \mathcal{R}\big(\text{Ins}(s(T, \sigma), j, m{+}1)\big) - \mathcal{R}\big(s(T, \sigma)\big),$$

with $m = k$. Hence $\text{SMC}_i(S, \tau, k) = \text{SMC}_j(T, \sigma, k)$.

Step 4: Injectivity, surjectivity, and weight preservation. Define $\Phi^{-1} : \mathcal{T}_j \to \mathcal{T}_i$ by symmetrically swapping $i \leftrightarrow j$: for $(T, \sigma, m)$, set

$$S = \begin{cases} T, & i \notin T, \\ (T \setminus \{i\}) \cup \{j\}, & i \in T, \end{cases} \qquad \tau = \begin{cases} \sigma, & i \notin T, \\ \text{repl}(\sigma, i \to j), & i \in T, \end{cases} \qquad k = m.$$

Then $\Phi^{-1}(\Phi(S, \tau, k)) = (S, \tau, k)$ and $\Phi(\Phi^{-1}(T, \sigma, m)) = (T, \sigma, m)$, so $\Phi$ is a bijection. Moreover, $|T| = |S|$ and $m = k$ imply $w(T, m) = w(S, k)$.

Step 5: Conclude by summation. Summing equal marginal contributions with preserved weights,

$$\phi_{\text{SSV}}(i) = \sum_{(S, \tau, k) \in \mathcal{T}_i} w(S, k) \, \text{SMC}_i(S, \tau, k) = \sum_{(T, \sigma, k) \in \mathcal{T}_j} w(T, k) \, \text{SMC}_j(T, \sigma, k) = \phi_{\text{SSV}}(j).$$

$\square$

### D.1.4 PROOF OF THEOREM 1.4: ZERO CREDIT TO NULL OR REDUNDANT STEPS (SEQUENTIAL DUMMY)

**Lemma 4** (Zero Credit to Null or Redundant Steps)**.** If $i \in \mathcal{N}$ satisfies $\mathcal{R}(\mathrm{Ins}(s, i, x)) = \mathcal{R}(s)$ for every $s \in \mathcal{S}$ and every $x \in \{1, \ldots, |s| + 1\}$ (equivalently, $\mathrm{SMC}_i(s, x) = 0$ for all admissible $(s, x)$), then $\phi_{\mathrm{SSV}}(i) = 0$.

*Proof.* Step 1: SSV as an expectation over $i$-contexts. For fixed $i$, define the context space

$$\Omega_i := \{(\sigma, x) : \ \sigma \in \mathfrak{S}(\mathcal{N}), \ x \in \{1, \ldots, |s_\sigma(i)| + 1\}\},$$

where $s_\sigma(i) := (\mathcal{Q}, \mathrm{pred}_\sigma(i))$ is the predecessor chain of $i$ under $\sigma$. Let $\mu_i$ be the product measure induced by $\sigma \sim \mathrm{Unif}(\mathfrak{S}(\mathcal{N}))$ and, given $\sigma$, $x \sim \mathrm{Unif}\{1, \ldots, |s_\sigma(i)| + 1\}$. Define

$$f_i(\sigma, x) := \mathrm{SMC}_i\big(s_\sigma(i), x\big).$$

By the definition of SSV (Eq. 2), we have

$$\phi_{\mathrm{SSV}}(i) = \mathbb{E}_{(\sigma, x) \sim \mu_i}\big[f_i(\sigma, x)\big].$$

Step 2: Apply the null (redundant) condition. By hypothesis, for every admissible $(s, x)$ one has $\mathrm{SMC}_i(s, x) = 0$. In particular, for all $(\sigma, x) \in \Omega_i$,

$$f_i(\sigma, x) = \mathrm{SMC}_i\big(s_\sigma(i), x\big) = 0.$$

Hence

$$\phi_{\mathrm{SSV}}(i) = \mathbb{E}_{\mu_i}[f_i(\sigma, x)] = \mathbb{E}_{\mu_i}[\,0\,] = 0,$$

which proves the claim.

$\square$

### D.2 PROOF OF THEOREM 2

Fix the insertion MDP $(\mathcal{Q}, \mathcal{N}, \mathcal{S}, \mathcal{A}, \mathcal{P}, \mathcal{R})$. For $n \in \mathcal{N}$, a chain $s = (\mathcal{Q}, n_{1:t}) \in \mathcal{S}$, and an insertion position $x \in \{1, \ldots, |s| + 1\}$,

$$\mathrm{SMC}_n(s, x) := \mathcal{R}(\mathrm{Ins}(s, n, x)) - \mathcal{R}(s).$$

Let $\mathfrak{S}(\mathcal{N})$ be all orderings of $\mathcal{N}$ and, for $\sigma \in \mathfrak{S}(\mathcal{N})$, set $s_\sigma := (\mathcal{Q}, \mathrm{pred}_\sigma(n))$. Define the *Sequential Shapley Value* (SSV)

$$\phi_{\mathrm{SSV}}(n) := \mathbb{E}_{\Sigma \sim \mathrm{Unif}(\mathfrak{S}(\mathcal{N}))}\, \mathbb{E}_{X|\Sigma \sim \mathrm{Unif}\{1, \ldots, |s_\Sigma| + 1\}}\big[\mathrm{SMC}_n(s_\Sigma, X)\big], \qquad s_\Sigma = (\mathcal{Q}, \mathrm{pred}_\Sigma(n)).$$

We also write $Y_n := \mathrm{SMC}_n(s_\Sigma, X)$ and $\mathcal{G} := \sigma(\mathcal{Q}, n)$ (allowed information: prompt and step identity).

We prove the following three claims for the SSV advantage $A_t^{\mathrm{SSV}} := \mathrm{SMC}_{n_t}(s_{t-1}, x_t) - \phi_{\mathrm{SSV}}(n_t)$, under the assumptions that (i) insert transitions are deterministic, (ii) $\phi_{\mathrm{SSV}}(n_t)$ depends only on $(\mathcal{Q}, n_t)$, is independent of $x_t$, and is used with stop-gradient, and (iii) its Monte Carlo estimate (when used) is computed from data independent of the rollout being updated.

### D.2.1 PROOF OF THEOREM 2.1: SSV AS AN UNBIASED VALUE BASELINE (ACTION-INDEPENDENCE)

Let the rollout be at state $s_t = (\mathcal{Q}, n_{1:t-1}) \in \mathcal{S}$ and sample $a_t \sim \pi_\theta(\cdot \mid s_t)$. For step $n \in \mathcal{N}$, let its realized marginal contribution on this rollout be

$$\mathrm{SMC}_n^{\mathrm{real}} = \mathrm{SMC}_n\big(s^{\mathrm{real}}, x^{\mathrm{real}}\big).$$

Let the baseline $B_n \in \{\phi_{\mathrm{SSV}}(n), \widehat{\phi}_{\mathrm{SSV},K}(n)\}$ depend only on $(\mathcal{Q}, n)$, be independent of the current action, and be used with stop-gradient. Define $A_n := \mathrm{SMC}_n^{\mathrm{real}} - B_n$.

We aim to show

$$\mathbb{E}\big[\nabla_\theta \log \pi_\theta(a_t \mid s_t)\, A_n\big] = \mathbb{E}\big[\nabla_\theta \log \pi_\theta(a_t \mid s_t)\, \mathrm{SMC}_n^{\mathrm{real}}\big].$$

**Lemma 5** (Action-independence baseline $\Rightarrow$ unbiased policy gradient)**.** If a baseline $B$ is measurable w.r.t. $\mathcal{G} := \sigma(\mathcal{Q}, n)$, independent of the current action, and used with stop-gradient, then $\mathbb{E}[\nabla_\theta \log \pi_\theta(a_t \mid s_t) B] = 0$; hence using $A_n = \mathrm{SMC}_n^{\mathrm{real}} - B$ yields an unbiased policy-gradient estimator.

*Proof.* Step 1: Tower property (condition on $(s_t, B_n)$). With $Z := \nabla_\theta \log \pi_\theta(a_t \mid s_t) A_n$ and $\mathcal{G} := \sigma(s_t, B_n)$,

$$\mathbb{E}[Z] = \mathbb{E}\big[\mathbb{E}[Z \mid \mathcal{G}]\big] = \mathbb{E}\Big[\mathbb{E}\big[\nabla_\theta \log \pi_\theta(a_t \mid s_t) A_n \mid s_t, B_n\big]\Big]. \tag{22}$$

Step 2: Expand $A_n$ and separate the baseline. Since $A_n = \mathrm{SMC}_n^{\mathrm{real}} - B_n$,

$$\mathbb{E}\big[\nabla \log \pi A_n \mid s_t, B_n\big] = \mathbb{E}\big[\nabla \log \pi \, \mathrm{SMC}_n^{\mathrm{real}} \mid s_t, B_n\big] - \mathbb{E}\big[\nabla \log \pi B_n \mid s_t, B_n\big]$$
$$= \mathbb{E}\big[\nabla \log \pi \, \mathrm{SMC}_n^{\mathrm{real}} \mid s_t, B_n\big] - B_n \, \mathbb{E}\big[\nabla \log \pi \mid s_t, B_n\big], \tag{23}$$

where $\nabla \log \pi$ abbreviates $\nabla_\theta \log \pi_\theta(a_t \mid s_t)$ and $B_n$ is constant under the inner conditioning.

Step 3: Use independence and the score-function identity. Independence (cross-fitting/stop-grad) implies

$$\mathbb{E}\big[\nabla \log \pi \mid s_t, B_n\big] = \mathbb{E}\big[\nabla \log \pi \mid s_t\big] = \sum_a \pi_\theta(a \mid s_t) \nabla_\theta \log \pi_\theta(a \mid s_t) \tag{24}$$

$$= \nabla_\theta \sum_a \pi_\theta(a \mid s_t) = \nabla_\theta 1 = 0. \tag{25}$$

Hence the baseline term in Eq. 23 vanishes and

$$\mathbb{E}\big[\nabla \log \pi A_n \mid s_t, B_n\big] = \mathbb{E}\big[\nabla \log \pi \, \mathrm{SMC}_n^{\mathrm{real}} \mid s_t, B_n\big]. \tag{26}$$

Step 4: Remove conditioning via the tower rule. Substitute Eq. 26 into Eq. 22 to obtain

$$\mathbb{E}\big[\nabla_\theta \log \pi_\theta(a_t \mid s_t) A_n\big] = \mathbb{E}\big[\nabla_\theta \log \pi_\theta(a_t \mid s_t) \mathrm{SMC}_n^{\mathrm{real}}\big]. \tag{27}$$

$\square$

### D.2.2 PROOF OF THEOREM 2.2: SSV AS A VARIANCE-MINIMAL VALUE BASELINE (OPTIMALITY)

Let $\mathfrak{S}(\mathcal{N})$ be all orderings of $\mathcal{N}$. For $\Sigma \sim \mathrm{Unif}(\mathfrak{S}(\mathcal{N}))$, set $s_\Sigma = (\mathcal{Q}, \mathrm{pred}_\Sigma(n))$ and draw $X \mid \Sigma \sim \mathrm{Unif}\{1, \ldots, |s_\Sigma| + 1\}$. Define the random variable

$$Y_n := \mathrm{SMC}_n(s_\Sigma, X). \tag{28}$$

Let the allowed information be the $\sigma$-algebra $\mathcal{G} := \sigma(\mathcal{Q}, n)$. Consider the class of legal baselines

$$\mathcal{B} := \big\{ b : (\mathcal{Q}, n) \mapsto \mathbb{R} \text{ and } b \text{ is } \mathcal{G}\text{-measurable} \big\}. \tag{29}$$

We aim to minimize, over $b \in \mathcal{B}$, the residual variance

$$\mathcal{V}(b) := \mathrm{Var}\big(Y_n - b(\mathcal{Q}, n)\big). \tag{30}$$

**Lemma 6** ($\mathcal{G}$-measurable baseline $\Rightarrow$ variance is minimized if $b = \phi_{\mathrm{SSV}}$)**.** If a baseline $b$ is $\mathcal{G}$-measurable, then $\mathrm{Var}(Y_n - b) = \mathbb{E}[\mathrm{Var}(Y_n \mid \mathcal{G})] + \mathrm{Var}(\mathbb{E}[Y_n \mid \mathcal{G}] - b)$, and thus the variance is minimized if and only if $b(\mathcal{Q}, n) = \mathbb{E}[Y_n \mid \mathcal{G}] = \phi_{\mathrm{SSV}}(n)$.

*Proof.* Step 1: Law of total variance. Apply the law of total variance to $Z := Y_n - b(\mathcal{Q}, n)$ with respect to $\mathcal{G}$:

$$\mathcal{V}(b) = \mathbb{E}\big[\mathrm{Var}(Z \mid \mathcal{G})\big] + \mathrm{Var}\big(\mathbb{E}[Z \mid \mathcal{G}]\big)$$
$$= \mathbb{E}\big[\mathrm{Var}\big(Y_n - b(\mathcal{Q}, n) \mid \mathcal{G}\big)\big] + \mathrm{Var}\big(\mathbb{E}\big[Y_n - b(\mathcal{Q}, n) \mid \mathcal{G}\big]\big). \tag{31}$$

Step 2: Use $\mathcal{G}$-measurability of $b$. Since $b(\mathcal{Q}, n)$ is $\mathcal{G}$-measurable, it is constant under $\mathcal{G}$-conditioning. Hence

$$\mathrm{Var}\big(Y_n - b(\mathcal{Q}, n) \mid \mathcal{G}\big) = \mathrm{Var}(Y_n \mid \mathcal{G}), \tag{32}$$

$$\mathbb{E}\big[Y_n - b(\mathcal{Q}, n) \mid \mathcal{G}\big] = \mathbb{E}[Y_n \mid \mathcal{G}] - b(\mathcal{Q}, n). \tag{33}$$

Substitute Eq. 32–Eq. 33 into Eq. 31:

$$\mathcal{V}(b) = \underbrace{\mathbb{E}\big[\mathrm{Var}(Y_n \mid \mathcal{G})\big]}_{\text{independent of } b} + \mathrm{Var}\big(\mathbb{E}[Y_n \mid \mathcal{G}] - b(\mathcal{Q}, n)\big). \tag{34}$$

Step 3: Pointwise minimization. For each realization of $(\mathcal{Q}, n)$, the second term in Eq. 34 is a nonnegative variance which is minimized to $0$ if and only if

$$b(\mathcal{Q}, n) = \mathbb{E}[Y_n \mid \mathcal{G}] = \mathbb{E}_{\Sigma, X}\big[\mathrm{SMC}_n(s_\Sigma, X) \mid \mathcal{Q}, n\big]. \tag{35}$$

Therefore, the unique minimizer is

$$b^\star(\mathcal{Q}, n) = \mathbb{E}[Y_n \mid \mathcal{Q}, n] = \phi_{\mathrm{SSV}}(n), \qquad \min_{b \in \mathcal{B}} \mathcal{V}(b) = \mathbb{E}\big[\mathrm{Var}(Y_n \mid \mathcal{G})\big]. \tag{36}$$

This proves the variance-optimality of SSV within the allowed information class. □

### D.2.3 Proof of Theorem 2.3: Convergence of the Unbiased, Variance-Minimal SSV Value Baseline (Policy Optimization)

Let $(\Sigma_k, X_k)_{k=1}^K$ be i.i.d. copies of $(\Sigma, X)$ as defined above and set

$$\widehat{\phi}_{\mathrm{SSV}, K}(n) := \frac{1}{K} \sum_{k=1}^K \mathrm{SMC}_n\big(s_{\Sigma_k}, X_k\big), \qquad s_{\Sigma_k} = (\mathcal{Q}, \mathrm{pred}_{\Sigma_k}(n)). \tag{37}$$

All draws are generated in an evaluation procedure independent of the rollout being updated (cross-fitting), and the estimate is used with stop-gradient.

**Lemma 7** (Independent i.i.d. orderings $\Rightarrow$ unbiased MC SSV with $O(K^{-1/2})$ rate). If $(\Sigma_k, X_k)_{k=1}^K$ are i.i.d. and independent of the rollout being updated, then $\widehat{\phi}_{\mathrm{SSV}, K}(n) = \frac{1}{K} \sum_{k=1}^K \mathrm{SMC}_n(s_{\Sigma_k}, X_k)$ satisfies $\mathbb{E}[\widehat{\phi}_{\mathrm{SSV}, K}(n) \mid \mathcal{Q}, n] = \phi_{\mathrm{SSV}}(n)$ and $\mathrm{Var}(\widehat{\phi}_{\mathrm{SSV}, K}(n) \mid \mathcal{Q}, n) = \mathrm{Var}(Y_n \mid \mathcal{Q}, n)/K$ (hence sd $= O(K^{-1/2})$).

*Proof.* Step 1: Unbiasedness. Conditioning on $(\mathcal{Q}, n)$ and using linearity of expectation,

$$\mathbb{E}\Big[\widehat{\phi}_{\mathrm{SSV}, K}(n) \mid \mathcal{Q}, n\Big] = \frac{1}{K} \sum_{k=1}^K \mathbb{E}[\mathrm{SMC}_n(s_{\Sigma_k}, X_k) \mid \mathcal{Q}, n]$$

$$= \mathbb{E}_{\Sigma, X}[\mathrm{SMC}_n(s_\Sigma, X) \mid \mathcal{Q}, n] = \phi_{\mathrm{SSV}}(n). \tag{38}$$

Step 2: Variance and $O(K^{-1/2})$ rate. By independence of the $K$ summands and identical conditional variance,

$$\mathrm{Var}\Big(\widehat{\phi}_{\mathrm{SSV}, K}(n) \mid \mathcal{Q}, n\Big) = \mathrm{Var}\left( \frac{1}{K} \sum_{k=1}^K \mathrm{SMC}_n(s_{\Sigma_k}, X_k) \,\middle|\, \mathcal{Q}, n \right)$$

$$= \frac{1}{K^2} \sum_{k=1}^K \mathrm{Var}(\mathrm{SMC}_n(s_{\Sigma_k}, X_k) \mid \mathcal{Q}, n)$$

$$= \frac{1}{K} \mathrm{Var}(Y_n \mid \mathcal{Q}, n). \tag{39}$$

Thus sd$(\widehat{\phi}_{\mathrm{SSV}, K}(n)) = O(K^{-1/2})$.

Step 3: Consistency and CLT. By the strong law of large numbers (SLLN),

$$\widehat{\phi}_{\mathrm{SSV}, K}(n) \xrightarrow{\text{a.s.}} \phi_{\mathrm{SSV}}(n) \qquad \text{as } K \to \infty. \tag{40}$$

By the (Lindeberg–Feller) central limit theorem,

$$\sqrt{K}\left(\widehat{\phi}_{\mathrm{SSV},K}(n) - \phi_{\mathrm{SSV}}(n)\right) \Rightarrow \mathcal{N}\left(0,\ \mathrm{Var}(Y_n \mid \mathcal{Q}, n)\right), \qquad K \to \infty. \tag{41}$$

**Step 4: Impact on the advantage variance (with cross-fitting).** Let the step-level advantage used for policy updates be

$$A_n := \mathrm{SMC}_n^{\mathrm{real}} - \widehat{\phi}_{\mathrm{SSV},K}(n), \qquad \mathrm{SMC}_n^{\mathrm{real}} = \mathrm{SMC}_n(s^{\mathrm{real}}, x^{\mathrm{real}}). \tag{42}$$

If $\widehat{\phi}_{\mathrm{SSV},K}(n)$ is computed independently of the rollout (cross-fitting), then $\mathrm{SMC}_n^{\mathrm{real}} \perp \widehat{\phi}_{\mathrm{SSV},K}(n)$ and

$$\mathrm{Var}(A_n) = \mathrm{Var}\left(\mathrm{SMC}_n^{\mathrm{real}}\right) + \mathrm{Var}\left(\widehat{\phi}_{\mathrm{SSV},K}(n)\right)$$

$$\approx \underbrace{\mathbb{E}[\mathrm{Var}(Y_n \mid \mathcal{Q}, n)]}_{\text{information-theoretic lower bound from Eq. 36}} + \underbrace{\frac{\mathbb{E}[\mathrm{Var}(Y_n \mid \mathcal{Q}, n)]}{K}}_{\text{MC estimation noise}}. \tag{43}$$

Hence, the advantage variance approaches the theoretical lower bound at the $O(1/K)$ rate. □

# E ADDITIONAL EXPERIMENTAL RESULTS

## E.1 EVALUATION ON QWEN3-1.7B

To examine the robustness of SSVPO across model scales and task difficulties, we conduct additional experiments using the Qwen3-1.7B backbone, one of the strongest publicly available small models, and extend the evaluation to OLYMPIADBENCH, a benchmark featuring olympiad-level scientific problems. We compare four methods under a unified training and evaluation setup: Vanilla, RLOO, VinePPO, and SSVPO. For each dataset, we report accuracy (Acc) and average response length (Len). The results are summarised in Table 4.

Table 4: Performance of Vanilla, RLOO, VinePPO and SSVPO on Qwen3-1.7B across all benchmarks.

| | MultiArith | | SVAMP | | GSM8K | | MATH-500 | | AMC23 | | AIME24 | | AIME25 | | OlympiadBench | |
|---|---|---|---|---|---|---|---|---|---|---|---|---|---|---|---|---|
| Method | Acc | Len | Acc | Len | Acc | Len | Acc | Len | Acc | Len | Acc | Len | Acc | Len | Acc | Len |
| Vanilla | 94.7 | 1034 | 94.0 | 1631 | 90.6 | 2437 | 77.4 | 5503 | 75.0 | 9159 | 40.0 | 13697 | 23.3 | 13724 | 55.9 | 9635 |
| RLOO | 98.9 | 966 | 93.3 | 1537 | 90.1 | 2265 | 78.4 | 4925 | 77.5 | 8866 | 43.3 | 13599 | 33.3 | 13285 | 56.2 | 9408 |
| VinePPO | 99.4 | 887 | 93.3 | 1123 | 90.4 | 1692 | 79.6 | 4077 | 75.0 | 7085 | 43.3 | 12788 | 33.3 | 12130 | 55.0 | 7717 |
| SSVPO | 99.4 | 740 | 99.4 | 1258 | 91.8 | 1706 | 82.0 | 4218 | 80.0 | 8110 | 43.3 | 13027 | 40.0 | 12987 | 57.6 | 9257 |

As shown in Table 4, SSVPO performs strongly across all benchmarks under the Qwen3-1.7B setting. On *simple reasoning tasks*, SSVPO achieves the best performance on both MultiArith and SVAMP, reaching 99.4% accuracy on each. Notably, these gains come with the shortest generation lengths among all methods (740 tokens on MultiArith and 1,258 tokens on SVAMP), indicating that SSVPO improves reasoning efficiency rather than relying on longer chains-of-thought.

For *medium-difficulty tasks*, SSVPO consistently outperforms all baselines on GSM8K and MATH-500. In particular, SSVPO attains 91.8% accuracy on GSM8K and 82.0% on MATH-500, surpassing both outcome-based methods (Vanilla, RLOO) and the credit-assignment baseline VinePPO, while keeping response lengths comparable or shorter than strong baselines.

On *competition-level tasks*, SSVPO continues to show clear advantages. It achieves the highest accuracy on AMC23, AIME25, and OlympiadBench, while matching the strongest baseline on AIME24. For example, SSVPO improves AMC23 accuracy to 80.0%, raises AIME25 accuracy to 40.0%, and reaches 57.6% accuracy on OlympiadBench. Across these harder benchmarks, SSVPO also maintains competitive or shorter response lengths compared to RLOO and VinePPO, suggesting that its step-level credit assignment allows the model to solve more challenging problems without resorting to excessively long reasoning.

Overall, these results demonstrate that under a unified Qwen3-1.7B configuration, SSVPO delivers top-1 accuracy on all evaluated benchmarks while preserving or improving token efficiency, supporting its robustness and scalability across a wide range of reasoning difficulties.

## E.2 Training Dynamics and Step Effectiveness

We further analyse the training dynamics of SSVPO on a new backbone (Qwen3-1.7B) and study how the number of reordered steps affects optimisation. We vary the step-merging rule using three settings: relaxed, standard, and strict, which correspond to 2-step, 3-step, and 4-step reordering regimes, respectively. These settings control how many merged steps are subject to reordering during training. We track the validation score (Acc), average response length (Len), and KL divergence with respect to the reference policy (KL), and compare SSVPO against RLOO under the same configurations. The numerical results are summarised in Table 5.

Table 5: Training dynamics of SSVPO with different reordering regimes (2-step, 3-step, 4-step) on Qwen3-1.7B, compared with RLOO.

| Step | 2-step Acc | 2-step Len | 2-step KL | 3-step Acc | 3-step Len | 3-step KL | 4-step Acc | 4-step Len | 4-step KL | RLOO Acc | RLOO Len | RLOO KL |
|---|---|---|---|---|---|---|---|---|---|---|---|---|
| 20 | 72.6 | 4992 | +0.0029 | 73.2 | 4969 | +0.0030 | 73.2 | 5055 | +0.0031 | 71.8 | 4996 | +0.0103 |
| 40 | 73.4 | 4762 | +0.0008 | 74.0 | 5015 | +0.0004 | 75.0 | 4834 | -0.0017 | 70.0 | 4937 | +0.0140 |
| 60 | 72.0 | 4754 | -0.0013 | 75.0 | 4832 | -0.0080 | 76.2 | 4623 | -0.0010 | 72.8 | 4895 | +0.0193 |
| 80 | 74.0 | 4959 | -0.0034 | 76.2 | 4666 | -0.0031 | 76.4 | 4417 | -0.0037 | 72.0 | 4849 | +0.0197 |
| 100 | 74.6 | 4855 | -0.0024 | 74.2 | 4733 | -0.0032 | 75.0 | 4443 | -0.0038 | 72.2 | 4798 | +0.0235 |
| 120 | 74.6 | 4725 | -0.0031 | 72.4 | 4841 | -0.0037 | 76.2 | 4523 | -0.0044 | 73.8 | 4799 | +0.0238 |
| 140 | 74.2 | 4306 | -0.0025 | 73.0 | 4322 | -0.0046 | 70.8 | 4111 | -0.0033 | 73.6 | 4791 | +0.0259 |
| 160 | 75.0 | 4441 | -0.0026 | 74.6 | 4106 | -0.0046 | 78.2 | 4293 | -0.0043 | 74.4 | 4807 | +0.0251 |
| 180 | 76.6 | 4299 | -0.0009 | 76.0 | 4227 | -0.0015 | 77.2 | 4103 | -0.0020 | 75.2 | 4866 | +0.0264 |
| 200 | 78.0 | 4012 | +0.0001 | 77.8 | 4010 | -0.0012 | 78.2 | 4055 | -0.0005 | 76.8 | 4980 | +0.0267 |
| 220 | 78.6 | 3888 | +0.0011 | 79.0 | 3990 | +0.0002 | 81.0 | 4097 | -0.0002 | 77.4 | 4812 | +0.0263 |
| 240 | 79.4 | 3909 | +0.0018 | 80.0 | 3996 | +0.0013 | 82.0 | 4166 | +0.0004 | 77.8 | 4796 | +0.0265 |

**Accuracy vs. step length.** As the number of reordered steps increases ($2 \to 3 \to 4$), accuracy improves monotonically, but the response length also grows. For example, at step $= 240$, the accuracies of the 2-step, 3-step, and 4-step regimes are $0.794$, $0.800$, and $0.820$, while their average lengths are $3909$, $3996$, and $4166$ tokens, respectively. This indicates that larger reordering regimes enable stronger reasoning ability but require more tokens. By contrast, while RLOO's accuracy also improves with training, its response length remains around 4.8k tokens and shows no ability to shorten outputs.

**Entropy divergence dynamics.** SSVPO displays a characteristic "positive $\to$ negative $\to$ positive" KL trajectory, whereas RLOO remains strictly positive and monotonically increasing. This means that SSVPO initially suppresses high-probability regions of the reference policy, indicating wider exploration, and then gradually increases KL once better solutions are found. RLOO, in contrast, performs mostly one-sided exploitation around the reference policy, which may lead to entropy collapse and reduced output diversity. The negative-then-positive KL pattern of SSVPO thus provides a clear exploration advantage during training.

## F Implementation Details

### F.1 Pseudo Code

We provide pseudo-code of the proposed Sequential Shapley Value Policy Optimization (SSVPO) in Algorithm 1.

---

**Algorithm 1** Sequential Shapley Value Policy Optimization (SSVPO)

---

**Require:** Query $Q$, policy $\pi_\theta$, rollout count $N$
**Ensure:** Updated parameters $\theta$

1: **Sample & pool.** Generate $N$ CoT responses $\{p^j\}_{j=1}^N$ with $\pi_\theta$ on $Q$. Apply `extra` to obtain normalized step sequences $p^j = (s_{(1)}^j, \ldots, s_{(L_j)}^j)$ *(exclude the answer step)*. Let $\mathcal{N}$ be the set of distinct step ids aggregated from all $\{p^j\}$.

2: **Construct all chains (batched).** Let $\mathfrak{S}(\mathcal{N})$ be all orderings of $\mathcal{N}$. For each $n \in \mathcal{N}$ and $\sigma \in \mathfrak{S}(\mathcal{N})$, set $s_\sigma = (\mathcal{Q}, \mathrm{pred}_\sigma(n))$. Form the batched set

$$\mathcal{B} = \{ s_\sigma \mid \sigma \in \mathfrak{S}(\mathcal{N}) \} \cup \{ \mathrm{Ins}(s_\sigma, n, x) \mid \sigma \in \mathfrak{S}(\mathcal{N}), n \in \mathcal{N}, x \in \{1, \ldots, |s_\sigma|+1\} \}.$$

3: **Answer oracle (batched, same policy).** Define $(x)$: append the fixed exit prompt (`timeout; answer now; </think>`) to chain $x$, then decode *with the same policy* $\pi_\theta$ deterministically once to return $\mathcal{R}(x) \in \{0,1\}$. Forward all $x \in \mathcal{B}$ in batched calls to obtain $\{\mathcal{R}(x)\}_{x\in\mathcal{B}}$; cache $\mathcal{R}(\cdot)$.

4: **Compute SMC table.** For every $(\sigma, n, x)$ with $x \in \{1, \ldots, |s_\sigma|+1\}$, set

$$\mathrm{SMC}_n(s_\sigma, x) = \mathcal{R}(\mathrm{Ins}(s_\sigma, n, x)) - \mathcal{R}(s_\sigma).$$

5: **Compute SSV by Eq. 2** For each $n \in \mathcal{N}$,

$$\phi_{\mathrm{SSV}}(n) = \mathbb{E}_{\sigma \in \mathfrak{S}(\mathcal{N})} \mathbb{E}_{x \in \{1, \ldots, |s_\sigma|+1\}} \big[ \mathrm{SMC}_n(s_\sigma, x) \big].$$

6: **Advantages on original samples Eq.3 and PPO update Eq.4.**
7: **for** $j = 1$ to $N$ **do**
8:     **for** $t = 1$ to $L_j$ **do**
9:         $n_t \leftarrow \mathrm{id}(s_{(t)}^j)$;   $s_{-1} \leftarrow$ chain $p^j$ with step $t$ removed (original order preserved); choose $x_t$ such that $\mathrm{Ins}(s_{-1}, n_t, x_t) = p^j$
10:         $\hat{A}_x^T(s_{-1}, n_t, x_t) \leftarrow \mathrm{SMC}_{n_t}(s_{-1}, x_t)$
11:         $A_t^{\mathrm{SSV}} \leftarrow \hat{A}_x^T(s_{-1}, n_t, x_t) - \phi_{\mathrm{SSV}}(n_t)$
12:     **end for**
13:     Map $\{A_t^{\mathrm{SSV}}\}$ to tokens with standard GAE (per §3.3), then update $\theta$ using the PPO clipped objective:

$$\mathcal{J}_{\mathrm{SSVPO}}(\theta) = \mathbb{E}\big[ \min\{ r_t(\theta) A_t^{\mathrm{SSV}}, \ \mathrm{clip}(r_t(\theta), 1-c, 1+c) A_t^{\mathrm{SSV}} \} \big].$$

14: **end for**

---

### F.2 SOURCE CODE

We release the source code of SSVPO in an Anonymous GitHub repository (`https://anonymous.4open.science/r/SSV-CB75`). The repository includes complete installation instructions and versioned dependency requirements. Our implementation is built upon OPENRLHF (Hu et al., 2024) and extends it to support step-level credit assignment, whose core additions are: (1) a robust module for extracting mathematical reasoning steps, (2) operations for reordering the reasoning chain, and (3) computation of per-step advantage signals with propagation to the corresponding steps for learning. We also provide out-of-the-box training and evaluation scripts to reproduce our main results on GSM8K, MATH, AMC, and AIME.

## G DETAILED EXPERIMENTAL SETTINGS

### G.1 TRAINING CONFIGURATIONS

**Hyperparameters.** We provide the detailed hyperparameters for all experiments conducted in this study in Table 6. Unless otherwise noted, the global sampling budget and the maximum prompt/response lengths are kept the same across all experiments, while per-round sampling, number of rounds, training batch size, and the number of Monte Carlo (MC) samples are specific to the outcome-based and credit-assignment RL methods.

For QWEN3-4B, we use a sampling temperature of 0.6, the ADAM optimizer with a learning rate of 0.01, and an initial KL coefficient of 0.01. The log-probability clipping range is set to $[-10, 10]$. For DEEPSEEK-R1-DISTILL-QWEN-1.5B, we adopt the same hyperparameters as QWEN3-4B. For VINEPPO, we draw three rollouts from each intermediate state. For SPO, we set the number of segments to 3, sample three rollouts per segment, and apply a probability-mask threshold of 0.9. All other sampling settings (e.g., number of rollouts/rounds), epochs, and maximum generation lengths are exactly as reported in Table 6. Both VinePPO and SPO are trained with the random seed fixed to 42. For all methods, we use a discount factor of $\gamma = 1$.

**Computing Details.** For all datasets, experiments with QWEN3-4B and DEEPSEEK-R1-DISTILL-QWEN-1.5B are run on $4\times$ H200 GPUs per run: 1 GPU hosts the reference model, 2 GPUs host the actor model, and 1 GPU runs the vLLM server. The experiments are conducted on UBUNTU 22.04 operating system. For more details on the dependency libraries and configurations, please refer to the anonymous repository provided in Appendix F.2.

### G.2 PARTITION STRATEGY FOR REASONING CHAINS

**Step Segments from Initial Reasoning Chains.** We extract candidate reasoning steps solely based on mathematical segments to obtain a granularity that is robust to formatting and line breaks. Concretely, we first scan inline and display formulas ($...$/, $$...$$,...) as well as structural constructs (e.g., \frac, \sqrt, matrices/vectors, sets/intervals/tuples) in reasoning chains. Segments containing \boxed{...} are used only to obtain the final answer and are excluded from the reasoning steps, preventing the "answer" from being treated as intermediate reasoning steps. The extracted segments are then lightly normalized by removing TeX delimiters and redundant formatting commands (e.g., \left/\right, \quad), unifying multiplication symbols and whitespace, and folding equivalent forms (e.g., $(a + b)/c$ and \frac{a+b}{c}, $a^{1/2}$ and \sqrt{a}). Each segment is assigned a type label (equation/expr/fraction/number/inequality/set/interval_or_tuple/matrix/vector/radical/congruence, etc.) to serve as the coordinate system for subsequent granularity reduction and training-signal backfilling. The extraction pipeline guarantees determinism and idempotence: the same input under the same configuration yields the same step sequence and type distribution; when no recognizable mathematical anchors are present, only the

Table 6: Hyperparameter settings for all experiments. Global sampling budgets and maximum sequence lengths are as follows: MultiArith (prompt 2048, response 2048, budget 2000); SVAMP (prompt 3072, response 2048, budget 3000); GSM8K (prompt 5120, response 4096, budget 30000); MATH (prompt 10,000, response 8196, budget 30000). "Samples/Round" denotes the product of problems per round and rollouts per problem. For methods with internal structures, the type of rollout is indicated in parentheses. *Notes:* "probs" = problems per round; "rollouts" = responses per problem; "inter." = intermediate rollouts; "cutpoint" = cutpoint rollouts; "reorder" = reordered chains.

| Dataset | Method | Samples/Round | Rounds | Batch | MC |
|---|---|---|---|---|---|
| MultiArith | RLOO / GRPO / DAPO | $16 \times 4$ | 27 | 64 | 64 |
| | VinePPO | $16 \times$ inter. (16) | 7 | 256 | $\leq 256$ |
| | SPO | $16 \times$ cutpoint (9) | 15 | 128 | $\leq 128$ |
| | SSVPO | $16 \times$ reorder (8) | 27 | 64 | $\leq 64$ |
| SVAMP | RLOO / GRPO / DAPO | $16 \times 4$ | 44 | 64 | 64 |
| | VinePPO | $16 \times$ inter. (16) | 11 | 256 | $\leq 256$ |
| | SPO | $16 \times$ cutpoint (9) | 22 | 128 | $\leq 128$ |
| | SSVPO | $16 \times$ reorder (8) | 44 | 64 | $\leq 64$ |
| GSM8K | RLOO / GRPO / DAPO | $16 \times 4$ | 469 | 64 | 64 |
| | VinePPO | $4 \times$ inter. (128) | 60 | 512 | $\leq 512$ |
| | SPO | $16 \times$ cutpoint (9) | 234 | 128 | $\leq 128$ |
| | SSVPO | $4 \times$ reorder (32) | 469 | 64 | $\leq 128$ |
| MATH | RLOO / GRPO / DAPO | $16 \times 4$ | 407 | 64 | 64 |
| | VinePPO | $4 \times$ inter. (512) | 60 | 512 | $\leq 512$ |
| | SPO | $16 \times$ cutpoint (9) | 205 | 128 | $\leq 128$ |
| | SSVPO | $2 \times$ reorder (128) | 205 | 128 | $\leq 256$ |

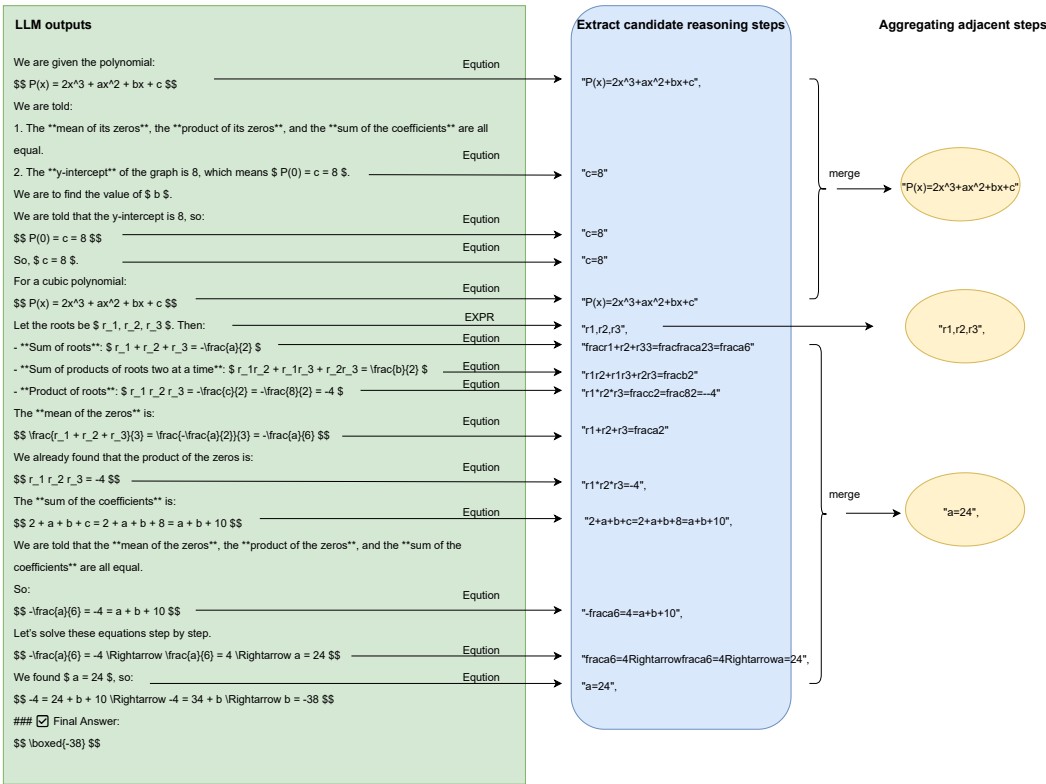

Figure 6: Partition strategy to obtain reasoning steps.

final answer (if any) is recorded. This extraction layer is decoupled from the SSV algorithm itself, facilitating replacement and reuse.

**Aggregation into New Reasoning Chains.** We apply a conservative granularity reduction rule by aggregating (merging) adjacent steps of the same type into a single "merged step," preserving the original order and the semantic continuity of equation/derivation chains. To achieve cross-sample alignment, we classify each merged step using the content of its last original step: we first check whether it contains a parsable numeric solution (e.g., $x = 24$, $b = -38$, $r \approx 1.732$, $y \in \{2,5\}$); if a variable–value pair can be extracted, steps are grouped as the same step by a normalized key (variable name and canonicalized value, with a tolerance for approximate numbers). If no numeric solution is present, we fall back to comparing one side of an equation/inequality only (e.g., $a + c = b + t$ and $e + d = b + t$ are considered the same step because the chosen side is identical), clustering by the algebraically normalized single-side expression. If neither criterion applies, the step is recorded as an isolated entry using its normalized text. This "merge $\rightarrow$ upper-level computation $\rightarrow$ backfill" structure anchors training objectives at a stable granularity, reduces variance introduced by formatting and filler text, and provides robust cross-sample anchors without altering the intra-sample order or the computation of learning signals on merged steps to be new reasoning chains. We provided an example to illustrate the entire processing procedure in Figure 6.

# H SSVPO IN ZERO-RL AND OFF-POLICY SETTINGS

## H.1 OFF-POLICY RL SETTINGS

We now analyse the effect of off-policy sampling, which is closely related to offline RL scenarios where trajectories are collected under a fixed behaviour policy.

**Effect on unbiasedness.** Under standard off-policy correction with proper importance sampling, SSV as a baseline does not alter the expected value of the policy gradient and remains an unbiased estimator. If async or partial rollouts are used as an approximation and the importance weights are not exact, then the overall gradient becomes slightly biased; however, this bias arises from the off-policy approximation itself, not from SSVPO. Importantly, SSV does not introduce any additional bias beyond what off-policy PPO already incurs.

**Effect on variance.** Theoretically, SSV is the conditional expectation of SMC under the sampling distribution and reward, making it the minimum-variance baseline among all baselines depending only on step identity. When the sampling distribution shifts due to off-policy or async rollout, the "strict minimum" becomes "approximately minimum". Nonetheless, SSV continues to provide greater variance reduction than conventional baselines (e.g., value-function baselines or mean baselines), which is particularly beneficial in offline or replay-based settings where variance can be high.

**Effect on convergence.** As in standard PPO, async or partial rollout introduces lag between the behaviour policy and the update policy. This increases the deviation between the empirical gradient and the theoretical gradient, and can cause more noise if the lag grows too large. However, this is a property of PPO-style async sampling itself, not specific to SSVPO. SSV does not worsen this behaviour; it simply inherits the same convergence characteristics as PPO under async or offline-style rollout.

Overall, off-policy or async sampling does not fundamentally conflict with SSVPO: the estimator remains unbiased (with proper importance sampling), retains its variance-reduction benefits, and behaves similarly to PPO in terms of convergence stability in offline and off-policy regimes.

## H.2 ZERO-RL TRAINING SCENARIOS AND OTHER RL DOMAINS

**Zero-RL training scenarios.** The short answer is that SSVPO is potentially applicable in zero-RL training scenarios, but not directly in its current form. At this stage, SSVPO is designed for mathematical reasoning tasks with well-structured formats, e.g., regulated answer extraction, clear step divisions, and consistent symbolic patterns. These assumptions do not fully hold in a zero-RL training scenario where the model lacks any prior long-form reasoning ability. However, SSVPO can be applied after an appropriate cold-start SFT stage, where the model is first trained to produce mathematical reasoning steps in a format that can be reliably extracted and recognized. Once this minimal structure is established, SSVPO can be used for further post-training to enhance reasoning quality, as seen in prior work.

**Beyond LLM post-training (e.g., robotic control).** This is a very insightful question. In principle, SSVPO could be applied beyond LLM post-training, including to domains such as robotic control. Although our work focuses on LLM reasoning, the underlying idea of marginal-contribution credit assignment is inherited directly from cooperative MARL, where such credit mechanisms are well-established and not specific to language models.

In fact, any sequential decision-making problem could, in theory, be compatible with SSV-style credit assignment. The main practical challenge is not the credit formulation itself, but how to construct permutations of trajectories under physical or dynamical constraints. Unlike LLM reasoning, where the set of reasoning steps is discrete and easily reorderable, robotic trajectories exist in continuous state–action spaces and obey strict physical dynamics. Determining how to permute or reorder such trajectories meaningfully, and how to discretise them into a comparable "step set" would be a non-trivial challenge similar to what current spatial-AI and continuous-control methods face today.

Thus, while SSVPO is not limited to LLMs in theory, applying it to embodied or continuous RL settings requires additional research on trajectory representation and feasible permutation schemes.

# I  EXTENDED RESULT ANALYSIS

We conduct three series of comparative case studies, each comprising five cases, to evaluate reasoning accuracy, efficiency, and the trade-off between them, using representative benchmark results.

## I.1  REASONING ACCURACY COMPARISONS

We select five representative cases (Figures 7–11) to analyze typical failure outcomes from Vanilla and GRPO. These failures generally fall into two categories: (i) Underweighted early effective steps: Vanilla and GRPO often apply weak updates to seemingly minor algebraic rewrites or equivalent transformations, which causes the derivation to drift and ultimately terminate with an incorrect answer. (ii) Redundancy and branching accumulation: extended reasoning chains frequently spawn ineffective branches and repeated computations, exhausting the token budget before convergence.

By contrast, SSVPO employs step-level credit assignment to consistently allocate effective return to transitions that meaningfully alter the chains. This suppresses ineffective expansions and enables steady progress in algebraic manipulation and geometric relation reconstruction. Consequently, SSVPO more reliably reaches the correct solution and achieves higher accuracy on complex problems, demonstrating the effectiveness of its fair credit assignment.

---

**AMC23-20** Q：Janet rolls a standard 6-sided die 4 times and keeps a running total of the numbers she rolls. What is the probability that at some point, her running total will equal 3? The final answer can be written in the form \\frac{m}{n}, where m and n are relatively prime positive integers. What is m+n?

| Vanilla | Vanilla+GRPO | Vanilla+SSVPO |
|---|---|---|
| Okay, so Janet is rolling a standard 6-sided die 4 times, and she keeps a running total of the numbers she rolls. The question is asking for the probability that at some point during these 4 rolls, her running total equals 3. Then, we need to express this probability as a reduced fraction m/n and find m + n. Alright, let me try to figure this out step by step.\n\nFirst, <omitted>it's 85 / 216 ≈ 0.393. These are different. Wait, but in the inclusion-exclusion, the calculation of P(A2) is 8/36 = 2/9. However, the total number of sequences where the running total is 3 at the second roll is 8, which includes sequences where it was already 3 at the first roll. Therefore, the number of sequences where it is 3 at the second roll but not at first is 8 - | Okay, so Janet is rolling a standard 6-sided die four times, and she keeps a running total of the numbers she rolls. The question is asking for the probability that at some point during these four rolls, her running total will equal 3. Then we need to express this probability as a reduced fraction m/n and find m + n. Alright, let me try to figure this out step by step.\n\nFirst, <omitted>15 + 10 + 6 = 92. Correct. dp[4][13] = 107. Which is dp[3][12] + dp[3][11] + dp[3][10] + dp[3][9] + dp[3][8] + dp[3][7] = 21 + 22 + 21 + 18 + 15 + 10 = 107. Correct. dp[4][14] = 115. Which is dp[3][13] + dp[3][12] + dp[3][11] + dp[3][10] + dp[3][9] + dp[3][8] = 18 + 21 + 22 + 21 + 18 + 15 = 115. Correct. dp[4][15] = 113. Which is dp[3][14] + dp[3][13] | Okay, so Janet is rolling a standard 6-sided die four times, and she keeps a running total. We need to find the probability that at some point during these four rolls, her running total equals 3. The answer should be in the form m/n where m and n are coprime, and then find m + n. Alright, let's start breaking this down.First, <omitted>Since 49 and 216 are relatively prime, this is the reduced form.### Step 5: Final Answer\n\nThe problem asks for the sum  m + n where the probability is written as \\frac{m}{n} $ in lowest terms.m = 49, \\quad n = 216, \\quad m + n = 49 + 216 = \\boxed{265} |
| 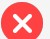 | 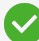 | ✅ |
| Token counts: 16,144 | Token counts: 16,144 | Token counts : 14,053 |

Figure 7: Reasoning accuracy case 1 (AMC23-Question20)

**AIME25-13** Q: Let ABCD be a tetrahedron such that AB=CD= \\sqrt{41}, AC=BD= \\sqrt{80}, and BC=AD= \\sqrt{89}. There exists a point I inside the tetrahedron such that the distances from I to each of the faces of the tetrahedron are all equal. This distance can be written in the form \\frac{m \\sqrt n}{p}, where m, n, and p are positive integers, m and p are relatively prime, and n is not divisible by the square of any prime. Find m+n+p?

### Vanilla

Okay, so I need to find the distance from the point I inside the tetrahedron ABCD to each of its faces. The tetrahedron has edges AB = CD = √41, AC = BD = √80, and BC = AD = √89. The distance is supposed to be equal to all faces, so I think this point I is the incenter of the tetrahedron. But wait, in a tetrahedron, the incenter is the point equidistant from all four faces, right? So I need to find the inradius of this tetrahedron.\n\nFirst<omitted>Then subtract 36,000: 240,800 - 36,000 = 204,800.\n\nSo det(minor1) = 204,800.\n\nNext, compute det(minor2):\n\n|41 -41 -9|\n|80 48 -48|\n|89 39 -89|\n\nLet me compute this determinant. Again, expand along the first row.\n\n41 * det(48, -48; 39, -89) - (-41) * det(80, -48

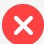

Token counts: 16,144

### Vanilla+GRPO

Okay, so I need to find the distance from a point I inside a tetrahedron ABCD to each of its faces, and this distance is equal for all faces. The tetrahedron has edges AB = CD = √41, AC = BD = √80, and BC = AD = √89. The answer should be in the form m√n / p, with m and p coprime, n square-free, and then find m + n + p. \n\nFirst, <omitted>Second term: -9 * [80*(-89) - (-48)*89] = -9 * [-7120 + 4272] = -9 * (-2848) = 25,632\n\nThird term: -9 * [80*(-39) - (-80)*89] = -9 * [-3120 + 7120] = -9 * 4000 = -36,000\n\nSo, D2 = 215,168 + 25,632 - 36,000 = (215,168 + 25,632) - 36,000 = 240,800 - 36,000 = 204,800.\n\nNext, minor13:\n\n[[41, -41, -9],\n[80, 48, -48],\n[89, 39, -89]]\n\nCompute determinant D3.\n\nD3 = 41 * det[[48, -48], [39, -89]] - (-41) * det[[80

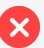

Token counts: 16,144

### Vanilla+SSVPO

Okay, so I need to find the distance from the point I inside tetrahedron ABCD to each of its faces, given the edge lengths. The tetrahedron has AB = CD = √41, AC = BD = √80, and BC = AD = √89. The distance is supposed to be in the form m√n / p, with m and p coprime, and n square-free. Then find m + n + p.\n\nFirst, <omitted>### Step 5: Final Answer\n\nThe inradius is \\frac{20\\sqrt{21}}{63}, so in the form \\frac{m\\sqrt{n}}{p}, we have:- $m = 20\n- n = 21\n- p = 63 Check:- 21 is square-free.- m = 20 and p = 63 are coprime.\n\nSo,m + n + p = 20 + 21 + 63 = \\boxed{104}

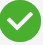

Token counts: 9,899

Figure 8: Reasoning accuracy case 2 (AIME25-Question13)

**AIME25-19** Q: Suppose \\triangle ABC has angles \\angle BAC = 84^\\circ, <omitted> label(\"H\", H, (-.2, -1));label(\"J\", J, (.2, -.8))

### Vanilla

Okay, so I need to solve this geometry problem. Let me first try to understand what's given and what's being asked.\n\nWe have triangle ABC with angles at A, B, and C being 84°, 60°, and 36°, respectively. Then D, E, F are midpoints of BC, AC, and AB. The circumcircle of triangle DEF intersects BD, AE, and AF at points G, H, J respectively. Then, the points G, D, E, H, J, F divide the circumcircle of DEF into six minor arcs. The question is asking for the sum of arc DE plus twice arc<omitted>angle at E is 60°, angle at F is 36°, as established before.\n\nBut the arc DE in the circumcircle of DEF corresponds to twice the angle at F, which is 36°, so arc DE would be 72°, which matches my previous calculation. Similarly, arc EF would be twice the angle at D, which is)

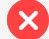

Token counts: 16,144

### Vanilla+GRPO

Okay, so I need to find the value of arc DE + 2*arc HJ + 3*arc FG, where all arcs are measured in degrees on the circumcircle of triangle DEF. The triangle ABC has angles given: angle BAC is 84°, angle ABC is 60°, and angle ACB is 36°. D, E, F are midpoints of BC, AC, AB . respectively. The circumcircle of DEF intersects BD, AE, AF at G, H, J respectively. Then, p <omitted>nAlternatively, maybe the arcs are not as I thought. For example, arc FG could be different. Wait, arc FG is between F and G. In our calculation, arc FG is 72°, which is the same as arc DE. But according to the central angles, arc FD is 120°, arc DG is 48°, so arc FG is arc FD - arc DG - arc something? Wait, no

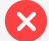

Token counts: 16,144

### Vanilla+SSVPO

Okay, so I need to find the value of the expression: arc DE + 2*arc HJ + 3*arc FG, where each arc is measured in degrees. The problem involves triangle ABC with given angles, midpoints D, E, F of the sides, and the circumcircle of triangle DEF intersecting BD, AE, AF at points G, H, J respectively.\n\nFirst, I should probably sketch the triangle to visualize everything. But since I can't draw here, I'll try to imagine it. Triangle ABC with angles at A: 84°, B: 60°, C: 36°. Then D, E, F are midpoints. So DEF is the medial triangle of ABC. The <omitted>Thus, the expression becomes:72 + 2 \\cdot 24 + 3 \\cdot 72 = 72 + 48 + 216 = 336 ### Final Answer \\boxed{336}

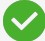

Token counts: 16,043

Figure 9: Reasoning accuracy case 3 (AIME25-Question19)

**AIME25-26** Q：Let A_1A_2\\dots A_{11} be a non-convex 11-gon such that The area of A_iA_1A_{i+1} is 1 for each 2 \\le i \\le 10, \\cos(\\angle A_iA_1A_{i+1})=\\frac{12}{13} for each 2 \\le i \\le 10, The perimeter of A_1A_2\\dots A_{11} is 20. If A_1A_2+A_1A_{11} can be expressed as \\frac{m\\sqrt{n}-p}{q} for positive integers m,n,p,q with n squarefree and \\gcd(m,p,q)=1, find m+n+p+q.

## Vanilla

Okay, let's try to tackle this problem. It's about a non-convex 11-gon with some specific area and cosine conditions. The goal is to find the sum $ A_1A_2 + A_1A_{11} $ expressed in a particular form and then compute $ m + n + p + q $.\n\nFirst, let me parse the problem again to make sure I understand all the conditions.\n\nWe have an 11-gon $ A_1A_2\\dots A_{11} $, which is non-convex. For each $ 2 \\leq i \\leq 10 $, the area of triangle $ A_iA_1A_{i+1} $ is 1. Also, the cosine of the angle <omitted>,n1360*9.61 ≈ -13059.6\n\n2704.\n\nTotal: 9142.8579 + 1191.64 - 13059.6 + 2704 ≈ (9142.8579 + 1191.64) + (-13059.6 + 2704) ≈ 10334.4979 - 10355.6 ≈ -21.1021 < 0.\n\nSo, between 3.1 and 3.2, f(a) crosses from -21 to +469. Let's try a =

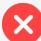

Token counts : 16,144

## Vanilla+GRPO

Okay, so I need to solve this geometry problem about an 11-gon. Let me try to parse the problem step by step.\n\nWe have a non-convex 11-gon labeled A₁A₂...A₁₁. The key information given is:\n\n1. For each 2 ≤ i ≤ 10, the area of triangle A_iA₁A_{i+1} is 1.\n2. For each 2 ≤ i ≤ 10, the cosine of the angle ∠A_iA₁A_{i+1} is 12/13.\n3. The perimeter of the 11-gon is 20.\n4. We need to find A₁A₂ + A₁A₁₁ expressed in the form (m√n - p)/q, where m, n, p, q are positive integers, n is square-free, and gcd(m, p, q) = 1. Then compute m + n + p + q.\n\nFirst, let me try to visualize the problem. There's an 1<omitted>quare root:\n\n9 * sqrt(y² - 20) = 20 - y\n\nSquare both sides:\n\n81(y² - 20) = (20 - y)^2\n\nExpand both sides:\n\n81y² - 1620 = 400 - 40y + y²\n\nBring all terms to

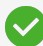

Token counts: 16,144

## Vanilla+SSVPO

Okay, so I need to solve this problem about a non-convex 11-gon. Let me try to parse the problem step by step.\n\nFirst, the problem says: Let A₁A₂…A₁₁ be a non-convex 11-gon such that the area of triangle A_iA₁A_{i+1} is 1 for each 2 ≤ i ≤ 10. Also, the cosine of the angle ∠A_iA₁A_{i+1} is 12/13 for each 2 ≤ i ≤ 10<omitted> From the expression \nS = \\frac{9\\sqrt{5} - 1}{4} We have: m = 9 \n- $ n = 5 \n- $ p = 1 - q = 4 Check \\gcd(9, 1, 4) = 1 , and n = 5 is square-free.Thus, the final answer is:m + n + p + q = 9 + 5 + 1 + 4 = \\boxed{19}

Token counts: 12,724

Figure 10: Reasoning accuracy case 4 (AIME25-Question26)

**AIME25-20** Q：Circle \\omega_1 with radius 6 centered at point A is internally tangent at point <omitted> label(\"H\", H, (-.8, -.8)); label(\"\\omega_1\", (9, -5)); label(\"\\omega_2\", (-1, -13.5))?

## Vanilla

Circle \\omega_1 with radius 6 centered at point A is internally tangent at point B to circle \\omega_2 with radius 15. Points C and D lie on \\omega_2 such that \\overline{BC} is a diameter of \\omega_2 and {\\overline{BC} \\perp \\overline{AD}}. The rectangle EFGH is inscribed in \\omega_1 such that \\overline{EF} \\perp \\overline{BC}, C is closer to \\overline{GH} than to \\overline{EF}, and D is closer to \\overline{FG}. <omitted>\n\n192t² + 12t²(1 + t)^2 - 3(1 + t)^2 = 0\n\nExpand 12t²(1 + 2t + t²) = 12t² + 24t³ + 12t^4\n\nAnd -3(1 + 2t + t²) = -3 - 6t - 3t²\n\nSo overall:\n\n192t² + 12t^4 + 24t³ + 12t² - 3 - 6t - 3t² = 0\n\nCombine like terms:\n\n12t^4 + 24t³ + (192t² + 12t² - 3t²) + (-6t)

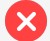

Token counts : 16,144

## Vanilla+GRPO

Circle \\omega_1 with radius $6$ centered at point A is internally tangent at point B to circle \\omega_2 with radius 15. Points C and$D lie on \\omega_2 such that \\overline{BC} is a diameter of \\omega_2$ and {\\overline{BC} \\perp \\overline{AD}}. The rectangle EFGH is inscribed in \\omega_1 such that \\overline{EF} \\perp \\overline{BC}, C is closer to \\overline{GH} than to \\overline{EF}, and $D$ is closer to \\overline{FG} . <omitted>\\frac{12}{\\sqrt{5}} \\text{Area} = \\frac{24}{\\sqrt{5}} \\cdot \\frac{12}{\\sqrt{5}} = \\frac{288}{5} Thus, the area is \\frac{288}{5}, and since 288 and 5 are coprime, the final answer is:m + n = 288 + 5 = \\boxed{293}

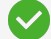

Token counts 14,036

## Vanilla+SSVPO

Circle \\omega_1 with radius 6 centered at point A is internally tangent at point B to circle \\omega_2 with radius 15. Points C and D lie on \\omega_2 such that \\overline{BC} is a diameter of \\omega_2 and {\\overline{BC} \\perp \\overline{AD}}. The rectangle EFGH is inscribed in \\omega_1 such that \\overline{EF} \\perp \\overline{BC}$, <omitted> So:\n\\text{Area} = \\frac{24}{\\sqrt{5}} \\cdot \\frac{12}{\\sqrt{5}} = \\frac{288}{5} Thus, m = 288 , n = 5 , and m + n = 293 .\n\n---\n\n### **Final Answer**\n\n$$\n\\boxed{293}\n

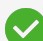

Token counts: 7,835

Figure 11: Reasoning accuracy case 5 (AIME25-Question20)

## I.2 REASONING EFFICIENCY COMPARISONS

We select five representative cases (Figures 12–16) in which all methods obtain the correct answer but differ in efficiency. Vanilla and GRPO typically produce valid yet longer derivations, containing more redundant equivalent steps. In contrast, SSVPO maintains accuracy while more quickly identifying effective steps, reducing unnecessary exploration of the solution space. As a result, it generates shorter and more efficient reasoning chains. This effect is particularly evident in Figure 12–16: although all three methods solve the same problems, SSVPO shows little to no redundant backtracking or repetition, indicating a stronger ability to capture effective steps. By comparison, GRPO and Vanilla often reach the answer with additional equivalence expansions, repeated checks, or excessive exploration.

---

**AMC23-0** Q: Cities A and B are 45 miles apart. Alicia lives in A and Beth lives in B. Alicia bikes towards B at 18 miles per hour. Leaving at the same time, Beth bikes toward A at 12 miles per hour. How many miles from City A will they be when they meet?

---

### Vanilla

Okay, so Alicia and Beth live in cities A and B, which are 45 miles apart. Alicia starts biking from A towards B at 18 mph, and Beth starts biking from B towards A at 12 mph. The question is asking how many miles from city A they will be when they meet each other. Hmm, let me think about how to approach this.\n\nFirst, I remember that when two objects move towards each other, their relative speed is the sum of their individual speeds. So maybe I can use that concept here. <omitted>This means in the time they meet, Alicia covers $ \\frac{3}{5} $ of the total distance.\\text{Distance from A} = \\frac{3}{5} \\times 45 = 27 \\text{ miles}---\n\n### **Final Answer**\\boxed{27}

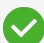

### Vanilla+GRPO

Okay, so Alicia and Beth live in two cities, A and B, which are 45 miles apart. Alicia starts biking from city A towards city B at 18 miles per hour, and Beth starts biking from city B towards city A at 12 miles per hour. The question is asking how many miles from city A they will be when they meet. Hmm, let me think about how to approach this.\n\nFirst, I remember that when two objects move towards each other, their relative speed is the sum of their individual speeds <omitted>\text{ miles}Adding both distances:\n27 + 18 = 45 \\text{ miles}This matches the original distance between the cities, confirming the calculation is correct.### Final Answer \\boxed{27}

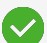

### Vanilla+SSVPO

Okay, so Alicia and Beth live in cities A and B, which are 45 miles apart. Alicia is biking from A to B at 18 mph, and Beth is biking from B to A at 12 mph. They leave at the same time, and I need to figure out how many miles from City A they'll meet each other. <omitted> we can calculate the distance Beth travels in the same time:\\text{Distance} = 12 \\times 1.5 = 18 \\text{ miles}Adding these distances:27 + 18 = 45 \\text{ miles} \n\nThis matches the total distance between the cities, confirming our answer is correct.\n\n---\n\n### **Final Answer** \\boxed{27}

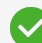

Token counts : 1,854          Token counts : 1,684          Token counts: 1,189

Figure 12: Reasoning efficiency case 1 (AMC23-Question0)

**AMC23-17** Q：There is a unique sequence of integers a_1, a_2, \\cdots a_{2023} such that\n\\[\\tan2023x = \\frac{a_1 \\tan x + a_3 \\tan^3 x + a_5 \\tan^5 x + \\cdots + a_{2023} \\tan^{2023} x}{1 + a_2 \\tan^2 x + a_4 \\tan^4 x \\cdots + a_{2022} \\tan^{2022} x}\\]whenever \\tan 2023x$ is defined. What is $a_{2023}?

### Vanilla

Okay, so I need to find the coefficient a_{2023} in the unique sequence of integers a_1, a_2, \\ldots, a_{2023} that satisfies the equation:\\tan2023x = \\frac{a_1 \\tan x + a_3 \\tan^3 x + a_5 \\tan^5 x + \\cdots + a_{2023} \\tan^{2023} x}{1 + a_2 \\tan^2 x + a_4 \\tan^4 x + \\cdots + a_{2022} \\tan^{2022} x}\\whenever  \\tan 2023x$ is defined.\n\nFirst, I remember that there are formulas for multiple-angle tangent identities. For example, <omitted>3{2023}\n\nNow, \\binom{2023}{2023} = 1, and since $1011$ is odd, we have:a_{2023} = (-1)^{1011} \\cdot 1 = -1 ### Final Answer \\boxed{-1}

✅

Token counts :7,460

### Vanilla+GRPO

Okay, so I need to find the value of a_{2023} in the unique sequence of integers a_1, a_2, \\ldots, a_{2023} that satisfies the equation:\\tan2023x = \\frac{a_1 \\tan x + a_3 \\tan^3 x + a_5 \\tan^5 x + \\cdots + a_{2023} \\tan^{2023} x}{1 + a_2 \\tan^2 x + a_4 \\tan^4 x \\cdots + a_{2022} \\tan^{2022} x}\n$$\n\nwhenever .<omitted>tep 2: Identify the Coefficient of  \\tan^{2023} x We are interested in the coefficient of $ \\tan^{2023} x  in the numerator. This corresponds to the term where the exponent is 2m + 1 = 2023 $, so:2m + 1 = 2023 \\Rightarrow m = 1011 Now compute the coefficient: (-1)^{1011} \\binom{2023}{2023} = (-1)^{1011} \\cdot 1 = -1 ### Final Answer \\boxed{-1}

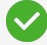

Token counts : 5,293

### Vanilla+SSVPO

Okay, so I need to find the coefficient a_{2023} in the unique sequence of integers a_1, a_2, \\ldots, a_{2023} that satisfies the equation:\\tan2023x = \\frac{a_1 \\tan x + a_3 \\tan^3 x + a_5 \\tan^5 x + \\cdots + a_{2023} \\tan^{2023} x}{1 + a_2 \\tan^2 x + a_4 \\tan^4 x \\cdots + a_{2022} <omitted> \\binom{2023}{2023} Since \\binom{2023}{2023} = 1, and $1011$ is odd, we have:\n(-1)^{1011} = -1 Therefore, the coefficient a_{2023} is: a_{2023} = -1### Final Answer \\boxed{-1}

✅

Token counts: 4,820

Figure 13: Reasoning efficiency case 2 (AMC23-Question17)

**AMC23-21** Q：Points A and B lie on the graph of y=\\log_{2}x. The midpoint of \\overline{AB} is (6, 2). What is the positive difference between the x-coordinates of A and B? The final answer can be written in the form m \\sqrt{n}, where m and n are relatively prime positive integers. What is m+n?

### Vanilla

Okay, let's try to tackle this problem. So, we have points A and B on the graph of $y = \log_2 x$. The midpoint of segment AB is (6, 2). We need to find the positive difference between the x-coordinates of A and B, and express it in the form $m\sqrt{n}$ where m and n are coprime positive integers, then find m + n. Alright, let's break this down.\n\nFirst, since both points A and B are on the graph of $y = \log_2 x$, their coordinates must satisfy that equation. Let me denote the coordinates of A <omitted>nWe are told the answer should be in the form  m\\sqrt{n} , where  m  and  n  are relatively prime positive integers.Here,  m = 4 ,  n = 5 , and since 4 and 5 are coprime, we have:m + n = 4 + 5 = \\boxed{9} --- ### Final Answer \\boxed{9}

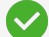

Token counts: 2,465

### Vanilla+GRPO

Okay, so I need to find the positive difference between the x-coordinates of points A and B on the graph of y = $\log_2 x$, given that their midpoint is (6, 2). The answer should be in the form $m\sqrt{n}$, where m and n are coprime, and then find m + n. Alright, let's start breaking this down.\n\nFirst, since points A and B are on the graph of y = $\log_2 x$, their coordinates must satisfy this equation. Let me denote point A as (a, $\log_2 a$) and point B as (b, $\log_2 b$). The midpoint of AB is given as (6, 2). The midpoint formula is ((x_1 + x_2)/2, (y_1 + y_2)/2). <omitted>= |4\\sqrt{5}| = 4\\sqrt{5}Thus, the positive difference between the  x -coordinates is 4\\sqrt{5} , which is in the form m\\sqrt{n}  with  m = 4 ,  n = 5 ### Final Step: Compute m + n = 4 + 5 = \\boxed{9}

✅

Token counts : 2,454

### Vanilla+SSVPO

Okay, so I need to solve this problem where points A and B are on the graph of y = log base 2 of x. The midpoint of segment AB is (6, 2). I have to find the positive difference between the x-coordinates of A and B, and express it in the form $m\sqrt{n}$, then find m + n. Alright, let me think step by step.\n\nFirst, let me recall that the midpoint of two points $(x_1, y_1)$ and $(x_2, y_2)$ is given by $((x_1 + x_2)/2, (y_1 + y_2)/2)$. Here, the midpoint is given as (6, 2). So, if I let point A be (a, $\log_2 a$) and point B be (b, $\log_2 b$), then according to the midpoint formula:\n\n( (a + b)/2 , ($\log_2 a$ + $\log_2 b$)/2 ) = (6, 2)\n\nSo, that gives me two equations:\n\n1. (a + b)/2 = 6 => a + b = 12<omitted> are relatively prime positive integers.Here,  m = 4 ,  n = 5 , so:m + n = 4 + 5 = \\boxed{9}

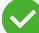

Token counts: 2,403

Figure 14: Reasoning efficiency case 3 (AMC23-Question21)

**MATH500-1** Q: Convert the point $(0,3)$ in rectangular coordinates to polar coordinates. Enter your answer in the form $(r,\theta),$ where $r > 0$ and $0 \le \theta < 2 \pi.$

## Vanilla

Okay, so I need to convert the rectangular coordinate (0, 3) to polar coordinates. Hmm, polar coordinates are given as (r, θ), right? Where r is the distance from the origin to the point, and θ is the angle made with the positive x-axis. Let me recall the formulas for converting between rectangular (x, y) and polar (r, θ) coordinates <omitted> ☑ \n- 0 \leq \theta < 2\pi → \frac{\pi}{2} is within the required range ☑ ## Final Answer boxed{(3, \frac{\pi}{2})}

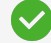

Token counts : 1,719

## Vanilla+GRPO

Okay, so I need to convert the rectangular coordinate (0, 3) to polar coordinates. Hmm, polar coordinates are given as (r, θ), right? Where r is the distance from the origin to the point, and θ is the angle made with the positive x-axis. Let me recall the formulas for converting between rectangular (x, y) and polar (r, θ) <omitted>the angle \theta$ is:\n\n$$\n\theta = \frac{\pi}{2}This is the standard angle measured counterclockwise from the positive x-axis to the point on the positive y-axis.### Step 3: Check the Conditions\n\n- r = 3 > 0 ### Final Answer \boxed{(3, \frac{\pi}{2})}

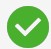

Token counts: 1,879

## Vanilla+SSVPO

Okay, so I need to convert the point (0, 3) from rectangular coordinates to polar coordinates. Hmm, polar coordinates... right, that's where you have a radius r and an angle θ. I remember that the formulas to convert from rectangular (x, y) to polar (r, θ) are r equals the square root of x squared plus y squared, and θ is the arctangent of y over x <omitted> the correct range 0 \leq \theta < 2\pi.\n\n---\n\n### Final Answer \boxed{(3, \frac{\pi}{2})}

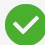

Token counts : 1,495

Figure 15: Reasoning efficiency case 4 (MATH500-Question1)

**AIME25-15** Q: Six points A, B, C, D, E, and F lie in a straight line in that order. Suppose that G is a point not on the line and that AC=26, BD=22, CE=31, DF=33, AF=73, CG=40, and DG=30. Find the area of \triangle BGE.

## Vanilla

Okay, so I need to find the area of triangle BGE. The points A, B, C, D, E, F are in a straight line in that order. There's also a point G not on the line. They've given me several distances: AC=26, BD=22, CE=31, DF=33, AF=73, CG=40, and DG=30. Hmm, okay. Let me try to visualize this. First, maybe I should sketch<omitted> triangle with coordinates**:\text{Area} = \frac{1}{2} \left| x_1(y_2 - y_3) + x_2(y_3 - y_1) + x_3(y_1 - y_2) \right| Plug in the coordinates:\text{Area} = \frac{1}{2} \left| 18(24 - 0) + 58(0 - 0) + 57(0 - 24) \right| = \frac{1}{2} \left| 18 \cdot 24 - 57 \cdot 24 \right| = \frac{1}{2} \left| 432 - 1368 \right| = \frac{1}{2} \cdot 936 = 468 Final Answer \boxed{468}

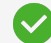

Token counts : 6,002

## Vanilla+GRPO

Okay, let me try to figure out how to solve this problem. So, we have six points on a straight line: A, B, C, D, E, F in that order. There's a point G not on the line. We're given several distances: AC=26, BD=22, CE=31, DF=33, AF=73, CG=40, and DG=30. <omitted>triangle $ \triangle BGE $ using the coordinates:\n- $ B = (18, 0) $\n- $ G = (58, 24) $\n- $ E = (57, 0) $\n\nWe use the **base-height formula**\n- The base $ BE = 57 - 18 = 39 - The height is the y-coordinate of G , which is 24 (since it is the vertical distance from the x-axis) \text{Area} = \frac{1}{2} \times \text{base} \times \text{height} = \frac{1}{2} \times 39 \times 24 = \frac{936}{2} = 468\n**Final Answer:**\boxed{468}

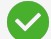

Token counts : 5,419

## Vanilla+SSVPO

Okay, so I need to find the area of triangle BGE. Let me start by trying to visualize the problem. There are six points A, B, C, D, E, F on a straight line in that order. Then there's a point G not on the line. The given distances are AC=26, BD=22, CE=31, DF=33, AF=73, CG=40, and DG=30. I need to find the area of triangle BGE.First <omitted> $ G $ to the x-axis, which is the y-coordinate of G , so 24 \text{Area} = \frac{1}{2} \times \text{base} \times \text{height} = \frac{1}{2} \times 39 \times 24 = \frac{936}{2} = 468 ### Final Answer \boxed{468}

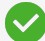

Token counts:5,263

Figure 16: Reasoning efficiency case 5 (AIME25-Question15)

## I.3 REASONING ACCURACY AND EFFICIENCY TRADE-OFF COMPARISONS

We select five cases (Figures 17–21) to examine how SSVPO achieves the correct answer within a reasonable token budget, compared to existing credit-assignment methods that consume fewer tokens but often return incorrect answers. Our observations reveal a consistent pattern across these and other cases: SPO typically generates shorter reasoning traces but, due to early termination or over-aggressive simplification, skips critical intermediate decision points needed to reach the correct solution, leading to failure. By contrast, SSVPO produces traces that are slightly longer yet still compact. It consistently concentrates updates on transitions that genuinely alter the reasoning chain, preserves the necessary intermediate steps, and thereby converges to the correct solution. In practice, SPO often stops after a few aggressive equivalence rewrites that commit to an incorrect path, whereas SSVPO continues past the pivotal turn and successfully arrives at the correct path.

As summarized in Table 2, existing credit-assignment baselines (e.g., SPO) appear more efficient in token usage, but this "efficiency" often arises from pruning mid-chain segments that actually encode crucial decision points, ultimately causing failure. In contrast, SSVPO strikes a more balanced trade-off between accuracy and efficiency: a modest increase in token budget leads to a substantial gain in accuracy.

**AMC23-49** Q: What is the area of the region in the coordinate plane defined by $||x|-1|+||y|-1|\le 1$?

### Vanilla

Okay, so I need to find the area of the region defined by the inequality $||x|-1|+||y|-1|\le 1$. Hmm, that looks a bit complicated with all the absolute values. Let me try to break it down step by step.\n\nFirst,.<omitted>the region.\n\nThe point (1,1): |1 - 1| + |1 - 1| = 0 ≤ 1. So it's inside.\n\nThe point (2,1): |2 - 1| + |1 - 1| = 1 + 0 = 1 ≤ 1. So it's on the boundary.\n\nThe point (1,2): same as above.\n\nThe point (0,1): |0 - 1| + |1 - 1| = 1 + 0 = 1 ≤ 1. So it's on the boundary.\n\nThe point (0.5, 0.5): |0.5 - 1| + |0.5 - 1| = 0.5 + 0.5 = 1 ≤ 1. So this point is on the boundary.\n\nWait, so the region includes points like (0.5, 0.5) which is in the center? Wait, but according to the equation in the uv-plane, the region is |u - 1| + |v - 1| ≤ 1. So (0.5

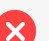

Token counts: 16,144

### Vanilla+SPO

Okay, so I need to find the area of the region defined by the inequality $||x|-1|+||y|-1|\le 1$. Hmm, that looks a bit complicated with all the absolute values. Let me try to break it down step by step.\n\nFirst, I remember that absolute value expressions can often be handled by <omitted>\ symmetric in all four quadrants, and we have already accounted for the entire region in the first quadrant, the **total area** of the region is:\text{Total Area} = 2.n### Final Answer \boxed{2}\

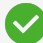

Token counts : 8,736

### Vanilla+SSVPO

Okay, so I need to find the area of the region defined by the inequality $||x|-1|+||y|-1|\le 1$. Hmm, that looks a bit complicated with all the absolute values. Let me try to break it down step by step.\n\nFirst, I remember that absolute value inequalities often define regions that are symmetric in some way. Since there are absolute values around both x and y,, <omitted> This is the area in the **first quadrant**.### Step 3: Total Area Since the region is symmetric in all four quadrants, the total area is:4 \\times 2 = 8.### Final Answer \\boxed{8}

✓

Token counts: 10,042

Figure 17: Reasoning accuracy and efficiency trade-off case 1 (AMC23-Question49)

**AIME24-86 Q**: Alice chooses a set $A$ of positive integers. Then Bob lists all finite nonempty sets $B$ of positive integers with the property that the maximum element of $B$ belongs to $A$. Bob's list has 2024 sets. Find the sum of the elements of A

### Vanilla

Okay, so I've come across this problem where Alice chooses a set A of positive integers, and then Bob lists all finite nonempty sets B of positive integers such that the maximum element of B is in A. It's given that Bob's list has 2024 .<omitted>but that doesn't directly correspond to a sum of distinct 2^{a-1}, because the subtraction isn't allowed in sums of positive integers.\n\nTherefore, perhaps A is a set where each a in A contributes a distinct power of 2, so the sum is just the sum of distinct 2^{a-1}. So, 2^{a-1} must be a subset of the powers of 2

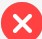

Token counts : 16,144

### Vanilla+SPO

Okay, so I have this problem where Alice chooses a set A of positive integers. Then Bob lists all finite nonempty sets B of positive integers such that the maximum element of B is in A. It's given that Bob's list has 2024 sets. I need to find the sum of the elements of A.\n\nHmm, let's break this down.<omitted>\the set \\( A \\) must be:\\[A = \\{3, 5, 6, 7, 8, 9, 10\\}\\] Summing the elements of \\( A \\):\\[3 + 5 + 6 + 7 + 8 + 9 + 10 = 48\\] Therefore, the sum of the elements of \\( A \\) is:\\[\\[\boxed{48}\\]

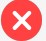

Token counts: 2,701

### Vanilla+SSVPO

Okay, so I have this problem where Alice chooses a set A of positive integers. Then Bob lists all finite nonempty sets B of positive integers with the property that the maximum element of B belongs to A. It turns out Bob's list has 2024 sets. I need to find the sum of the elements of A.\ <omitted>,28 + 64 + 32 + 8\\]\nThe corresponding exponents are 10, 9, 8, 7, 6, 5, 3. Thus, the set \\( A \\) is \\(\\{11, 10, 9, 8, 7, 6, 4\\}\\).The sum of the elements of \\( A \\) is:\\[4 + 6 + 7 + 8 + 9 + 10 + 11 = 55\\] Therefore, the sum of the elements of \\( A \\) is \\(\boxed{55}\\).

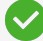

Token counts: 3,084

Figure 18: Reasoning accuracy and efficiency trade-off case 2 (AIME24-Question86)

**AIME25-1 Q**: Find the sum of all integer bases b>9 for which 17_b is a divisor of 97_b.

### Vanilla

Okay, so I need to find the sum of all integer bases \\( b > 9 \\) for which \\( 17_b \\) is a divisor of \\( 97_b \\). Hmm, let me make sure I understand the problem correctly. \n\nFirst, \\( 17_b \\) and \\( 97_b \\) are numbers in base \\( b \\). I need to treat these numbers as base 10 values and see when \\( 17_b \\) divides \\( 97_b \\). That means \\( 17_b \\) should be a factor of \\( 97_b \\) in decimal.\n\nAlright, so let me recall how to convert numbers from base \\( b \\) t<omitted>Wait, let me think.\n\nWait, \\( d = 9 - k \\). Since \\( k \\) is a positive integer (as it's a divisor in the equation \\( 9b + 7 = k(b + 7) \\)), \\( k \\geq 1 \\). Therefore, \\( d = 9 - k \\leq 8 \\).\n\nBut also, since \\( b > 9 \\), from the expression \\( b = \\frac{7(k - 1)}{d} \\), \\( d \\) must be positive. So, \\( d \\) is a positive

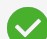

Token counts : 16,144

### Vanilla+SPO

Okay, so I have this problem here: I need to find the sum of all integer bases \\( b > 9 \\) for which \\( 17_b \\) is a divisor of \\( 97_b \\). Hmm, okay. Let me think about how to approach this.\n\nFirst, I know that when a number is written in base \\( b \\), each digit represents a power of \\( b \\). So, for example, \\( 17_b \\) would be equal to \\( 1 \\times b + 7 \\) in decimal<omitted>(invalid as \\( b \\leq 9 \\))Thus, the valid bases are 98 and 19. The sum of these bases is:\\[ 98 + 19 = 117 \\]The final answer is:\\ [\\\boxed{117}\\]

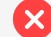

Token counts: 2,130

### Vanilla+SSVPO

Alright, so I have this problem here: I need to find the sum of all integer bases \\( b > 9 \\) for which \\( 17_b \\) is a divisor of \\( 97_b \\). Hmm, okay. Let me try to parse this out step by step.\n\nFirst, I remember that when we write a number in a certain base \\( b \\), each digit represents a power of \\( b \\). So, for example, \\( 17_b \\) would be equal to \\( 1 \\times b^1 + 7 \\times b^0 \\), which simplifies to \\( b + 7 \\). Similarly, \\( 97_b \\) w  <omitted>Thus, we have:- \\( b + 7 = 28 \\) implies \\( b = 21 \\)- \\( b + 7 = 56 \\) implies \\( b = 49 \\)The possible bases are 21 and 49. Summing these bases, we get:\\[ 21 + 49 = 70 \\]Thus, the sum of all integer bases \\( b > 9 \\) is \\(\boxed{70}\\)

Token counts : 4,801

Figure 19: Reasoning accuracy and efficiency trade-off case 3 (AIME25-Question1)

**AIME25-25 Q:** Let $S$ be the set of vertices of a regular 24-gon. Find the number of ways to draw 12 segments of equal lengths so that each vertex in S is an endpoint of exactly one of the 12 segments

### Vanilla

Okay, so I need to figure out how many ways there are to draw 12 segments of equal length in a regular 24-gon, such that each vertex is an endpoint of exactly one segment. Let me try to break this down step by step.\n\nFirst, let me .<omitted>Since there are **two** such step sizes (2 and 10), the total number of matchings is:\n\n4 + 4 = 8 ### Final Answer \\boxed{8}

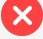

Token counts : 10,100

### Vanilla+SPO

Okay, so I need to figure out how many ways there are to draw 12 segments of equal length in a regular 24-gon, such that each vertex is the endpoint of exactly one segment. Let me try to break this down step by step.\n\nFirst, let me visualize a regular 24-gon <omitted>Summing Up All Valid Cases \text{Total} = 2 + 4 + 8 + 16 + 64 + 1 = \\boxed{95} ### Final Answer \\boxed{95}

Token counts : 13,666

### Vanilla+SSVPO

Okay, so I need to figure out how many ways there are to draw 12 segments of equal lengths in a regular 24-gon, where each vertex is an endpoint of exactly one segment. Hmm, let me start by understanding the problem better.\n\nFirst, a regular 24-gon has 24 vertice, <omitted>2 \\times 8 = 16 \n- k = 4 : 1 \\times 16 = 16 \n- k = 6 : 1 \\times 64 = 64 - k = 12 : 1 \\times 1 = 1  Adding these: 8 + 8 + 16 + 16 + 64 + 1 = \\boxed{113} ### Final Answer \\boxed{113}

Token counts : 15,413

Figure 20: Reasoning accuracy and efficiency trade-off case 4 (AIME25-Question25)

**AIME25-18 Q:** The product\\[\\prod^{63}_{k=4} \frac{\\log_k (5^{k^2 - 1})}{\\log_{k + 1} (5^{k^2 - 4})} = \frac{\\log_4 (5^{15})}{\\log_5 (5^{12})} \\cdot \frac{\\log_5 (5^{24})}{\\log_6 (5^{21})}\\cdot \frac{\\log_6 (5^{35})}{\\log_7 (5^{32})} \\cdots \frac{\\log_{63} (5^{3968})} {\\log_{64} (5^{3965})}\\\]is equal to $\\tfrac mn,$ where m and n are relatively prime positive integers. Find m + n.

### Vanilla

Alright, so I have this product to evaluate:\n\n\\[\n\\prod^{63}_{k=4} \\frac{\\log_k (5^{k^2 - 1})}{\\log_{k + 1} (5^{k^2 - 4})}\\n\\]\n\nAnd the problem says that this product is equal to \\ (\\frac{m}{n}\\) where \\(m\\) and \\(n\\) are coprime positive integers, and I need to find \\(m + n\\).\n\nHmm, okay. So, first, I need to understand what this product is. I ,<omitted> but in fraction form, 93/13.\n\nWait, but hang on, 31 * 3 is 93, times 1/13 is 93/13, which is reducible?\n\nWait, 93 is 3*31, and 13 is prime, so 93/13 is in lowest terms.\n\nThus, m=93, n=13, so m + n=106.\n\nBut wait, now I'm confused because 31 and 1/13 is equal to 2 and 5/13, but 31 * 3 is 93, so 93/13 is equal to 7 and 2/13.\n\nWait, but

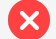

Token counts: 16,144

### Vanilla+SPO

Okay, so I have this product to compute:\n\n\\[\n\\prod^{63}_{k=4} \\frac{\\log_k (5^{k^2 - 1})}{\\log_{k + 1} (5^{k^2 - 4})} = \\frac{\\log_4 (5^{15})}{\\log_5 (5^{12})} \\cdot \\frac{\\log_5 (5^{24})}{\\log_6 (5^{21})} \\cdot \\frac{\\log_6 (5^{35})}{\\log_7 (5^{32})} \\cdots \\frac{\\log_{63} (5^{3968})} {\\log_{64} (5^{3965})}\\n\\]\n\nAnd the problem <omitted> these results together:\\[\\frac{1}{21} \\cdot \\frac{1}{13} \\cdot 3 = \\frac{3} {273} = \\frac{1}{91}\\] Thus, the product simplifies to \\ (\\frac{1}{91}\\), where \\(m = 1\\) and \\ (n = 91\\). Therefore, \\(m + n = 1 + 91 =92\\).\\[\\boxed{92}\\]

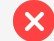

Token counts: 2,816

### Vanilla+SSVPO

Okay, so I need to compute this big product from k=4 to k=63. Each term in the product is a fraction where the numerator is log base k of $5^{k^2 -1}$ and the denominator is log base (k+1) of $5^{k^2 -4}$. \n\nFirst, let me write out the general term:\n\nlog_k(5^{k^2 -1}) / log_{k+1}(5^{k^2 -4}) I remember that log_b(a) = (ln a) / (ln b)<omitted>\\frac{(k - 1)(k +1)}{(k - 2)(k + 2)}\\\]This producttelescopes to:\\[\\frac{31} {13}\\] Combining these results, we get:\\[\\frac{31}{13} \\times 3 = \\frac{93}{13}\\\] Since 93 and 13 are coprime, the fraction is in simplest form. Thus, \\(m = 93\\) and \\(n = 13\\),giving \\(m + n = 106\\).\\ [\\boxed{106}\\]

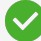

Token counts : 3,871

Figure 21: Reasoning accuracy and efficiency trade-off case 5 (AIME25-Question18)

