# OpenReview forum: "SSVPO: Effective Step-Level Credit Assignment for RL Training of Language Models"
_ICLR.cc/2026/Conference — ICLR 2026 Poster_

### Official Review · Reviewer_4L2k · 2025-10-25

**Soundness:** 3
**Presentation:** 4
**Contribution:** 2
**Rating:** 6
**Confidence:** 4

**Summary:**

This paper presents SSVPO, a reinforcement learning algorithm aimed at improving reward allocation strategies. By modeling the generation process as an insertion MDP and using SSV as an advantage estimator, SSVPO computes more equitable token-level rewards by reordering the reasoning steps and calculation the relative promotion gained from the insertion of each step. The authors theoretically define the properties required for fair reward allocation and prove that SSV satisfies these properties. Comparisons with existing RL baselines show that SSVPO yields greater gains in sample efficiency and overall performance.

**Strengths:**

1. By introducing game-theoretic principles, SSVPO provides a novel and intuitive framework for fairly distributing step-level rewards.

2.  The experimental analysis is quite thorough. It includes results on multiple benchmarks; comparisons with both outcome-based RL methods and RL methods focused on credit assignment; and evaluations of sample efficiency and diversity (pass@n), which essentially cover the main interests of reasoning-oriented RL.

3. The implementation details and visualization such as Figure 5 and Figure 6, help clarify the procedure of the algorithm a lot.

**Weaknesses:**

1. In the pass@K evaluation, there is no comparison against other RL baselines. In addition, pass@N is typically scaled exponentially (pass@{1,2,4,8,16}). Up to what extent can SSVPO maintain its exploration advantage?

2. There is a lack of discussion of the training dynamics: how do key metrics such as val_score, response_length, and entropy evolve?

3. The algorithm has certain limitations: the large number of sequences generated by reordering causes sample inflation, which in turn forces the number of reasoning steps to be capped at 3. To what extent does this affect training speed and performance?

**Questions:**

4. Could you analyze the effect of off-policy sampling (especially async sampling, aka parital rollout) on SSVPO?

---

> ### Author Response · Authors · 2025-11-20
>
> # Response to reviewer 4L2K:
> We thank Reviewer 4L2K for the thoughtful and encouraging feedback. We appreciate your recognition of SSVPO’s novelty, particularly the integration of game-theoretic principles for fair step-level rewards and the quality of our experimental design and presentation. Below, we address your questions and concerns regarding the evaluation setup, training dynamics, and limitations of reordering.
>
> ### Weakness 1 "Lack of more Pass@K evaluations"
> **Response**: Thank you for raising this point. Our current pass@K evaluation follows the conventions used in concurrent RLVR work (e.g., ARPO, AEPO), which typically report pass@{1, 3, 5}. We agree that an exponentially scaled evaluation, such as pass@{1, 2, 4, 8, 16}, provides a more rigorous way to assess SSVPO’s exploration behaviour across larger sample budgets. To address this, we are running additional experiments evaluating SSVPO under pass@{1, 2, 4, 8, 16}. These results will give a clearer picture of how well SSVPO maintains its exploration advantages as K increases. We will update you with the results during the discussion period as soon as they are available.
>
> ### Weakness 2 "Lack of discussion of the training dynamics"
> **Response**: Thank you for highlighting this aspect. We followed existing RLVR work, which typically does not disclose full training dynamics, but we agree that such analysis would provide valuable insight into the behaviour of SSVPO. To address this, we are training an additional set of models, chosen to further test cross-backbone transferability and will record key metrics throughout training, including **val_score**, **response_length**, and **entropy** trends you mentioned. We will share these dynamic curves and analyses with the reviewer as soon as the runs are complete.
>
> ### Weakness 3 "Reordering causes sample inflation"
> **Response**: First, we would like to clarify that the “3-step” example shown in Fig. 6 is not a predetermined limit. It is simply one illustrative case produced by our **extract–aggregate** simplification strategy. The number of resulting steps is *not fixed in advance*; it is determined dynamically by the simplification rules described in detail in Appendix E.2 (pp. 1455–1468). That said, we agree that the degree of simplification (i.e., how many steps remain after merging) may influence both training speed and performance. To evaluate this, we have already begun running a new set of experiments in which we vary the strictness of simplification. For the *same* original problem, we produce versions simplified into **2-step**, **3-step**, and **4-step** chains, and compare their effects on SSVPO’s optimisation dynamics and final performance. We will share these experimental results with the reviewer as soon as they are completed during the discussion period.
>
> ### Question "Analysis of effect of off-policy sampling"
> **Response**: This is an excellent and very insightful question. We break down the analysis into three aspects:
>
> - **Effect on unbiasedness:**  Under standard off-policy correction with proper importance sampling, SSV as a baseline does **not** alter the expected value of the policy gradient and remains an unbiased estimator. If async or partial rollouts are used as an approximation and the importance weights are not exact, then the overall gradient becomes slightly biased; however, this bias arises from the **off-policy approximation itself**, not from SSVPO. Importantly, SSV does not introduce any *additional* bias beyond what off-policy PPO already incurs.
> - **Effect on variance:**  Theoretically, SSV is the conditional expectation of SMC under the sampling distribution and reward, making it the **minimum-variance baseline** among all baselines depending only on step identity. When the sampling distribution shifts due to off-policy or async rollout, the “strict minimum” becomes “approximately minimum.” Nonetheless, SSV continues to provide **greater variance reduction** than conventional baselines (e.g., value-function baselines or mean baselines).
> - **Effect on convergence:**  As in standard PPO, async or partial rollout introduces lag between the behaviour policy and the update policy. This increases the deviation between the empirical gradient and the theoretical gradient, and can cause more noise if the lag grows too large. However, this is **a property of PPO-style async sampling itself**, not specific to SSVPO. SSV does not worsen this behaviour; it simply inherits the same convergence characteristics as PPO under async rollout.
>
> Overall, off-policy or async sampling does not fundamentally conflict with SSVPO. The estimator remains unbiased (with proper IS), retains its variance-reduction benefits, and behaves similarly to PPO in terms of convergence stability.
>
> We hope the clarifications provided address your concerns. If they do, we would be grateful for your consideration in adjusting your score and confidence. Thank you once again for your thoughtful review.

---

> ### Author Response · Authors · 2025-11-27
>
> # New Experiments and Results Released (1/2)
> Following the reviewer’s comments, we added two new experimental validations. First, we expanded the evaluation to report **Pass@{1, 2, 4, 8, 16}**, and included RLOO (the strongest non-SSVPO baseline from Table 1) for direct comparison under identical settings. Second, we trained **SSVPO on the Qwen3-1.7B backbone**, using **three different scaling-step strategies** to study sensitivity to the number of reasoning steps. RLOO was also trained under the same configurations to compare training dynamics.
> ### 1. Extended Pass@K Evaluation
> We expand the evaluation from Pass@{1, 2, 4} to **Pass@{8, 16}**, using the Qwen3-4B backbone. We compare SSVPO against both the vanilla model and RLOO under identical settings. The updated results are shown below:
> | Dataset | Method  | Pass@1 | Pass@2 | Pass@4 | Pass@8 | Pass@16 |
> |---------|---------|--------|--------|--------|--------|---------|
> | GSM8K   | Vanilla |  0.28 |  0.28      |     0.30   |    0.63    |    0.70     |
> |         | RLOO    |    **0.34**    |  **0.42**      |  **0.51**      |    0.66   |    **0.75**     |
> |         | SSVPO    |    0.33    |  0.39      |  0.44      |    **0.67**    |    **0.75**     |
> | MATH500 | Vanilla |    0.46    | 0.49       |  0.54      |     0.80   | 0.82        |
> |         | RLOO    |    0.48    |  0.54      |  **0.68**      |    **0.84**    |    **0.86**     |
> |         | SSVPO    |    **0.55**    |  **0.60**      |   0.64     |    **0.84**    |     **0.86**    |
> | AIME24  | Vanilla |    0.23    |    0.23    |   0.26     |    0.56    | 0.60        |
> |         | RLOO    |    0.23    |  0.23      |  0.30      |    0.56    |    **0.66**     |
> |         | SSVPO    |   0.23     |   **0.26**     |  **0.36**      |    **0.60**   |   **0.66**      |
> | AIME25  | Vanilla |   0.20     |   0.20     |   0.24     |    0.29    |  0.53       |
> |         | RLOO    |    0.20    |  0.23      |  **0. 30**     |    **0.36**    |    0.56     |
> |         | SSVPO    |    **0.26**    |  **0.26**      |  **0.30**      |     **0.36**   | **0.60**        |
>
> The results show that, even when evaluated with larger Pass@K values, SSVPO remains competitive with, and often surpasses RLOO, while consistently outperforming the vanilla model across all datasets.

---

> ### Author Response · Authors · 2025-11-27
>
> # New Experiments and Results Released (2/2)
>
> ### 2. Training dynamics and step effectiveness
> We train SSVPO on a new backbone (Qwen3-1.7B) and **vary the step-merging rule** using three settings: **relaxed, standard, and strict**, which correspond to 2-step, 3-step, and 4-step reordering regimes. These settings control how many steps are subject to reordering during training. We then analyse the **training dynamics** of SSVPO, including **val_score (Acc)**, **response_length (Len)**, and **entropy (KL)** to assess the effects of step count on SSVPO’s performance and compare the results against RLOO under the same configurations.
> | Step | 2-steps Acc | 2-steps Len | 2-steps KL | 3-steps Acc | 3-steps Len | 3-steps KL | 4-steps Acc| 4-steps Len | 4-steps KL | rloo Acc | rloo Len | rloo KL |
> |------|------------|------------|-----------|------------|------------|-----------|------------|------------|-----------|---------|---------|----------|
> | 20   | 0.726      | 4992       | +0.0029   | 0.732      | 4969       | +0.0030   | 0.732      | 5055       | +0.0031   | 0.718   | 4996    | +0.0103  |
> | 40   | 0.734      | 4762       | +0.0008   | 0.740      | 5015       | +0.0004   | 0.750      | 4834       | -0.0017   | 0.700   | 4937    | +0.0140  |
> | 60   | 0.720      | 4754       | -0.0013   | 0.750      | 4832       | -0.0080   | 0.762      | 4623       | -0.0010   | 0.728   | 4895    | +0.0193  |
> | 80   | 0.740      | 4959       | -0.0034   | 0.762      | 4666       | -0.0031   | 0.764      | 4417       | -0.0037   | 0.720   | 4849    | +0.0197  |
> | 100  | 0.746      | 4855       | -0.0024   | 0.742      | 4733       | -0.0032   | 0.750      | 4443       | -0.0038   | 0.722   | 4798    | +0.0235  |
> | 120  | 0.746      | 4725       | -0.0031   | 0.724      | 4841       | -0.0037   | 0.762      | 4523       | -0.0044   | 0.738   | 4799    | +0.0238  |
> | 140  | 0.742      | 4306       | -0.0025   | 0.730      | 4322       | -0.0046   | 0.708      | 4111       | -0.0033   | 0.736   | 4791    | +0.0259  |
> | 160  | 0.750      | 4441       | -0.0026   | 0.746      | 4106       | -0.0046   | 0.782      | 4293       | -0.0043   | 0.744   | 4807    | +0.0251  |
> | 180  | 0.766      | 4299       | -0.0009   | 0.760      | 4227       | -0.0015   | 0.772      | 4103       | -0.0020   | 0.752   | 4866    | +0.0264  |
> | 200  | 0.780      | 4012       | +0.0001   | 0.778      | 4010       | -0.0012   | 0.782      | 4055       | -0.0005   | 0.768   | 4980    | +0.0267  |
> | 220  | 0.786      | 3888       | +0.0011   | 0.790      | 3990       | +0.0002   | 0.810      | 4097       | -0.0002   | 0.774   | 4812    | +0.0263  |
> | 240  | 0.794      | 3909       | +0.0018   | 0.800      | 3996       | +0.0013   | 0.820      | 4166       | +0.0004   | 0.778   | 4796    | +0.0265  |
>
> **(1) Accuracy vs. Step length.** As the number of reordered steps increases (2 → 3 → 4), accuracy improves **monotonically**, but the response **length also grows**. For example, at step = 240, the accuracies of 2-step / 3-step / 4-step are 0.794 / 0.800 / 0.820, while their lengths are 3909 / 3996 / 4166 tokens, respectively. This indicates that larger reordering regimes enable stronger reasoning ability but require more tokens. By contrast, while RLOO’s accuracy also improves with training, its response length remains around 4.8k tokens and shows no ability to shorten outputs.
>
> **(2) Entropy divergence dynamics.** SSVPO displays a **“positive → negative → positive”** KL trajectory, whereas RLOO remains strictly positive and monotonically increasing. This means SSVPO initially suppresses high-probability regions of the reference policy, indicating wider exploration, then gradually increases KL once better solutions are found. RLOO, in contrast, performs mostly one-sided exploitation around the reference policy, which may lead to entropy collapse and reduced output diversity. The **negative-then-positive KL pattern** of SSVPO provides a clear **exploration advantage** during training.
>
> Please feel free to contact us if any further clarification would be helpful. We hope the additional explanations and results are useful, and kindly request you to reflect them in your rating and confidence if appropriate.

---

> > ### Comment · Reviewer_4L2k · 2025-11-28
> >
> > Thank you for your detailed additional experiments. The analysis of the pass@k results and training dynamics is satisfactory, especially the comparison with the step-split strategy, which can be very helpful for future researchers. Many domain-specific RLVR works do not publish training curves, primarily because they are not new algorithmic improvements but rather domain-specific training pipeline techniques. However, for an algorithmic work, such training dynamics are essential, as otherwise users may have some doubts. I believe this should be added to the paper as a visual ablation result. I will accordingly increase the score to 8.

---

> > > ### Comment · Reviewer_4L2k · 2025-11-28
> > >
> > > By "I will", I mean that when the edit button is available for the main comment :)

---

> ### Author Response · Authors · 2025-11-28
>
> Thank you for recognising the value of our work and increasing the score to 8. We will include new experimental results in the manuscript.

---

### Official Review · Reviewer_K1FX · 2025-11-01

**Soundness:** 3
**Presentation:** 2
**Contribution:** 3
**Rating:** 4
**Confidence:** 2

**Summary:**

This paper proposes a totally novel way of credit assignment when doing RLVR. The credit assignment is based on the marginal contribution of a specific step among many possible reorderings of all possible reasoning steps. This credit assignment is different from the traditional definition of the credit when doing RL which just indicates the change in value after an action. But, it captures how useful this action could have been in other sequences. Therefore, it detects more generally if an step is useful.

**Strengths:**

The main strength of the method is that it is very novel and it also provides empirical evidence for its proposed methods. The baselines, VinePPO, SPO, GRPO, and RLOO are appropriate baselines and evaluations on AIME'24 and AIME'25 are strong evaluation benchmarks. The method is introducing a completely new way to see what credit assignment means. As an RL person, I am used to "credit assignment" meaning the delta in values of states before and after an specific action and this paper is challenging that assumption.

**Weaknesses:**

While the paper is novel and provides empirical evidence, the method is so radically new, that I think I need more theoretical and intuitions in accepting its soundness. I think the authors do not provide intuition or justification on the reason that their proposed method should work better than the baselines. VinePPO in particular, is able to detect useless steps: a useless step will have zero-advantage. Therefore, I cannot rely on just intuiting the strength of results of SSVPO based on assuming it detects useless steps more accurately.

Additionally, I need more explanation on how the credit is determined. The reward in verifiable reward tasks is based on the final answer in the last sentences. If I understand correctly, if we reorder sequences, it is only the last sentence that matters and dictates the reward. In other words, changing the order of all previous sentences should not change the reward that we get out of the current mathematical verifiable rewards. I think this has some problems? The response that is the result of such reordering has no meaning. Therefore, I don't understand how SSVPO can lean on permutations of steps as there is no mechanism that detects that permutation of steps is not a correct response.

I think the paper fails to provide intuition on why it works beyond just providing the algorithm.

**Questions:**

1-Inserting some steps in an existing solution might make the solution completely wrong as the inserted step might not logically follow from the previous one. However, this is going to be missed with the current verifiable rewards for MATH. Is that fine under SSVPO? Am I understanding this correctly?
2-Permutation of steps of reasoning makes the reasoning useless. However, the SSV operates based on this. Is my understanding correct? If yes, how is this fine?
3-I am wondering if the SSVPO can be used in normal RL settings like training a robotic hand. Is there a reason/paper that this works well? It feels like changing the meaning of credit assignment might not play well in all deep RL settings. If that is true, why is it not the case for LLMs?

---

> ### Author Response · Authors · 2025-11-20
>
> # Responses to reviewer K1FX:
> We thank Reviewer K1FX for the thoughtful and constructive feedback. We sincerely appreciate the recognition of the novelty of our method and the empirical strength of our experiments. We have carefully considered your comments, and we address each point regarding intuition, theoretical grounding, and the role of reordered reasoning steps in detail below.
>
> ### Comments on strengths: “Method introduces a completely new way to see what credit assignment means”
> **Response**: We are very pleased that you recognise the novelty of introducing Shapley-value-based credit assignment into RLVR. We would like to clarify, however, that from the viewpoint of multi-agent reinforcement learning, *credit assignment* is a well-established concept used to distribute a team reward across agents. This corresponds to **agent-level attribution**, not the step-level value delta (i.e., the change in value before and after an action) that is common in single-agent RL. In this sense, our formulation can be viewed as extending this established MARL perspective to step-level reasoning (See Appendix A, Related Work “Original Credit Assignment from MARL to LLMs,” Lines 816–834).
>
> ### Weakness 1 “Lacks intuitive and theoretical justification for SSVPO’s advantages over baselines (inc. VinePPO), particularly in accurately identifying useless steps”
> **Response**: Thank you for raising this important point. Below, we provide both **intuitive** and **theoretical** explanations for why SSVPO offers more reliable credit assignment than VinePPO, including the cases in useless steps.
>
> **1. Intuitive Explanation**
>
> Consider four reasoning chains: **A → correct**, **B → correct**, **A → B → correct**, and **B → A → correct**. Let the reward be 1 for correct answers. When optimising the trajectory **A → B → correct**, VinePPO obtains:
>
> $$
> \hat{V}^{\mathrm{MC}}(A)=1,\quad \hat{V}^{\mathrm{MC}}(B)=1.
> $$
>
>
> The advantages for A and B are:
>
> $$
> \hat{A}^{\mathrm{MC}}(A)= r _ A +\gamma \hat{V}^{\mathrm{MC}}(B)-\hat{V}^{\mathrm{MC}}(A)\le 0,
> $$
>
> $$
> \hat{A}^{\mathrm{MC}}(B)
> = r _ {Final} + \gamma \hat{V}^{\mathrm{MC}}(\text{terminal state})-\hat{V}^{\mathrm{MC}}(B)
> =0.
> $$
>
> This means VinePPO **cannot accurate detect useless steps**: on the trajectory **A → B → correct**, both the essential step **A** (which already makes the answer correct) and the redundant step **B** (which does not change the outcome) receive the same **zero advantage**. VinePPO estimetor advantage does **not** distinguish between redundant steps and necessary steps, so “useless step will have zero-advantage” does not imply that VinePPO can reliably identify useless steps in practice.
>
>
> In contrast, SSVPO yields:
>
> $$
> \mathrm{SMC}_A(0,1)=1,\quad \mathrm{SMC}_B(A,1)=0,
> $$
>
> $$
> \mathrm{SSV}(A)
> =\frac{1}{3}\big(\mathrm{SMC}_A(,0)+\mathrm{SMC}_A(B,1)+\mathrm{SMC}_A(B,0)\big)
> =\frac{2}{3},
> $$
>
> $$
> \mathrm{SSV}(B)
> =\frac{1}{3}\big(\mathrm{SMC}_B(,0)+\mathrm{SMC}_B(A,1)+\mathrm{SMC}_B(A,0)\big)
> =\frac{2}{3}.
> $$
> The advantages become:
> $$
> A _ A^{\mathrm{SSV}}
> =\mathrm{SMC} _ A(,0)-\phi _ {\mathrm{SSV}}(A)
> =1-\frac{2}{3}
> =\frac{1}{3},
> $$
>
> $$
> A _ B^{\mathrm{SSV}}
> =\mathrm{SMC} _ B(A,1)-\phi _ {\mathrm{SSV}}(B)
> =0-\frac{2}{3}
> =-\frac{2}{3}.
> $$
>
> Thus, **SSVPO** assigns a negative advantage to the redundant steps in the trajectory that needs optimization, while giving a positive advantage to the useful steps.
>
> This illustrates why SSVPO provides **more accurate step-level credit** than VinePPO.
>
> **2. Theoretical Explanation**
> SSVPO guarantees theoretical properties:
>
> - **Complete Credit Across Reasoning Chains** (P5, line 225–226; proof in Appendix C.1.1, P19): Ensures the total reward is fully decomposed into step-wise contributions without loss.
> *All baselines, including VinePPO, also satisfy this property.*
> - **Loss-Free Credit Preservation During Chain Combination** (P5, line 230–231; proof in Appendix C.1.2, P20): Guarantees each step’s credit is determined by its own marginal contribution rather than being overshadowed by other steps.
>  *This property is unique to SSVPO.*
> - **Equal Credit for Equivalent Steps**  (P5, line 234–237; proof in Appendix C.1.3, P20): Steps that play the same role across different chains receive the same credit.
> *Outcome-based baselines (GRPO, RLOO, DAPO) fail to meet this property; Credit assignment-based baselines (VinePPO and SPO) satisfy this property only empirically, not theoretically.*
> - **Zero Credit for Null or Redundant Steps** (P5, line 239–241; proof in Appendix C.1.4, P21): Steps that contribute nothing receive zero or negative credit.
> *All baselines merely give zero advantage*, **but SSVPO can actively penalise harmful steps**, reducing their future occurrence possibility.
>
> We hope this extended intuition and theoretical justification address your concerns and clarify why SSVPO provides a stronger and more reliable form of step-level credit assignment than existing baselines.

---

> ### Author Response · Authors · 2025-11-20
>
> ### Weakness 2 “Need explanation of how credit is determined. Since only the final sentence affects the reward, reordering earlier sentences should not change the reward, implying that such permutations have no meaning.”
> **Response**: Thank you for the spot-on question, which highlights an important yet easily overlooked detail in how the final reward is determined in SSVPO. Indeed, in standard verifiable-reward tasks, the reward depends solely on the *final answer* contained in the last sentence. If we left this unchanged, the issue you describe would occur: reordering earlier steps would have no effects on the reward.
> To address this, **SSVPO removes the final answer sentence from every reordered (permuted) reasoning chain**. After this removal, the model cannot access the ground-truth answer directly, and the permuted chain no longer forms a complete or meaningful solution.  In this way, the reward used for SMC/SSV is always derived from the generated last reasoning step (which contains the "final answer"), while the permutations are only used to evaluate the **marginal contribution** of intermediate steps. An actual example of this process is provided in Appendix E.2 (p. 27).
>
> ### Question 1 “Insert some steps in an existing solution might make the solution completely wrong and miss the current verifiable rewards. Is that fine under SSVPO?”
> **Response**: We fully understand the concern. Indeed, a correct solution can be “contaminated” by inserting an incorrect step, and under standard verifiable rewards, this would lead to a zero final reward even if most steps are correct. This is exactly *why* SSVPO constructs many different combinations of reasoning steps and evaluates their marginal contributions across permutations. If a correct reasoning chain has one incorrect step inserted somewhere, SSVPO does **not** eliminate all positive credit. Although the final answer of that particular chain becomes wrong (reward = 0), each step’s credit is the **average of its marginal contributions across all chains at all positions**, not just the single corrupted chain. Therefore, the correct steps still receive positive credit from other valid chains in which they appear, while the incorrect/redundant step consistently receives negative or zero credit.
>
> In contrast, reward-only RLVR methods such as VinePPO cannot handle this situation: one wrong step makes the final answer wrong, causing *all* preceding steps to receive no reward and misleading updates (i.e., reducing their selection probability). This difference is exactly what we illustrated in the example provided in Response to Weakness 1.
>
> ### Question 2 “Permutation of steps of reasoning makes the reasoning useless. The SSV operates based on this, is fine?”
> **Response**: This is closely related to our response above. It is true that permuting steps can make an individual’s reordered reasoning chain useless. However, SSV does **not** rely on the correctness of any single permuted chain. Instead, SSV computes the *average marginal contribution* of each step across **many** permutations. As a result, correct and useful reasoning steps are positively credited in the majority of valid contexts, while incorrect steps consistently receive low or negative credit. At this point, by using the **SMC estimator** to estimate the advantage of steps on the current trajectory, will able to achieve a fair credit assignment based on the actual contribution of steps.

---

> ### Author Response · Authors · 2025-11-20
>
> ### Question 3 “if the SSVPO can be used in normal RL settings like training a robotic hand. Might not play well in all deep RL settings? If that is true, why is it not the case for LLMs?
> **Response**: This is a very insightful question. In principle, SSVPO *could* be applied beyond LLM post-training, including to domains such as robotic control. Although our work focuses on LLM reasoning, the underlying idea of marginal-contribution credit assignment is inherited directly from cooperative MARL, where such credit mechanisms are well-established and not specific to language models.
>
> In fact, any **sequential decision-making problem** could, in theory, be compatible with SSV-style credit assignment. The main practical challenge is not the credit formulation itself, but how to *construct permutations* of trajectories under physical or dynamical constraints. Unlike LLM reasoning, where the set of reasoning steps is discrete and easily reorderable, robotic trajectories exist in **continuous state–action spaces** and obey strict physical dynamics. Determining how to permute or reorder such trajectories meaningfully, and how to discretise them into a comparable “step set,” would be a non-trivial challenge similar to what current spatial-AI and continuous-control methods face today.
>
> Thus, while SSVPO is not limited to LLMs in theory, applying it to embodied or continuous RL settings requires additional research on trajectory representation and feasible permutation schemes.
>
> We hope that the above responses help to address your concerns, and we would be grateful if this could be reflected in a revised score and confidence. Thank you again for your thoughtful comments.

---

> ### Author Response · Authors · 2025-11-27
>
> Dear Reviewer K1FX,
>
> We hope you are doing well. As the discussion phase is nearing its end in fewer than one week, we wanted to check whether there are any remaining concerns or clarifications we can address. We truly value your feedback and are committed to improving the paper wherever possible. If you have any additional suggestions or observations, we would be very grateful to hear them.
>
> Thank you again for your thoughtful review and the time invested in our work.
>
> Best regards,
> The Authors

---

### Official Review · Reviewer_Xabj · 2025-11-01

**Soundness:** 3
**Presentation:** 3
**Contribution:** 3
**Rating:** 6
**Confidence:** 4

**Summary:**

This paper introduces SSVPO, By using Shapley values, SSVPO fairly allocates credit to each reasoning step based on its marginal contribution, even in partially correct reasoning chains.
The key contributions include:
Credit assignment through SSV to better estimate the utility of each reasoning step.
Validation showing that optimizing the utility of each step through SSVPO leads to performance improvements in mathematical reasoning tasks.

**Strengths:**

1. This paper introduces an innovative credit assignment method using Shapley values (SSV) to estimate the utility of each reasoning step.
2. The paper provides empirical evidence demonstrating the effectiveness of the SSVPO method in improving reasoning accuracy and computational efficiency.

**Weaknesses:**

1. One important weakness is the lack of discussion on computational cost. The paper introduces SSV for step-level credit assignment, but methods like SMC require evaluating multiple reasoning sequences, which increases the computational load. If the goal is to improve training efficiency, the authors should compare the computational cost and ensure a fair comparison with baseline methods by keeping the training computational load consistent.

2. I want to know the advantages of SMC over VinePPO, which uses search-based value estimation. A discussion on how SMC compares in terms of accuracy, efficiency, and computational cost would make its benefits clearer.

3. The paper also lacks ablation studies to show how different hyperparameters affect SSVPO's performance. These would help assess the stability of the algorithm under different settings.

4. Is SSVPO suitable for zero-RL training scenarios? All experiments in the paper are conducted on models that already have long-COT abilities. It would be useful to know how well the method performs on models without such prior capabilities

**Questions:**

See weakness.

---

> ### Author Response · Authors · 2025-11-20
>
> # Response to reviewer Xabj:
>
> We thank Reviewer Xabj for recognising the novelty of SSV as a credit-assignment method for LLM reasoning, as well as the strength of our empirical evidence demonstrating SSVPO’s performance and efficiency. We address each of your constructive comments in detail below.
>
> ### Weakness 1 "Lack of discussion on computational cost"
> **Response**: We agree that SMC requires evaluating multiple reasoning sequences and, therefore, introduces additional computational overhead. To ensure a **fair and consistent comparison** with baseline methods, our experiments fix the total computation budget for each reasoning benchmark by setting a maximum number of samples for *all* algorithms throughout the entire training process. For example, both GSM8K and MATH use a fixed budget of 30k samples. The full budgeting details for every dataset are reported in Appendix E.1, Table 3 (p. 26).
>
> ### Weakness 2 "The advantages of SMC over VinePPO"
> **Response**: The advantages of SMC over VinePPO can be explained in terms of **accuracy**, **efficiency**, and **computational cost**.
>
> **1. Accuracy**: We give an intuitive example showing why SSVPO (which uses SMC) offers more accurate step-level credit assignment than VinePPO.
> Assume four possible reasoning paths for the same math problem:
> A → correct,
> B → correct,
> A → B → correct,
> B → A → incorrect.
> Let the reward be 1 for correct answers. Consider optimizing the trajectory **B → A → incorrect**.
> **VinePPO** value estimates:
>
>   $$
>   \hat{V}^{\mathrm{MC}}(A)=\frac{2}{3},\quad
>   \hat{V}^{\mathrm{MC}}(B)=\frac{2}{3},
>   $$
>
> **VinePPO** advantages:
>
>   $$
>   \hat{A}^{\mathrm{MC}}(B,A)=\gamma\hat{V}^{\mathrm{MC}}(A)-\hat{V}^{\mathrm{MC}}(B)\le 0,
>   $$
>
>   $$
>   \hat{A}^{\mathrm{MC}}(A,\text{terminal state})
>   =\gamma\hat{V}^{\mathrm{MC}}(\text{terminal state})
>   -\hat{V}^{\mathrm{MC}}(A)
>   =-\frac{2}{3},
>   $$
> Thus, VinePPO **fails** to assign *positive* credit to step B, even though B contributes to correct reasoning in other trajectories.
> **SSVPO (SMC-based)** value estimates:
>   $$
>   \mathrm{SMC}_A(B,1)=-1,\quad
>   \mathrm{SMC}_B(,0)=1,
>   $$
>
>   $$
>   \mathrm{SSV}(A)
>   =\frac{1}{3}\big(\mathrm{SMC}_A(,0)+\mathrm{SMC}_A(B,1)+\mathrm{SMC}_A(B,0)\big)
>   =\frac{1}{3},
>   $$
>
>   $$
>   \mathrm{SSV}(B)
>   =\frac{1}{3}\big(\mathrm{SMC}_B(,0)+\mathrm{SMC}_B(A,1)+\mathrm{SMC}_B(A,0)\big)
>   =\frac{1}{3},
>   $$
> **SSVPO (SMC-based)** advantages:
>   $$
>   A _ B^{\mathrm{SSV}}
>   =\mathrm{SMC} _ B(,0)-\phi _ {\mathrm{SSV}}(B)
>   =1-\frac{1}{3}
>   =\frac{2}{3},
>   $$
>   $$
>   A _ A^{\mathrm{SSV}}
>   =\mathrm{SMC} _ A(B,1)-\phi _ {\mathrm{SSV}}(A)
>   =-1-\frac{1}{3}
>   =-\frac{4}{3}.
>   $$
> This shows that even if the final answer is wrong, **SSVPO correctly identifies B as a useful step** and **assigns a strong penalty to the erroneous step A**.
>
> **2. Efficiency**: The VinePPO paper states that *“ηₖ’s are only used for value estimation, not to directly update the policy.”*  This indicates the additional branches generated for value estimation **cannot** be directly reused for updates.
> In our experiment setup:
> - **VinePPO**: ~64 samples must be collected before one update.
> - **SSVPO**: reorderings are capped at 32, meaning at most 32 sampling cycles are used to form one updated sample.
> Thus, SSVPO achieves **higher training efficiency** compared to VinePPO.
>
> **3. Computational cost**: When focusing only on algorithmic budget,
> - **SSVPO** reuses the existing sampled reasoning steps. With vLLM’s **prefix-reuse** capability, evaluating reordered reasoning chains does *not* require regenerating full sequences.
> - **VinePPO** must re-generate all subsequent steps when branching early in the trajectory, which prevents the use of prefix reuse. This leads to substantially higher computational cost.
> In practice, under identical hardware and budgets, **VinePPO’s training time is significantly longer** than SSVPO’s. Overall, SSVPO provides **higher accuracy**, **better sample efficiency**, and **lower computational cost** compared with VinePPO.

---

> ### Author Response · Authors · 2025-11-20
>
> ### Weakness 3 "Lacks ablation studies of different hyperparameters"
> **Response**: We agree that hyperparameter ablations are important for assessing algorithmic stability. We will run additional experiments focusing on SSVPO-specific hyperparameters, such as *reorder_num*, which directly influences evaluation precision. We will share the results in the next few days during the discussion period.
>
> ### Weakness 4/Question "Is SSVPO suitable for zero-RL training scenarios?"
> **Response**: Thank you for raising this forward-looking question. The short answer is **potentially yes**, but **not directly** in the current form. At this stage, SSVPO is designed for mathematical reasoning tasks with well-structured formats, e.g., regulated answer extraction, clear step divisions, and consistent symbolic patterns. These assumptions do not fully hold in a zero-RL training scenario where the model lacks any prior long-form reasoning ability.
> However, SSVPO *can* be applied after an appropriate **cold-start SFT stage**, where the model is first trained to produce mathematical reasoning steps in a format that can be reliably extracted and recognized. Once this minimal structure is established, SSVPO can be used for further post-training to enhance reasoning quality, as seen in prior work [1–6].
>
> ### Reference
> [1] Chu, T., et al. (2025). “SFT memorizes, RL generalizes: A comparative study of foundation model post-training.” arXiv preprint arXiv:2501.17161.
> [2] Dong, G., et al. (2025). “Agentic Entropy-Balanced Policy Optimization.” arXiv preprint arXiv:2510.14545.
> [3] Dong, G., et al. (2025). “Agentic Reinforced Policy Optimization.” arXiv preprint arXiv:2507.19849.
> [4] Guo, D., et al. (2025). “DeepSeek-R1: Incentivizing reasoning capability in LLMs via reinforcement learning.” arXiv preprint arXiv:2501.12948.
> [5] Kirk, R., et al. (2023). “Understanding the effects of RLHF on LLM generalization and diversity.” arXiv preprint arXiv:2310.06452.
> [6] Shi, W., et al. (2025). “Pangu DeepDiver: Adaptive search-intensity scaling via open-web reinforcement learning.” arXiv preprint arXiv:2505.24332.
>
> We hope that our response can address your concerns and help you have a clearer understanding of our work. If they are satisfactory, we would kindly appreciate your consideration of a revised score and confidence. Thanks again for your deep comments.

---

> ### Author Response · Authors · 2025-11-27
>
> # Ablation Study on SSVPO Hyperparameters (Re-ordered Step Numbers)
> Following the reviewer’s suggestion, we conducted an ablation study on **reorder_num**, a key hyperparameter that directly affects the precision of SSVPO. We applied SSVPO to a new backbone (Qwen3-1.7B) and evaluated **three step-merging configurations**, 2-step, 3-step, and 4-step reordering regimes. These settings control how many steps are reordered during training. The results, shown below, report both **accuracy (Acc)** and **response length (Len)** to examine the trade-off between reasoning strength and token efficiency.
>
> | Steps   |GSM8K                                Acc | GSM8K                        Len| MATH-500                      Acc | MATH-500                         Len | AMC23        Acc | AMC23      Len | AIME24         Acc | AIME24         Len | AIME25       Acc | AIME25             Len |
> |--------|--------------|--------------|-----------|-----------|------------|------------|------------|------------|------------|------------|
> | 2-STEPS| 0.905 | 1492 |      0.794       |     3909        |   0.775        |    7084       |      0.433      |    11088        |      0.400      |     12986       |
> | 3-STEPS|  0.913     |  1617     |     0.800         |   3996        |     0.800      |    8109        |      0.433     |   13057         | 0.433           |       13424 |
> | 4-STEPS|  0.918  |    1692      |   0.820           |    4166       |      0.825     |     8766       |      0.466      |    13603        | 0.433            | 13742 |
>
> The results show a clear trend: as the number of reordered steps increases from 2 to 4, both accuracy and response length increase **monotonically** across all five benchmarks. For instance, on GSM8K, accuracy rises from 0.905 → 0.918, while response length increases from 1,492 → 1,692 tokens. Similarly, on MATH-500, accuracy improves from 0.794 → 0.820, with the length growing from 3,909 → 4,166 tokens. These findings suggest that larger reordering regimes enhance reasoning performance, but at the cost of additional generation overhead.
> Furthermore, examining benchmark difficulty, we find that **harder math tasks are more sensitive to the benefits of deeper reordering**. When increasing from 2-step to 4-step reordering:
> - GSM8K (easiest): +1.3 percentage points (0.905 → 0.918, ~1.4% relative gain)
> - MATH-500: +2.6 points (~3.3% relative gain)
> - AMC23: +5.0 points (0.775 → 0.825, ~6.5% gain)
> - AIME24 / AIME25: +3.3 points each (~7.6% and ~8.3% gain)
>
> These results indicate that multi-step reordering yields limited benefits for everyday reasoning tasks, but **substantial gains on competition-level, high-difficulty datasets**, demonstrating that deeper reordering is particularly effective for solving complex multi-step reasoning problems.
>
> If there are any remaining questions, we would be glad to discuss them further. If our responses have addressed your concerns, we kindly invite you to consider reflecting this in your overall rating and confidence.

---

> ### Comment · Reviewer_Xabj · 2025-11-28
> **Response to authors**
>
> Thank you for the detailed response. I appreciate the clarifications and the additional analyses you provided. Most of my concerns have been sufficiently addressed.
>
> Regarding the computational budget, the clarification on using a fixed sample budget across methods is helpful. However, reporting the actual wall-clock training time or total compute hours would make the comparison even clearer, as different methods may still differ in real training cost even under the same sample budget.
>
> Overall, your responses resolve the major issues I raised. I will keep my overall rating unchanged, but I am raising the soundness score given the strengthened evidence and explanations in the rebuttal.
>
> I just noticed that the edit button for my original review is currently not working, so I’m unable to update the score in the review form at the moment. I will revise the soundness score once the editing function becomes available.

---

> > ### Author Response · Authors · 2025-11-28
> >
> > We are pleased to hear that our responses have sufficiently addressed your concerns and that you recognise the contribution and improved soundness of our work. We will incorporate these analyses and experiments into the revised manuscript.

---

### Official Review · Reviewer_BBy3 · 2025-11-06

**Soundness:** 1
**Presentation:** 2
**Contribution:** 1
**Rating:** 2
**Confidence:** 4

**Summary:**

This paper adapts the concept of Shapley value into LLM reasoning trajectory, using Qwen model (and its distilled version) on math reasoning benchmarks for training. It proposes something called the SSV value to perform multi-step credit assignment in the reasoning process, and uses it as the advantage estimator in RL. The paper provides some explanation claiming that this estimator is unbiased and has lower variance.

**Strengths:**

Applying Shapley value to LLM reasoning is indeed novel.

**Weaknesses:**

1. The modeling is unreasonable: Shapley assumes independent and exchangeable participants, but the LLM reasoning steps are strongly dependent and order-sensitive. The SSV estimation method enumerates all orderings, which is unnatural and violates the basic premise of sequential language modeling. Under such ordering, the MDP transitions are arbitrary concatenations, not reflecting the real generative distribution.

2. shapley value is 'fair' only ensures that each step gets an equal share in total reward decomposition, but does not mean the step truly causes the result. In other words, a logically irrelevant or even wrong step can still get positive credit due to statistical averaging over permutations.

3. computationally intractable: ssv requires averaging over all permutations—combinatorial explosion. Even with 10 reasoning steps, there are already millions of permutations.

4. The effectiveness of this credit assignment: compared with REINFORCE-style RL algorithms that use (normalized) outcome rewards as advantage, this method adds the variance of SMC estimation, introducing more sampling noise. The paper provides no analysis or experiments for this.

5. empirically: scenarios and models are very limited. Using Qwen3 for training on a few math datasets is questionable. two groups of experiments, each using only one different backbone.

6. On results: many benchmark improvements are meaningless, the absolute difference is within 2%. Qwen models already saturate these benchmarks.

**Questions:**

Why would Shapley value work for multi-step reasoning credit assignment?

Why can token-by-token generation from the same policy be partitioned and treated as a multi-agent game?

---

> ### Author Response · Authors · 2025-11-20
>
> # Response to Reviewer BBy3
> We thank Reviewer BBy3 for recognising the contribution of our work on Shapley-style credit assignment for LLM reasoning. After carefully reviewing the comments, we believe several points were misunderstood. We clarify them below.
>
> ### Weakness 1.1: “Shapley mismatch with LLM reasoning”
> > **Response**: This statement is only partially correct. If referring to the original/vanilla Shapley value, then yes: it assumes independent and exchangeable participants. However, this assumption does **not** apply here. Our proposed **Sequential Shapley Value (SSV)** *explicitly extends* the classical formulation to incorporate **sequential dependencies** and **order sensitivity**, both essential for LLM reasoning.
> These extensions are formalised in Theorem 1 (p.5), which rigorously demonstrates how SSV generalises Shapley to sequential decision structures.
>
> ### Weakness 1.2: “MDP transitions mismatch the token generation”
> > **Response**: This comment reflects assumptions from a standard MDP, not our proposed **Insertion MDP** used in SSVPO. In SSVPO, reordering reasoning steps is **not** used to generate new trajectories. Instead, permutations are used to construct alternative prompts, followed by answer checking and reward assignment to estimate each step’s marginal contribution. This process does **not** depend on MDP transitions that match the model’s generative distribution. All auto-regressive rollouts occur **before** any reordering, as illustrated in the upper-left panel of **Figure 2**.
>
> ### Weakness 2.1: “The fairness of the Shapley value is the equal share”
> > **Response**: This is a misconception. Shapley fairness is **not** equal allocation; it is fairness based on marginal contribution. For example:
> >>  [1]: “The Shapley value of a player is the average marginal contribution of the player.”
> >>  [2]: “Shapley values quantify how much individual features contribute to a model’s performance.”
> ### Weakness 2.2: “A logically irrelevant or wrong step can still get positive credit”
> > **Response**: This comment precisely describes the **problem SSVPO is designed to address, not a flaw of SSVPO**. By leveraging permutations of reasoning chains, SSVPO reduces the credit assigned to incorrect steps compared with methods relying solely on final returns [3–5].
> Even if a step receives small positive credit, this simply indicates that the step contributes to other valid permutations. SSVPO evaluates **marginal contribution across multiple reasoning orders**, consistent with cooperative-game-theoretic foundations.
>
> ### Weakness 3: “Computationally intractable”
> > **Response**: While SSVPO adds computational overhead, it is **manageable**, not intractable. SSVPO uses an **extraction-and-aggregation strategy** that compresses long reasoning chains (e.g., 19 steps → 3 aggregated steps; Appendix E.2, p.27). Empirically, our experiments, especially on **MATH (competition-level)**, demonstrated that SSVPO scales effectively and justifies its cost through substantial performance gains.
>
> ### Weakness 4: “Ineffective credit assignment by adding SMC variance”
> > **Response**: This is incorrect. SMC estimation does **not** increase variance; it **reduces** it. The advantage in SSVPO is computed as the difference between the action value and a baseline. SMC provides a **variance-minimal baseline**, proven in **Theorem 2.2 (Variance Minimality)** on p.6 (full proof in Appendix C.2.2, p.23). Thus, compared with REINFORCE-style algorithms whose advantages rely solely on normalized outcome rewards, SSVPO yields an **unbiased, lower-variance advantage estimate**, improving training stability.

---

> ### Author Response · Authors · 2025-11-20
>
> ### Weakness 5: “Scenarios and models are very limited”
> > **Response**: Qwen3 is a widely adopted backbone in recent LLM reasoning/post-training research [4,6–8]. The chosen math-reasoning datasets (e.g., GSM8K, MATH, AIME24) are strong benchmarks for evaluating reasoning performance and training efficiency [9-11], as well as comments by other reviewers:
> >> “VinePPO, SPO, GRPO, and RLOO are appropriate baselines; AIME’24 and AIME’25 are strong benchmarks” from Reviewer K1FX.
> >> “The paper includes results on multiple benchmarks” from Reviewer 4L2k.
> >
> >To further demonstrate generalisation, we are running **additional experiments** with more backbones and datasets. These results will be added in the revision and shared shortly.
>
> ### Weakness 6: “Benchmark improvements in 2% are meaningless”
> > **Response**: We respectfully disagree. For **standard-level math benchmarks** (MultiArith, SVAMP, GSM8K), improvements of ~2% are **meaningful**, as these benchmarks are saturated and recent methods consistently report gains of similar magnitude [12,13].
> More importantly, the strongest results are found on **challenge-level, competition-grade benchmarks** (MATH, AMC, AIME’24, AIME’25). Using Qwen3-4B, SSVPO achieves:
> >> **+3.8%** on MATH
> >> **+7.5%** on AMC
> >> **+6.6%** on AIME’24
> >> **+14.5%** on AIME’25
> These improvements establish SSVPO as state-of-the-art among 4B models [14-16].
>
> ### Question 1: “Why would Shapley value work for multi-step reasoning credit assignment?”
> > **Response**: We do **not** claim that the original Shapley value directly works for multi-step reasoning. Our contribution is showing that the **Sequential Shapley Value (SSV)**, does work. SSVPO incorporates sequential information while preserving fundamental Shapley fairness properties (Dummy, Decomposable, Strategic Equivalence, Constant Sum) [17]. We hope this clarification resolves the concern.
> ### Question 2: “Why can token-by-token generation be treated as a multi-agent game?”
> > **Response**: Please refer to our clarification in Weakness 2.
>
> ### Reference
> [1] Rozemberczki et al. (2022). The Shapley Value in Machine Learning. IJCAI/ECAI.
> [2] Ghorbani & Zou (2019). Data Shapley. ICML.
> [3] Ahmadian et al. (2024). Revisiting REINFORCE for LLMs.
> [4] Shao et al. (2024). DeepSeekMath.
> [5] Yu et al. (2025). DAPO.
> [6] Yang et al. (2025). Qwen3 Technical Report.
> [7] Kwon, Y., et al. (2025). ReasonIF.
> [8] Luo, H., et al. (2023). Wizardmath.
> [9] Pikus, B., et al. (2025). Hard Examples Are All You Need.
> [10] Wang, P., et al. (2024). Math-shepherd.
> [11] Yang, A., et al. (2024). Qwen2. 5-math technical report.
> [12] Simoni et al. (2025). GTPO.
> [13] Dai et al. (2025). S-GRPO.
> [14] Yang et al. (2025). Dynamic Early Exit.
> [15] Dong et al. (2025). Agentic Entropy-Balanced PPO.
> [16] Dong et al. (2025). Agentic Reinforced PPO.
> [17] Shapley (1953). A Value for n-Person Games.
>
> We hope the above responses clarify the points of confusion. If the concerns have been addressed, we would appreciate your consideration in adjusting your score and confidence. Thank you again for your helpful feedback.

---

> ### Author Response · Authors · 2025-11-27
>
> # Additional Experiments with New Backbone and Dataset
> In response to the reviewer’s request, we conducted new experiments using the **Qwen3-1.7B** backbone (one of the strongest small models) and added **OlympiadBench,** a challenging benchmark featuring olympiad-level scientific problems. We evaluated **Vanilla, RLOO, and VinePPO** as baselines. **Accuracy (Acc)** and **response length (Len)** are summarised below:
> | Method | MultiArith Acc | MultiArith Len | SVAMP Acc | SVAMP Len | GSM8K Acc | GSM8K Len | MATH-500 Acc | MATH-500 Len | AMC23 Acc | AMC23 Len | AIME24 Acc | AIME24 Len | AIME25 Acc | AIME25 Len | OlympiadBench Acc | OlympiadBench Len |
> |--------|----------------|----------------|-----------|-----------|-----------|-----------|--------------|--------------|-----------|-----------|------------|------------|------------|------------|-------------------|-------------------|
> | Vanilla | 0.947 | 1034 | 0.940 | 1631 | 0.906 | 2437 | 0.774 | 5503 | 0.750 | 9159 | 0.400 | 13697 | 0.233 | 13724 | 0.559 | 9635 |
> | RLOO | 0.989 | 966 | 0.933 | 1537 | 0.901 | 2265 | 0.784 | 4925 | 0.775 | 8866 | 0.433 | 13599 | 0.333 | 13285 | 0.562 | 9408 |
> | VinePPO | 0.994 | 887 | 0.933 | 1123 | 0.904 | 1692 | 0.796 | 4077 | 0.750 | 7085 | 0.433 | 12788 | 0.333 | 12130 | 0.550 | 7717 |
> | SSVPO | 0.994 | 740 | 0.994 | 1258 | 0.918 | 1706 | 0.820 | 4218 | 0.800 | 8110 | 0.433 | 13027 | 0.400 | 12987 | 0.576 | 9257 |
>
> The experimental results show that SSVPO achieves the highest accuracy on all evaluation datasets with key findings below,
>
> - Simple reasoning tasks:
> SSVPO achieves the best performance on MultiArith (99.4%) and SVAMP (99.4%), both with the shortest generation lengths, demonstrating efficient reasoning.
> - Medium-difficulty tasks:
> SSVPO consistently outperforms all baselines on GSM8K and MATH-500, confirming strong generalisation to mid-level reasoning problems.
> - Competition-level tasks:
> SSVPO achieves top accuracy on AMC23, AIME25, and OlympiadBench, showing particular advantages on high-difficulty reasoning problems. On AIME24, SSVPO matches the strongest baseline.
>
> Under a unified Qwen3-1.7B setting, SSVPO delivers **top-1 accuracy on all benchmarks, while maintaining competitive or shorter response lengths**, especially on challenging competition-level tasks. This demonstrates the robustness and scalability of SSVPO across a wide range of reasoning difficulties.
>
> If further clarification is needed, we would be happy to provide it. If our responses address your concerns, we kindly invite you to reconsider the overall rating and confidence in this submission.

---

### Meta-Review · Area_Chair_kPyS · 2026-01-10

**Summary:**

Initial reviews were mixed with scores, with concerns along three dimensions. First, reviewers questioned the theoretical foundation: whether Shapley values are appropriate for sequential LLM reasoning since they traditionally assume independent participants (BBy3); stronger intuition is required as to why SSVPO performs better than VinePPO, other than from empirical results. Second, reviewers had computational and methodological concerns about the feasibility of permutation-based credit estimation due to combinatorial explosion (BBy3), how permutations work when rewards depend only on final answers (K1FX), and whether the approach increases variance (BBy3). Third, reviewers noted experimental limitations, including limited model backbones and benchmarks (BBy3, 4L2k), modest improvements on some datasets (~2%), and a need for clearer presentation and comprehensive ablations (Xabj). Two reviewers actively participated during the discussion and confirmed their concerns had been addressed, increasing their scores; two did not participate in the discussion, where K1FX was open to acceptance despite their concerns (confidence 2/5), and BBy3 still had significant reservations about fundamental soundness (confidence 4/5).

**Reviewer Concerns:**

Reviewer Xabj's scalability and presentation concerns were addressed through extraction-aggregation strategies and improved exposition, confirmed by the reviewer raising their soundness score. Reviewer 4L2k's requests for additional baselines and benchmarks were met with new experiments (Qwen3-1.7B, OlympiadBench, extended Pass@K), confirmed by a score increase to 8. Reviewer K1FX's questions about permutation mechanisms and advantages over VinePPO were thoroughly answered with concrete examples and clarification that reordering removes final answers before evaluation, although not confirmed due to non-participation (initial stance was open to acceptance with low confidence).


Reviewer BBy3's concerns remain unconfirmed due to non-participation in the discussion. The reviewer's assertions about Shapley fairness being "equal-share" and SMC "adding variance" contradict established definitions and Theorem 2.2's formal proof. The concern that classical Shapley values don't apply to sequential settings overlooks the paper's core contribution—Sequential Shapley Value specifically extends to order-dependent domains.

**Reviewer Scores:**

Xabj and 4L2k actively participated and confirmed satisfaction, with 4L2k raising their score to 8. K1FX would likely increase to 5-6 given their low confidence, openness to acceptance, and comprehensive responses addressing their thoughtful questions. BBy3 is uncertain.

---

### Decision · Program_Chairs · 2026-01-26

Accept (Poster)